# MEHGT-LKG: Multimodal edge-enhanced heterogeneous graph transformer with LLM-driven knowledge graph for stock trend prediction

## Abstract

Stock trend prediction plays a central role in optimal investment decision-making, and has attracted extensive research from both investors and institutions. Although recent studies have employed graph structures to model the complex relationships among financial entities, the corresponding models fail to efficiently capture semantically rich edge features across heterogeneous entities, thereby limiting the ability to fuse and align multimodal data such as market indicators, financial events, and heterogeneous graph structures. Therefore, in this paper, we propose a Multimodal Edge-Enhanced Heterogeneous Graph Transformer with LLM-driven Knowledge Graphs (MEHGT-LKG) for stock trend prediction. Specifically, we first fine-tune a large language model (LLM) by using instruction tuning datasets to design a financial event-centric knowledge extraction agent (FinEX). Subsequently, we encode the structured tuples generated from FinEX into financial event-centric knowledge graphs (FEKGs) and then construct multimodal heterogeneous graphs by incorporating multimodal information. Finally, we design a Multimodal Edge-Enhanced Heterogeneous Graph Transformer (MEHGT) to fully encode a series of semantically enriched multimodal heterogeneous graphs spanning different time horizons. MEHGT models edge-level features through type-specific encoders and integrates them into both multi-head attention and message propagation, significantly enriching the representation of relational semantics and target nodes. Extensive experimental results and trading simulations on multiple real-world datasets demonstrate the superior performance of the proposed approach beyond other state-of-the-art models.

## 1 Introduction

The stock market is a core component of the financial system, providing capital allocation functions for enterprises and investment opportunities for individuals. However, due to its high volatility, complex influencing factors, and nonlinear dynamics, forecasting stock trends remains a challenging and important research area.

Stock price fluctuations are typically driven by two major types of factors: intrinsic market signals and external shocks from related financial entities. The former includes trading behaviors and technical indicators; the latter involves financial events, company announcements, and policy changes. To better improve prediction accuracy and provide more informed investment decisions, integrating multimodal information and capturing complex market dynamics has become a key research direction Cheng et al. (2022); Liu et al. (2024c); Sheng et al. (2024); Huang et al. (2024).

Among various multimodal representation approaches, knowledge graphs have attracted increasing attention due to their ability to structurally represent relationships among financial events Zhao et al. (2022); Wang et al. (2023). However, traditional knowledge graph construction methods rely on predefined schemas and use deep learning models to extract entities and relations from financial texts. These approaches are heavily dependent on fixed rules and templates, making it difficult to capture the diverse and complex expression of financial events. Notably, with advances in Natural

Language Processing (NLP), large language models (LLMs) have shown exceptional abilities in semantic understanding, knowledge reasoning, and information extraction in many fields Wang et al. (2024); Li & Sanna Passino (2024). Therefore, fine-tuning LLMs to extract key financial events and structured tuples is crucial for constructing accurate and reliable financial event-driven graphs (FEKGs).

To further learn expressive representations and model complex relationships from knowledge graphs, graph-based learning has been increasingly applied to stock trend prediction owing to its ability to capture the complex dependencies among entities Hsu et al. (2021); Li et al. (2024); Liao et al. (2024). Heterogeneous Graph Neural Networks (HGNNs) as advanced variants of Graph Neural Networks (GNNs), distinguish between node and edge types, enabling deeper modeling of complex financial interactions. However, existing HGNNs often overlook encoding edge features, which carry rich semantic information about financial events and capital flows Zhang et al. (2023); Ma et al. (2024); Liu et al. (2024b). Stock price movements are often triggered and propagated along relations—such as financial events, correlations, northbound capital flows, and margin-trading activities—rather than by isolated node attributes. These signals or messages are naturally formulated as edge-level features and propagated through the message-passing process. Thus, we propose a multimodal edge-enhanced Heterogeneous Graph Transformer (MEHGT) that incorporates structured financial event tuples as edge features into both attention calculation and message passing, enabling more effective integration of multimodal data and improving stock trend prediction.

In summary, to address these issues, we propose a Multimodal Edge-Enhanced Heterogeneous Graph Transformer with LLM-driven Knowledge Graphs (MEHGT-LKG) for stock trend prediction. Specifically, we first fine-tune LLM by using instruction datasets to design a financial event-centric knowledge extraction agent (FinEX). Subsequently, we leverage FinEX to construct FEKGs and build multimodal heterogeneous graphs by incorporating multimodal information with sliding windows. Finally, we design MEHGT to encode edge features via type-specific encoders into attention and message passing, enhancing relational semantics and target node representations within multimodal heterogeneous graphs. The contributions are summarized as follows:

- We design the FinEX agent by fine-tuning LLM with instruction-based dataset. It generates financial events and structured tuples from financial texts accurately, supporting the automatic construction of FEKGs.
- Based on FEKGs, we construct multimodal heterogeneous graphs within a sliding time window by integrating trading data, market indicators, and other relevant information.
- We propose MEHGT, which explicitly incorporates edge-level features into both the attention computation and message passing process. By modeling financial relations and actions, the model leverages multimodal data to capture entity associations and information flow patterns, thereby enhancing stock trend prediction.
- To verify the superiority of the proposed model, extensive experiments are conducted with the state-of-the-art baselines on multiple real stock datasets.

## 2 PROBLEM DEFINITION

We formulate the problem of predicting the trend of stocks for excess return as a classification task. The objective of this research is to leverage constructed multimodal financial heterogeneous graphs during $w$ days to predict the rise or fall of the target stock at trading day $t + 1$ ($w$ denotes the actual time window). We represent these dynamic heterogeneous graphs as $G_{1:T} = \{G_1, G_2, \ldots, G_T\}$. Each graph $G_t = (V_t, E_t, R_t)$ at time $t$ contains five types of nodes: $V_t = \{V_t^{KS} \cup V_t^{OE} \cup V_t^{HK} \cup V_t^F \cup V_t^L\}$ and seven types of edges $E_t = \{E_t^{Corr} \cup E_t^{HI} \cup E_t^{Long} \cup E_t^{Short} \cup E_t^{SRE} \cup E_t^{ERS} \cup E_t^{KS} \cup E_t^{OE}\}$. Meanwhile, $R_t$ denotes seven types of relations corresponding to edges. We have $(u \overset{r}{\leftrightarrow} v, u \overset{r}{\rightarrow} v) \in E_t$, where $r \in R_t$ is the relation type and $\{u, v\} \in V_t$. The input features of graph during time window includes node attributes $X_t^V$ and edge attributes $X_t^E$. Finally, let $Y^{t+1}$ be the predicted targets of a key stock at time $t$. Given the heterogeneous graph $G_t$, the node attributes $X_t^V$ of the node set $V_t$, and the edge attributes $X_t^E$ of the edge set $E_t$, the aim of our model is to forecast the trend of a key stock in the next time point $\hat{Y}_{t+1}(s)$, using the proposed MEHGT model (denoted as $f_\theta$):

$$\hat{Y}_{t+1}(s) = f_\theta(G_t, X_t^V, X_t^E), \tag{1}$$

# 3 THE PROPOSED METHOD

The architecture of the proposed methodology, MEHGT-LKG is shown in Figure 1. It comprises three main stages: fine-tuning LLM for knowledge extraction, multimodal heterogeneous graphs construction, and designing MEHGT for graph learning and stock trend prediction.

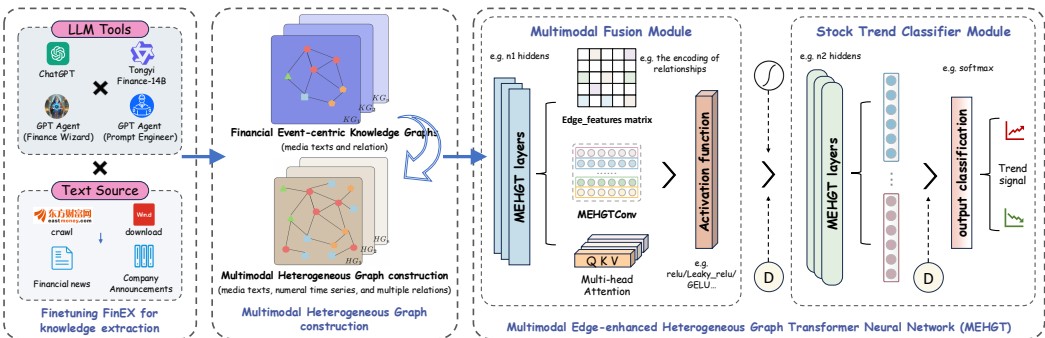

Figure 1: Graphical illustration of the proposed methodology MEHGT-LKG.

## 3.1 DESIGNING FINEX AGENT FOR KNOWLEDGE EXTRACTION

To extract financial events and structured tuples, we design an LLM Agent, namely FinEX.

High-quality instruction datasets are critical for LLM-based information extraction, yet remain scarce in the financial domain. To address this, we construct an instruction dataset (in Figure 2) by collecting financial news and company announcements, and use GPT tools to extract structured financial events. Detailedly, guided by optimized prompts, ChatGPT-4 serves as a financial analyst, generating key events and structured tuples. These outputs are further refined by the Finance Wizard Agent based on domain knowledge and validated by experts to ensure accuracy and completeness. Notably, the dataset preserves both triplets, such as ⟨ Kunlun Tech CO.,LTD — plans to acquire — YOOZOO GAMES CO.,LTD ⟩ and event pairs like ⟨ CATL CO.,LTD — experiences a severe explosion ⟩ allowing flexible representations for different event structures. By capturing multi-entity relations and single-entity events, this dual-format design improves semantic precision and completeness in extraction.

Building on the constructed instruction-based dataset, we fine-tune Qwen model to design FinEX agent. An overview of the fine-tuning process is shown in Figure 2. Each training sample includes both the event description and its corresponding structured tuples, which effectively reduces hallucination during large model reasoning and improves the reliability of outputs. We choose Tongyi-Finance-14B (TF-14B), a domain-specific variant of Qwen-14B pre-trained on extensive financial corpora, as the base model Bai et al. (2023). Fine-tuning is performed with LoRA in the Llama-Factory framework Zheng et al. (2024), updating a small subset of parameters while keeping the backbone frozen to greatly reduce

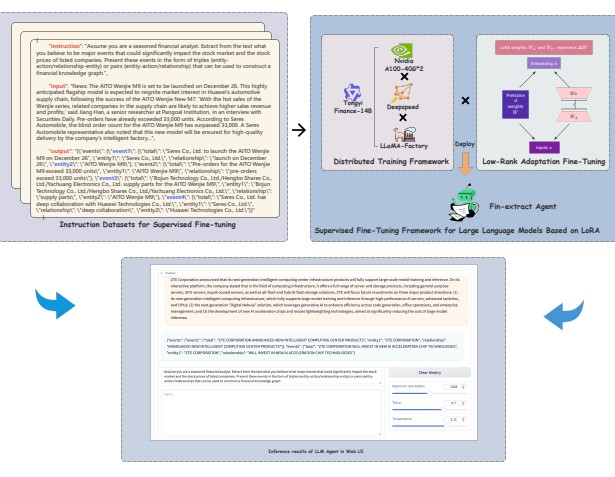

Figure 2: The procedure of fine-tuning LLM to build the FinEX Agent.

memory and computation costs. And training is performed with the DeepSpeed framework on NVIDIA A100 GPUs, ensuring efficient handling of long instruction texts. Finally, FinEX supports

accurate, large-scale extraction of key financial events and structured tuples from raw news and announcements, providing essential inputs for constructing the financial event-centric knowledge graphs (FEKGs).

## 3.2 Multimodal Heterogeneous Graphs Construction

Financial event-centric knowledge graphs are the foundation to build multimodal heterogeneous graphs. We leverage the FinEX Agent to process financial news and announcements from 2021 to 2024 about selected stocks to construct FEKGs. The structured tuples include entities such as companies, products, and key individuals, along with relations such as mergers, investments, and short-selling events. Events may involve either a single entity or a pair of entities; in both cases, we represent them in a unified ⟨head, relation, tail⟩ form, where single-entity events are encoded as self-loop on the same entity. To ensure data quality, entities and relationships were standardized to avoid duplication caused by inconsistent naming.

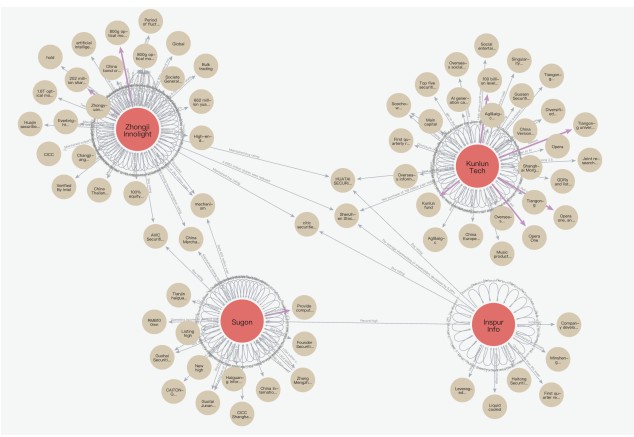

Figure 3: An example subgraph of FEKGs.

Multimodal heterogeneous graphs extend conventional heterogeneous graphs by incorporating multimodal sources. Additional node types, such as Northbound Trading, margin financing, and securities lending, are introduced to represent key financial activities. Edge types are enriched with stock co-movement correlations and capital flow information to reflect long-short market dynamics. Finally, a temporal sequence of multimodal heterogeneous graphs is generated via sliding time windows. These graphs effectively capture intricate financial interactions across modalities, laying a strong foundation for multimodal fusion, graph learning, and trend prediction. The demo of the subgraph developed with Neo4j is shown in Figure 3

## 3.3 MEHGT for Stock Trend Prediction

MEHGT is specifically designed to align the representation of multimodal heterogeneous graphs with the stock-trend prediction task. Each edge type in our graph encodes a distinct financial mechanism: financial semantic relations, northbound trading and margin-trading activities, and correlation between stocks. It explicitly integrates edge-level features into both the attention computation and message passing process, and further conducts a more comprehensive and effective multimodal fusion across both nodes and edges during graph representation learning. The overall architecture is illustrated in Figure 4.

**Heterogeneous mutual attention:** Given a target node $t_1$ of type $\tau(t_1)$ (i.e., key stock) and a source node $s \in N(t_1)$ in a heterogeneous subgraph, we compute their attention via a multi-head mechanism. To better capture relation semantics, edge features are incorporated through a type-specific transformation and scaling function. Specifically, each target node $t_1$ and its neighbor $s$ are transformed into a Query vector and a Key vector, respectively.

$$Q_i(t_1) = Q\text{-}Linear_{\tau(t_1)} \left( H^{(l-1)}[t_1] \right), \tag{2}$$

$$K_i(s_1) = K\text{-}Linear_{\phi(s_1)} \left( H^{(l-1)}[s_1] \right), \tag{3}$$

$$K_i(s_2) = K\text{-}Linear_{\phi(s_2)} \left( H^{(l-1)}[s_2] \right), \tag{4}$$

where $Q\text{-}Linear_{\tau(t_1)}$, $K\text{-}Linear_{\tau(s_1)}$, and $K\text{-}Linear_{\tau(s_2)}$ represent the linear projection functions for the Query and Key vectors. $H^{(l)}$ denotes the node embedding of the $l$-th layer, with $H^{(0)}$ being the initial node embedding.

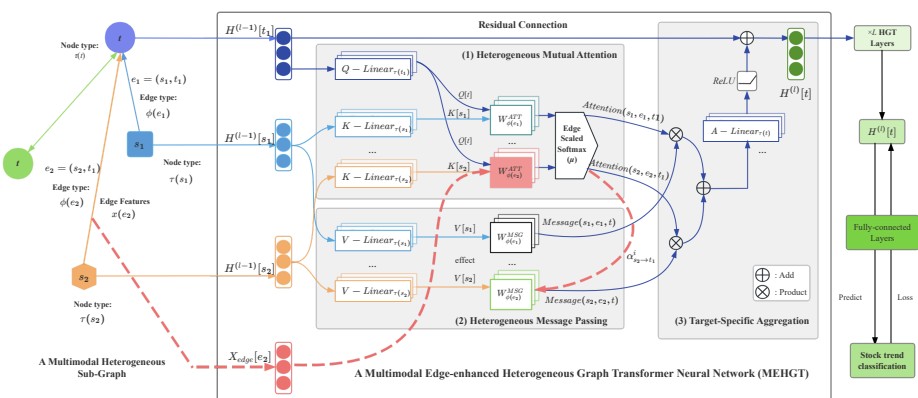

Figure 4: Overview of the MEHGT framework.

The similarity between $Q_i(t_1)$ and $K_i(s)$ is calculated as the attention weight between them:

$$\text{ATT-head}^i(t_1, e_1, s_2) = \left(K_i(s_2)W^{ATT}_{\phi(e_2)}Q_i(t_1)^T\right) \cdot \frac{\mu(\tau(t_1), \phi(e_2), \tau(s_2))}{\sqrt{d}} \cdot f(X_{\text{edge}}[e_2]) \quad (5)$$

where $W^{ATT}_{\phi(e_2)}$ is the edge-based transformation matrix for capturing the semantic information of different types of edges between $t_1$ and $s_2$, and $f(X_{\text{edge}}[e_2])$ represents how the edge feature $X_{\text{edge}}$ influences the attention score. The edge feature $X_{\text{edge}}[e_1]$ is used to scale the attention weight with the specific edge type, which helps refine the attention mechanism by considering financial relationships and actions or short events from graphs. The representation of $ATT - head^i(t_x, e_y, s_z)$ is similar to the above.

Finally, the attention vectors between each node pair are obtained by concatenating the $h$ attention heads. Then, for each target node $t$, the attention vectors from all its neighboring nodes $N(t)$ are gathered:

$$\text{Attention}(s, e, t) = \underset{\forall s \in N(t)}{\text{Softmax}} \left( \underset{i \in [1,h]}{\|} \text{ATT-head}^i(s, e, t) \right), \quad (6)$$

where $\underset{i \in [1,h]}{\|}$ is the concatenating function.

**Heterogeneous message passing:** A message operator is employed to pass messages between various nodes such as stocks, financial entities, and stock markets. The multi-head message is computed by the following process. The source node $s_2$ is projected into a message vector using a linear transformation:

$$\text{MSG-head}^i(s_2, e_2, t_1) = \alpha^i_{s_2 \to t_1} \cdot \left(\text{M-Linear}^i_{\phi(s_2)} H^{(l-1)}(s_2) \times W^{\text{MSG}}_{\phi(e_2)}\right), \quad (7)$$

where the function M-Linear$^i_{\phi(v_s)}$ is the linear projection function corresponding to the ith message head, and $W^{\text{MSG}}_{\phi(e_{s,t})}$ is the edge-type transformation matrix. The final step is to concatenate h message headers to get the $Message(s_2, e_2, t_1)$ for each node pair:

$$\text{Message}(s_2, e_2, t_1) = \underset{i \in [1,h]}{\|} \text{MSG-head}^i(s_2, e_2, t_1). \quad (8)$$

The representation of MSG-head$^i(s_1, e_1, t_1)$ and $Message(s_1, e_1, t_1)$ is similar to the aforementioned.

Actually, incorporating edge features into both the attention and the message channels serves two complementary roles in MEHGT. In the heterogeneous mutual attention module, the multimodal edge vector $X_{\text{edge}}[e]$ acts as a soft relational gate that modulates how much information flows from a source node s to a target node $t$ along a specific financial relation. By scaling the similarity term in Eq. (5), edge features encode whether a relation corresponds to, for example, event-driven

shocks, numeric correlation patterns, long/short positions, and capital-flow interactions, and thus selectively amplify or suppress the corresponding attention weights. In contrast, the heterogeneous message passing module uses the same edge information to modulate what content is propagated: the edge-type-specific transformation $W_{\phi(e)}^{\text{MSG}}$ defines distinct feature subspaces for different financial relations, so that messages carried by financing edges, stock–event edges, and correlation edges are transformed differently before aggregation. This joint design allows MEHGT to simultaneously control the intensity of information flow and to encode relation-specific transformations in the representation space, which is crucial for capturing multimodal financial relations in practice.

**Target-specific aggregation:** To update embedding of a key stock node $t$, this module uses multi-headed attention and message passing to refine its representation from neighboring nodes.

$$\widetilde{H}^{(l)}[t] = \oplus_{\forall s \in N(t)} \text{Attention}(s, e, t) \cdot \text{Message}(s, e, t). \tag{9}$$

where $\widetilde{H}^{(l)}[t]$ denotes the updated embedding of the target node, which aggregates the information of all neighboring nodes. The updated embedding of the key stock node is projected to its type-specific distribution:

$$H^{(l)}[t] = \text{A-Linear}_{\phi(t)}\big(\sigma(\widetilde{H}^{(l)}[t])\big) + H^{(l-1)}[t], \tag{10}$$

where $\text{A-Linear}_{\phi(t)}$ is a type-aware linear projection, $\sigma(\cdot)$ denotes the nonlinear activation function. The module incorporates the residual structure, where $H^{(l-1)}[t]$ is the node embedding of the target node in the $(l-1)^{th}$ layer.

(4) **Target forecasting network and optimization:** Given the learned representations of target nodes from the MEHGT model, we employ a shallow neural network to predict stock trend. The output is defined as:

$$\hat{Y}_s = \text{softmax}(\text{NN}_f(W_n H^{(l)}[t] + b_n)), \tag{11}$$

where $\text{NN}_f$ represents a shallow neural network with two fully connected layers, and $W_n \in \mathbb{R}^{d_s \times 2}$, $b_n \in \mathbb{R}^2$ are the weight matrix and bias, respectively.

The model is trained with cross-entropy loss: $d_l$ is the number of target categories. In this work, we set the $d_l = 2$.

$$\mathcal{L}\int\int_{\text{target}} = -\sum_{s \in \mathcal{V}_T} \sum_{c=1}^{d_l} Y_{t+1}(s) \ln \hat{Y}_{t+1}(s), \tag{12}$$

where $\hat{Y}_{t+1}(s)$ is the ground-truth label of $c_{th}$ price movement category for stock $s$, which is marked as 1 for the "up" price movements, 0 for the "down" movement, respectively. $\mathcal{V}_T$ denotes the set of target nodes.

Hence, after forecasting networks, MEHGT can effectively leverage the learned node representations from multimodal heterogeneous graph to predict stock trend.

## 4 EXPERIMENTS

### 4.1 EXPERIMENT SETTINGS

#### 4.1.1 DATASETS.

We select stocks from CSI300 and CSI500 to construct datasets Duan et al. (2025); Zhou et al. (2025), which include AI and renewable energy sectors. Data are collected from Wind (numerical) and Eastmoney (textual), spanning Jan 5, 2021 – Mar 29, 2024. We split the datasets into mutually exclusive training/validation/testing sets in the ratio of 7:2:1. Moreover, the original dataset, weights of FinEX, code of MEHGT-LKG, and implementation details will be provided in our GitHub.

#### 4.1.2 COMPARED METHODS.

To show the performance of our proposed model, we compare MEHGT-LKG with SOTA methods. We select the following models as the baseline for comparison: (1) Time series modeling methods : Informer Zhou et al. (2021), TCN Bai et al. (2018), CNN-LSTM Vidal & Kristjanpoller (2020), Time-MoE Xiaoming et al. (2025), and KRONOS Shi et al. (2026); (2) Graph-based modeling methods: GAT Velickovic et al. (2018), HGT Hu et al. (2020), MAC Ma et al. (2023b), MDGNN Qian et al. (2024), COGRASP Li et al. (2025), and ENHANCER Chen et al. (2025)

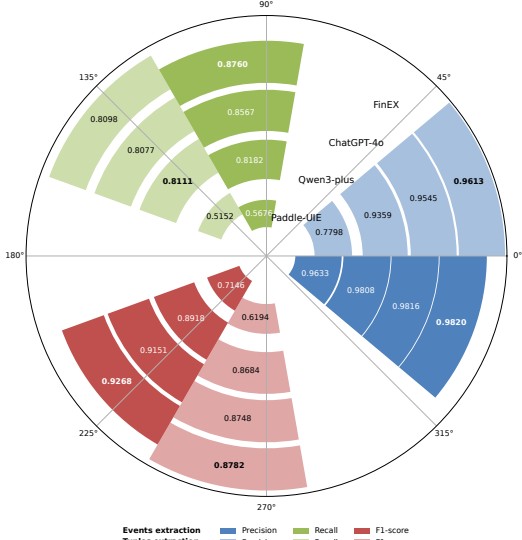

Figure 5: Performance comparison of FinEX against other NLP models (darker and lighter hues represent events and tuples extraction tasks)

### 4.1.3 EVALUATION METRICS.

Following previous study Zeng et al. (2018); Wadden et al. (2019), we use Precision, Recall, and F1 score for two NLP tasks (events extraction and tuples extraction). And we select Accuracy (ACC), Matthew's Correlation Coefficient (MCC), precision, recall, F1 score and Area Under Curve (AUC) to evaluate stock trend prediction performance. Among them, MCC serves main metric, as provides a balanced measure of performance under noisy and imbalanced financial time series and is widely regarded as more informative than Accuracy or F1 for such tasks (Chicco & Jurman (2020); Koa et al. (2024)). During backtesting, we choose Cumulative Return Rate (CRR), Maximum Drawdown (MDD), and Sharpe Ratio (Sharpe) to assess the profitability Liu et al. (2024b); Ma et al. (2024).

### 4.2 COMPARISON RESULTS OF NLP MODELS

We conduct experiments on a manually annotated news dataset to evaluate FinEX agent in event and tuple extraction by comparing it with baselines including Paddle-UIE (0.5B ), Qwen3-plus (235B), and ChatGPT-4o (200B). The results are shown in Figure 5. From the center outward, the four concentric rings correspond to Paddle-UIE, Qwen3-plus, ChatGPT-4o, and FinEX. The radial length serves as a direct indicator of performance magnitude, where a longer radius signifies superior extraction quality. Additionally, darker and lighter hues distinguish between Event and Tuple extraction tasks, respectively.

FinEX outperforms all baselines in both event and tuple extraction tasks, and exhibits exceptional performance in generating rigorously structured triples in JSON. It reaches precision scores of 0.9820 and 0.9613, respectively, and reaches F1 score of 0.9268 and 0.8782. This may be attributed to fine-tuning the LLM with instruction data, which significantly enhances its capacity to comprehend lengthy financial news and announcements deeply, accurately extract key financial events, and generate structured tuples in JSON format aligned with standard requirements for constructing knowledge graphs. ChatGPT-4o and Qwen3-plus, though competitive, occasionally generate malformed or redundant JSON outputs. Paddle-UIE struggles with long, unstructured financial texts due to its schema-constrained decoding, leading to low performance.

Additionally, all LLM-based models show strong extraction ability due to their language understanding and generalization. FinEX, with only 14B parameters, offers high performance and fast inference, making it suitable for deployment. Notably, designed as a standalone module, FinEX exhibits robust plug-and-play capabilities compatible with both heterogeneous and homogeneous graphs. We have open-sourced the full pipeline to facilitate immediate download and deployment for the community, thereby fostering the adoption and development of FinEX.

### 4.3 MAIN COMPARISON RESULTS

We compare MEHGT-LKG with a range of state-of-the-art baselines, including time-series models and graph-based models. As shown in Table 1, MEHGT-LKG consistently achieves superior perfor-

mance across all datasets. From Table 8 and Table 9, we can see notable MCC scores of 0.3718 on Inspur, 0.3681 on IFLYTEK, and 0.3337 on Sungrow.

Compared with time-series methods, our model significantly outperforms them, as these methods focus on single-stock sequences and fail to capture inter-stock dependencies. Time-MoE, despite employing a Mixture-of-Experts (MoE) architecture pre-trained with financial data, shows limited generalization during zero-shot inference when facing unseen data distributions, leading to sub-optimal results. KRONOS, the latest advancement in financial time series forecasting, leverages large-scale autoregressive pre-training on massive K-line corpora and a specialized tokenization strategy. After fine-tuning from our data, it outperforms most time-series baselines and remains competitive against some graph-based models. However, due to the lack of explicit graph topology and multimodal information, KRONOS still underperforms MEHGT. Among graph-based models, MAC leverages sentiment features via a GCN framework, but has weak expressive power due to simplistic feature aggregation. HGT demonstrates stronger performance by employing type-aware multi-head attention over heterogeneous graphs. MEHGT-LKG builds upon this by further injecting edge-level features into the attention calculation and message-passing processes, enhancing its capacity to model complex financial relations. MDGNN benefits from dynamic multi-relational modeling with Transformers, but lacks multimodal integration, particularly of financial events. COGRASP achieves competitive results by mining stock co-occurrence relationships and text information, though its retrieval-based graphs often contain noise. ENHANCER effectively addresses distribution shifts via its Temporal and Relational Meta-Learners (TML and RML), delivering better performance. The proposed MEHGT-LKG not only achieves more precise semantic tuples extraction via FinEX compared to co-occurrence methods, but crucially, it explicitly incorporate these semantic edge features into the attention calculation and message passing within the multimodal heterogeneous graph, thereby achieving excellent predictive performance. Overall, compared with the baselines, our method has the following advantages.

- We design FinEX by fine-tuning an LLM to accurately extract structured tuples of financial events, enabling the construction of FEKGs with rich domain-specific semantics.
- We construct multimodal heterogeneous graphs within a sliding time window by integrating trading data, market indicators, and other relevant information.
- The MEHGT model explicitly incorporates edge-level features into both the attention computation and the message passing process, enabling deeper multimodal fusion and enhancing stock trend prediction performance.

Table 1: Average prediction performance of different methods across all datasets.

| Model | ACC | MCC | Precision | Recall | F1 | AUC |
|---|---|---|---|---|---|---|
| *Time-series models* | | | | | | |
| Informer | 0.6218 | 0.2340 | 0.5737 | 0.5701 | 0.5612 | 0.5871 |
| TCN | 0.5844 | 0.1827 | 0.5436 | 0.5753 | 0.5235 | 0.5484 |
| CNN-LSTM | 0.5747 | 0.2178 | 0.5391 | 0.6860 | 0.5768 | 0.5652 |
| Time-MoE | 0.6033 | 0.2192 | 0.5530 | 0.5994 | 0.5709 | 0.6107 |
| KRONOS | 0.6189 | 0.2462 | 0.5635 | 0.6323 | 0.5889 | 0.6196 |
| *Graph-based models* | | | | | | |
| GAT | 0.6163 | 0.2591 | 0.5480 | 0.6822 | 0.6214 | 0.6200 |
| HGT | 0.6272 | 0.2743 | 0.5668 | 0.6831 | 0.6137 | 0.6093 |
| MAC | 0.5985 | 0.2314 | 0.5702 | 0.5882 | 0.5520 | 0.5897 |
| MDGNN | 0.6353 | 0.2754 | 0.5941 | 0.6240 | 0.5902 | 0.5980 |
| COGRASP | 0.6056 | 0.2728 | 0.5851 | 0.6696 | 0.6079 | 0.6198 |
| ENHANCER | 0.6338 | 0.2702 | 0.5675 | 0.6353 | 0.6091 | 0.6175 |
| **MEHGT-LKG(ours)** | **0.6559** | **0.3191** | **0.6010** | **0.6861** | **0.6361** | **0.6562** |

## 4.4 MARKET TRADING SIMULATION

To further evaluate the profitability of our method, we conduct an investment simulation. Figure 6 presents the cumulative return curves on six representative stocks during the backtesting. MEHGT-LKG outperforms all baselines, remaining the highest equity curve throughout the trading period under bullish and bearish conditions. Especially, on Zhongji Innolight, it attains a CRR of 274.49% and on Inspur, it attains 104.50%. And MDGNN also exhibits competitive performance.

Detailedly, MEHGT-LKG achieves the highest return among all baseline models, and consistently delivers positive returns throughout the entire backtesting period. Notably, on upward-trending

stocks (e.g., Zhongji Innolight), MEHGT-LKG captures strong buy signals driven by accurate trend prediction and achieves a remarkable cumulative return. Meanwhile, for stocks experiencing downward trends (e.g., CATL), the model generates timely exit signals and flexibly adjusts positions, resulting in a solid and positive return.

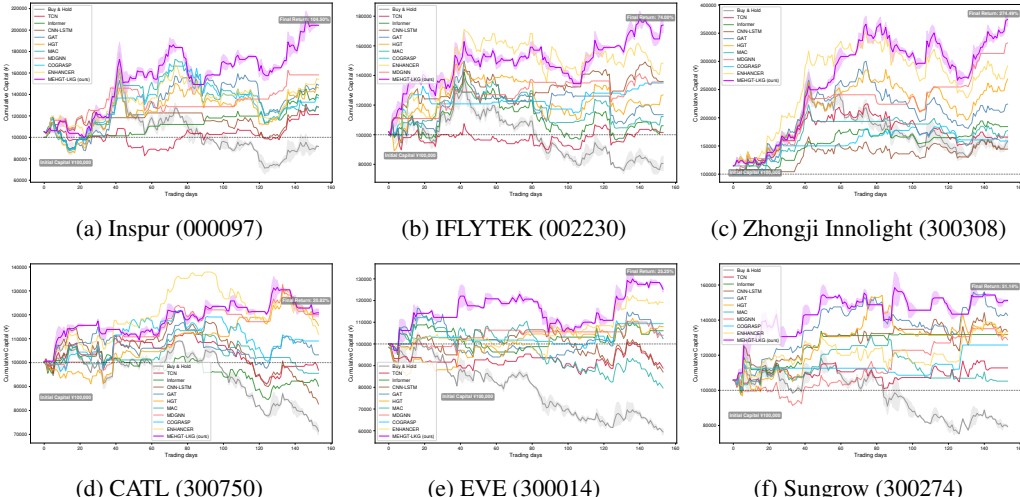

| (a) Inspur (000097) | (b) IFLYTEK (002230) | (c) Zhongji Innolight (300308) |
|---|---|---|
| (d) CATL (300750) | (e) EVE (300014) | (f) Sungrow (300274) |

Figure 6: Simulated trading performance of all models during backtesting (The results on other datasets are shown in the Appendix. F).

## 4.5 ABLATION STUDY

In this section, several ablation experiments are performed to examine the effectiveness of each component of MEHGT-LKG.

As shown in Table 2, MEHGT-LKG model achieves the best performance. The method removing financial text data performs worse, verifying that structured tuples extracted from financial texts enrich graph semantics and enhance trend prediction. Notably, the variant without edge features performs comparably to the one w/o financial text data, as the edge features in MEHGT-LKG are primarily constructed from event-centric financial text. And the variant w/o numerical indicators underperforms, confirming that market indicators and trading features serve as essential signals for stock-level inference.

These observations indicate that MEHGT does not simply benefit from adding more parameters to a generic transformer architecture; instead, the gains come from encoding task-specific financial mechanisms at the edge level. When event-driven edges or their features are removed, the model loses its ability to distinguish structurally similar but semantically different relations (e.g., normal price co-movements vs spillovers triggered by financial events), leading to degraded MCC and weaker trading performance.

## 4.6 HYPERPARAMETER ANALYSIS

We use Sankey diagram to analyze the impact of key layer hyperparameters in MEHGT-LKG. The analysis covers the hidden dimensions of three MEHGT layers and one Linear layer. In Figure 7, distinct colored bars represent the various layers, with annotated values corresponding to specific hidden unit dimensions. These layers are interconnected by left-to-right flows that form complete hyperparameter configuration paths. Each path denotes a specific hyperparameter combination, and MCC performance. The color of these flows intuitively reflects predictive performance, where darker shades denote superior prediction results for that combination.

The best combination—128–128–64–16—achieves the highest MCC of 0.372, suggesting a balanced architecture with good learning and classification abilities. In contrast, over-parameterized settings (e.g., 256–128–128–16, MCC=0.218) and under-parameterized ones (e.g., 64–32–32–8, MCC=0.240) perform worse, likely due to overfitting and limited representation capacity, respectively. Additionally, configurations with sharp reductions between layers (e.g., 256–32–32–32,

Table 2: Comparison results of ablation analysis across selected stock datasets (The results on other datasets are shown in the Appendix).

| Methods | Inspur (000977) | | | | | | CATL (300750) | | | | | |
|---|---|---|---|---|---|---|---|---|---|---|---|---|
| | ACC | MCC | Precision | Recall | F1 | AUC | ACC | MCC | Precision | Recall | F1 | AUC |
| w/o events | 0.6169 | 0.2328 | 0.6250 | 0.5333 | 0.5755 | 0.5967 | 0.6104 | 0.2728 | 0.5306 | 0.7879 | 0.6341 | 0.6121 |
| w/o edge feats | 0.6175 | 0.2570 | 0.6270 | 0.5500 | 0.5860 | 0.6175 | **0.6227** | 0.2744 | 0.5608 | 0.7734 | 0.6502 | 0.6286 |
| w/o indicators | **0.6364** | 0.2757 | 0.6145 | 0.6800 | 0.6456 | 0.6409 | 0.6104 | 0.2614 | **0.5319** | 0.7576 | 0.6250 | 0.5989 |
| MEHGT-LKG | 0.6234 | 0.3718 | **0.6883** | **0.7713** | **0.7184** | **0.6322** | 0.6039 | 0.2748 | 0.5243 | **0.8182** | **0.6391** | **0.6307** |
| | **IFLYTEK (002230)** | | | | | | **EVE (300014)** | | | | | |
| w/o events | 0.6364 | 0.2893 | 0.5843 | **0.7324** | 0.6500 | 0.6302 | 0.6688 | 0.2492 | 0.5102 | 0.4808 | 0.4950 | 0.6110 |
| w/o edge feats | 0.6287 | 0.2799 | 0.5724 | 0.7269 | 0.6405 | 0.6186 | 0.6457 | 0.2601 | 0.5187 | 0.4995 | 0.5089 | 0.6222 |
| w/o indicators | 0.6169 | 0.2546 | 0.5652 | **0.7324** | 0.6380 | 0.5817 | 0.6623 | 0.2595 | 0.5000 | 0.5385 | 0.5185 | 0.6283 |
| MEHGT-LKG | **0.6818** | 0.3681 | 0.6375 | 0.7183 | **0.6755** | **0.6930** | **0.6883** | 0.3166 | 0.5357 | 0.5769 | 0.5556 | 0.6640 |
| | **Zhongji Innolight (300308)** | | | | | | **Sungrow (300274)** | | | | | |
| w/o events | 0.6494 | 0.3136 | 0.7119 | 0.5316 | 0.6087 | 0.6508 | 0.6429 | 0.3307 | 0.5625 | **0.8060** | **0.6626** | 0.6174 |
| w/o edge feats | 0.6589 | 0.3351 | **0.7204** | 0.5562 | 0.6277 | 0.6625 | 0.6234 | 0.2873 | 0.5512 | 0.7815 | 0.6465 | 0.6159 |
| w/o indicators | 0.6299 | 0.2615 | 0.6146 | **0.7468** | 0.6743 | **0.6518** | 0.6299 | 0.2710 | 0.5610 | 0.6866 | 0.6174 | 0.6021 |
| MEHGT-LKG | **0.6688** | 0.3375 | 0.6591 | 0.7324 | **0.6946** | 0.6485 | **0.6753** | 0.3337 | 0.6393 | 0.5821 | 0.6094 | **0.6523** |

256–32–32–16, and 256–32–32–8) also perform poorly, possibly because abrupt dimensional drops hinder the model's ability to extract high-level features effectively.

# 5 CONCLUSION

In this work, we propose MEHGT-LKG for stock trend prediction. By fine-tuning Qwen with custom instruction datasets, we design a financial event-centric knowledge extraction agent (FinEX), and build financial event-centric knowledge graphs. Then, with sliding windows, these graphs are integrated with numerical indicators from multiple sources to form a sequence of multimodal heterogeneous graphs. Finally, we design MEHGT to learn a series of semantically enriched multimodal heterogeneous graphs spanning different time horizons. MEHGT models edge-level features through type-specific encoders and integrates them into both multi-head attention and message propagation, significantly enriching the representation of relational semantics and target nodes.

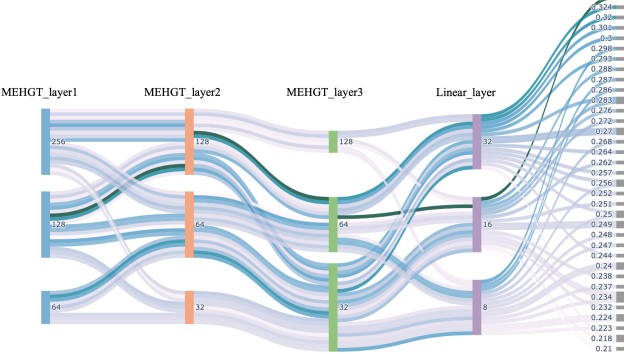

Figure 7: Hyperparameter analysis based on Sankey graph.

In the future, we will extend the framework to larger cross-sector and cross-market universes to assess generalization. Second, more fine-grained architectural variants will be explored to isolate their respective contributions to modeling information intensity and information content. Finally, the core concept of MEHGT—intergrating multimodal edge semantics into attention to modulate information flow—holds significant potential for frontier domains like AI for Science (e.g., modeling physicochemical bond properties in molecular graphs) or LLM-based Multi-Agent Systems (e.g., weighting agent influence based on interaction content). We will explore the potential transferability of the proposed edge-enhanced mechanism to other heterogeneous graph domains in the future. We will explore the application of MEHGT in other domains.

# 6 ETHICS STATEMENT

We confirm that our study adheres to the ICLR Code of Ethics . Our research uses publicly available financial datasets and does not involve any personally identifiable information or human subjects. We have carefully considered potential risks, including privacy, fairness, and security issues, and

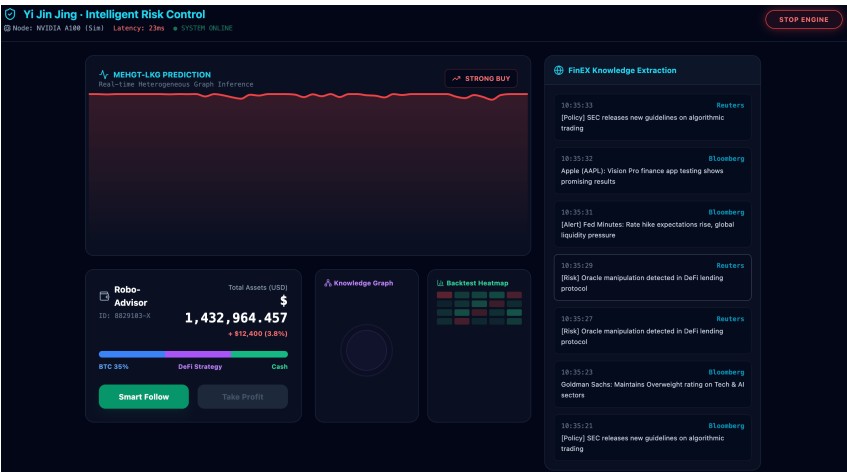

Figure 8: YiJinJing Intelligent Investment Service Platform

found no evidence of harm or ethical concerns. All data preprocessing and experimental protocols comply with legal and research integrity requirements.

## 7 REPRODUCIBILITY STATEMENT

We provide an anonymous repository link in the supplementary material containing partial experimental demonstrations and code to facilitate initial reproducibility checks. While the current version includes core training scripts and example data preprocessing pipelines, the full set of model details, hyperparameters, and datasets will be publicly released after the review process to ensure compliance with anonymity and data license requirements. We will also include complete documentation and environment configurations in the final release to support exact replication of our results.

## A EXTRA NOTE

A portion of the source code and data is available at anonymous repository `https://anonymous.4open.science/r/MEHGT-LKG-D17C`.

Currently, we have integrated the core functionalities of the MEHGT-LKG framework to construct an intelligent investment service platform (as shown in Figure 8). This system extracts structured knowledge from real-time news streams to automatically update the backend FEKGs, enabling MEHGT to perform online stock prediction for investment decision support. Notably, FinEX functions as a highly transferable plug-and-play module, demonstrating exceptional adaptability across diverse data environments.

## B DETAILS OF FINE-TUNING LLM

The process of LLM instruction fine-tuning is shown in Figure 9

### B.1 INSTRUCTION DATASET SOURCE

We ensure that our instruction dataset is constructed exclusively from authentic financial texts, rather than relying on LLM-generated synthetic news. The collected raw financial documents undergo a systematic preprocessing pipeline (including de-duplication and cleaning, removing html tags, and some symbols) to improve corpus quality and label consistency. We then select a diverse set of financial text types—such as exchange disclosures, corporate announcements, sell-side analyst reports, macroeconomic and sector commentaries, post-market recaps, and breaking news flashes—as the source for instruction fine-tuning. The selected financial texts to construct instruction dataset

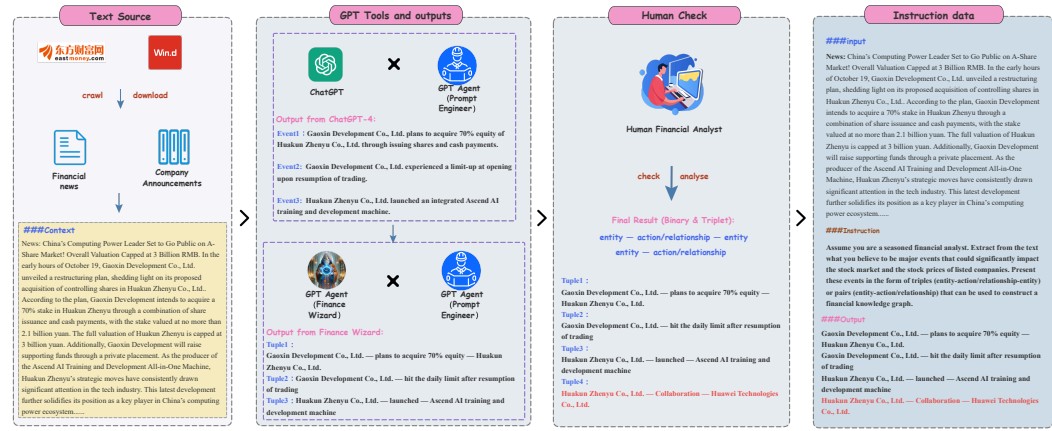

Figure 9: The process of constructing instruction fine-tuning dataset.

comprise approximately 5,000 instances, accounting for about one-tenth of the overall financial text corpus.

### B.2 THE FLOW OF INSTRUCTION DATASET CONSTRUCTION

High-quality instruction datasets are crucial for fine-tuning LLMs. To address the scarcity of open-source instruction datasets in the financial domain, we established a semi-automated data construction pipeline. Drawing upon the Teacher-Student distillation paradigm, we leveraged OpenAI's advanced LLM ecosystem to construct high-quality instruction datasets. This process is orchestrated through three specialized agents, each handling a distinct phase of the extraction workflow:

**Stage 1**: The Prompt Engineer Agent for prompt optimization

The workflow initiates with the Prompt Engineer Agent, which functions as the architect of the extraction logic.

- **Role**: This agent is responsible for the iterative design and refinement of system instructions via prompt engineering.
- **Function**: It constructs precise prompts that condition the downstream agents to act as "seasoned financial analysts", ensuring that the subsequent generation strictly adheres to the requirements of financial event extraction and Knowledge Graph construction.

**Stage 2**: ChatGPT-4 for semantic reasoning & event extraction

Guided by the optimized instructions from the Prompt Engineer, ChatGPT-4 assumes the role of the primary semantic analyzer (the "Teacher").

- **Role**: It adopts the persona of a financial analyst, leveraging its powerful reasoning capabilities and extensive world knowledge.
- **Function**: ChatGPT-4 processes raw financial news and company announcements, then analyze and summarize key financial events (e.g., U.S. Places 54 Chinese entities, including Inspur, on Export Control 'Entity List'; NVIDIA will launch the B100 GPU next month).

**Stage 3**: The Finance Wizard Agent for structured Knowledge constructing

The final extraction phase is executed by the Finance Wizard Agent, which specializes in domain-specific structuring.

- **Role**: Acting as specialized parser and data formatter, this agent processes the analytical summaries generated by ChatGPT-4.
- **Function**: Based on professional financial knowledge, Finance Wizard Agent can process the outputs of ChatGPT-4 and further generate binary and triplet consisting financial entities and relationships for each financial event.

Finally, to guarantee the reliability of the dataset, we introduce human verification mechanism. Seasoned financial experts meticulously review the extracted tuples against the original news reports. They manually correct any factual errors or logical inconsistencies, ensuring the final dataset achieves high standards of accuracy and completeness before being used for fine-tuning. In summary, this process culminates in the construction of an instruction-based fine-tuning dataset, comprising inputs, optimized instructions, and high-quality outputs.

### B.3 ADDITIONAL EXAMPLES OF FINEX INSTRUCTIONS AND RESPONSES

A JSON-formatted instruction–input–output instance used for instruction tuning.

```
[
  {
    "Instruction": "Assume you are a seasoned financial analyst. Extract
        ↪ from the text what you believe to be major events that could
        ↪ significantly impact the stock market and the stock prices of
        ↪ listed companies. Present these events in the form of triples (
        ↪ entity-action/relationship-entity) or pairs (entity-action/
        ↪ relationship) that can be used to construct a financial
        ↪ knowledge graph.",
    "Input": "News: China's Computing Power Leader Set to Go Public on A-
        ↪ Share Market! Overall Valuation Capped at 3 Billion RMB. In the
        ↪  early hours of October 19, Gaoxin Development Co., Ltd.
        ↪ unveiled a restructuring plan, shedding light on its proposed
        ↪ acquisition of controlling shares in Huakun Zhenyu Co., Ltd..
        ↪ According to the plan, Gaoxin Development intends to acquire a
        ↪ 70% stake in Huakun Zhenyu through a combination of share
        ↪ issuance and cash payments, with the stake valued at no more
        ↪ than 2.1 billion yuan. The full valuation of Huakun Zhenyu is
        ↪ capped at 3 billion yuan. Additionally, Gaoxin Development will
        ↪  raise supporting funds through a private placement. As the
        ↪ producer of the Ascend AI Training and Development All-in-One
        ↪ Machine, Huakun Zhenyu's strategic moves have consistently
        ↪ drawn significant attention in the tech industry. This latest
        ↪ development further solidifies its position as a key player in
        ↪ China's computing power ecosystem......",
    "Output": [
      {
        "events": {
          "event1": {
            "total": "Gaoxin Development Co., Ltd. plans to acquire 70%
                ↪ equity in Huakun Zhenyu Co., Ltd.",
            "entity1": "Gaoxin Development Co., Ltd.",
            "relationship": "plans to acquire 70% equity",
            "entity2": "Huakun Zhenyu Co., Ltd."
          },
          "event2": {
            "total": "Gaoxin Development Co., Ltd. hit the one-word limit
                ↪ -up after resumption of trading",
            "entity1": "Gaoxin Development Co., Ltd.",
            "relationship": "hit the one-word limit-up after resumption
                ↪ of trading"
          },
          "event3": {
            "total": "Huakun Zhenyu Co., Ltd. launched the Ascend AI
                ↪ Training and Development All-in-One Machine",
            "entity1": "Huakun Zhenyu Co., Ltd.",
            "relationship": "launched",
            "entity2": "Ascend AI Training and Development All-in-One
                ↪ Machine"
          },
          "event4": {
            "total": "Huakun Zhenyu Co., Ltd. and Huawei Technologies Co
                ↪ ., Ltd. are strategic partners",
```

```
27              "entity1": "Huakun Zhenyu Co., Ltd.",
28              "relationship": "strategic partners",
29              "entity2": "Huawei Technologies Co., Ltd."
30          }
31       }
32     }
33   ]
34  }
35 ]
```

## C  Additional Description of Raw Data and Graph Structure

### C.1  Raw data

Table 3 summarizes several basic attributes of the key stocks in our study, including stock code, industry affiliation, main business, and the number of financial texts. The financial news and company announcements (total of 48476 pieces)are collected or crawled from the EastMoney platform, which consolidates disclosures from the Shanghai and Shenzhen Stock Exchanges, CSRC reporting channels, and multiple major financial media outlets. It is widely regarded as one of China's leading and most authoritative financial data providers.

Table 3: Fundamental information of key stocks.

| Code | Name | Amounts of texts | Main business | Industry |
|------|------|------------------|---------------|----------|
| 000977 | Inspur Electronic Information Industry Co., Ltd | 2658 | Server Manufacture | |
| 603019 | Dawning Information Industry Co., Ltd | 2395 | High-Performance Computing | |
| 002230 | iFLYTEK Co., Ltd | 4082 | Software Development | Artificial |
| 300418 | BEIJING KUNLUN TECH CO., LTD | 2818 | Internet Services | Intelligence |
| 300308 | Zhongji InnoLight Technology Co., Ltd | 2887 | Optical Communication Equipment | |
| 300474 | Jingjia Micro Co., Ltd | 2874 | Semiconductors | |
| 300750 | Contemporary Amperex Technology Co., Ltd | 8430 | Power Batteries | |
| 002594 | BYD Co., Ltd | 8631 | New Energy Vehicles | |
| 601012 | LONGi Green Energy Technology Co., Ltd | 4478 | Photovoltaic Products | Renewable |
| 300014 | EVE Energy Co., Ltd | 3248 | Lithium Batteries | Energy |
| 600438 | Tongwei Co., Ltd | 3584 | Photovoltaic Manufacturing | |
| 300274 | Sungrow Power Supply Co., Ltd | 2655 | Photovoltaic Inverters | |

### C.2  Graph Structure and statistics description

As shown in Table 4, this table summarizes all node and edge types used in our graph construction, along with their names, semantic meanings, and feature descriptions. The node types include key stocks, financial entities, stock exchanges, and margin-related actors, while the edge types capture diverse relationships such as price correlation, capital flow, and financial events. The associated features are derived from both structured financial indicators and semantic embeddings generated by pretrained language models, incorporating information like time series, net cash flows, and event triples.

To better describe the constructed FEKGs, we report basic statistics on their scale, structure, and connectivity. We compute the time span of the dataset, the number and window size of temporal graphs, the numbers of extracted events and documents, and the counts and density of nodes and edges. These statistics give a clear picture of the data on FEKGs. Detailed statistics are summarized in Table 5.

## D  Theoretical Discussion and Analysis

Table 4: The construction definition of multimodal heterogeneous graphs.

| Nodes & Edges | Names | Notation | Descriptions and Features | Feature Notation |
|---|---|---|---|---|
| Node types | Key Stocks | $V_t^{KS}$ | **Desc.:** leading companies in the subfields of green computing and new energy industries
**Feat.:** time series matrix of OHLCV within an w-days window | $X_{vt}^{KS}$ |
| | Other Entities | $V_t^{OE}$ | **Desc.:** other financial entities in the FinKG
**Feat.:** global semantic meaning of the input generated by FinBERT | $X_{vt}^{OE}$ |
| | HKEX | $V_t^{HK}$ | **Desc.:** Hong Kong Stock Exchange
**Feat.:** time series of status of SH/SZ-HK Stock Connect within an w-days window | $X_{vt}^{HK}$ |
| | Margin Financing | $V_t^F$ | **Desc.:** Margin Financing
**Feat.:** one-hot encoding | $X_{vt}^F$ |
| | Securities Lending | $V_t^L$ | **Desc.:** Securities Lending
**Feat.:** one-hot encoding | $X_{vt}^L$ |
| Edge types | Key Stocks—correlation—Key Stocks (undirected) | $E_t^{Corr}$ | **Desc.:** correlation of prices among Key Stocks.
**Feat.:** time series of Spearman correlation coefficients within an w-days window. | $X_{et}^{Corr}$ |
| | HKEX—Invest—Stocks (directed) | $V_t^{HK}$ | **Desc.:** Hong Kong Stock Exchange investing
**Feat.:** Northbound capital (funds flowing via SH/SZ-HK Stock Connect) | $X_{et}^{HI}$ |
| | Margin Financing—go long—Stocks (directed) | $E_t^{Long}$ | **Desc.:** net inflows from margin financing
**Feat.:** time series of net margin financing cash flows within an w-days window. | $X_{et}^{Long}$ |
| | Securities Lending—go short—Stocks (directed) | $E_t^{Short}$ | **Desc.:** net short-selling of securities
**Feat.:** time series of daly securities sold short within an w-days window | $X_{et}^{Short}$ |
| | Key Stocks—relationship/action—Other Entities (directed) | $E_t^{SRE}$ | **Desc.:** triples from FEKG
**Feat.:** global semantic meaning of the input generated by FinBERT | $X_{et}^{SRE}$ |
| | Other Entities—relationship/action—Key Stocks (directed) | $E_t^{ERS}$ | **Desc.:** triples from FEKG
**Feat.:** global semantic meaning of the input generated by FinBERT | $X_{et}^{ERS}$ |
| | Key Stocks—relationship/action—Key Stocks Key Stocks—short event (emergency event) (directed) | $E_t^{KS}$ | **Desc.:** tuples from FEKG (triples and event pairs )
**Feat.:** global semantic meaning of the input generated by FinBERT | $x_{et}^{KS}$ |
| | Other Entities—relationship/actiont—Other Entities Other Entities—short event (emergency event) (directed) | $E_t^{OE}$ | **Desc.:** tuples from FEKG (triples and event pairs )
**Feat.:** global semantic meaning of the input generated by FinBERT | $x_{et}^{OE}$ |

Table 5: Statistical description of FEKGs.

| Statistic Item | Description | Value |
|---|---|---|
| Time span ($t_{span}$) | Total data collection period | 2021–2025 |
| The number of FEKGs ($t$) | Total number of temporal graphs in the series | 1317 graphs |
| Time window of each graph ($t_{win}$) | Duration covered by one graph instance | 10 days |
| Total events of one graph ($|\mathcal{E}_{event}|$) | Number of events processed by FinEX | 1804 |
| Source documents of one graph ($N_{docs}$) | Number of documents processed by FinEX for one graph | 568 |
| Total entities (nodes) of one graph ($|\mathcal{V}|$) | Total unique entities (nodes) within one graph instance | 363 |
| Total relations (edges) of one graph ($|\mathcal{E}_{FEKG}|$) | Total edges derived from all sources in one graph instance | 1804 |
| Event Edge Ratio ($r_{event}$) | Percentage of edges derived from FinEX/Event analysis | 87.95% |
| Average Degree ($Avg(Deg)$) | Total connectivity measure ($2|\mathcal{E}|/|\mathcal{V}|$) | 4.9 |
| Total nods of FEKGs | The amount of nodes of a series of FEKGs | 485,026 |
| Total edges of FEKGs | The amount of edges of a series of FEKGs | 2,5873,694 |

In this section, we provide a comprehensive theoretical analysis of the proposed MEHGT-LKG framework. We justify the model's effectiveness from four perspectives: the Information Bottleneck principle in knowledge extraction, the structural expressivity of edge-enhanced attention, the information flow in multimodal fusion, and the computational complexity.

## D.1 INFORMATION BOTTLENECK PERSPECTIVE ON FINEX

We justify the necessity of the LLM-driven FinEX module through the lens of the Information Bottleneck (IB) principle. Let $\mathcal{D}$ be the raw unstructured financial text (e.g., news, announcements), which contains both predictive signals $S$ (e.g., "mergers and acquisitions", "product launches") and other noise $N$ (e.g., redundant phrasing, irrelevant background). Direct modeling of $\mathcal{D}$ often leads to overfitting on $N$ due to the high dimensionality of linguistic variance.

The objective of our FinEX agent is to learn a mapping $\mathcal{M} : \mathcal{D} \to G_{KG}$ (the Knowledge Graph) that maximizes the relevant information regarding the stock trend $Y$ while compressing the input representation. This is equivalent to optimizing the IB objective:

$$\max_{\theta_{LLM}} \quad I(G_{KG}; Y) - \beta I(G_{KG}; \mathcal{D}) \tag{13}$$

where $I(\cdot; \cdot)$ denotes Mutual Information. By strictly extracting structured tuples (Entities, Relations, Events) and discarding non-causal narrative text, FinEX explicitly minimizes the complexity term $I(G_{KG}; \mathcal{D})$. This process effectively acts as a **semantic denoiser**, ensuring that the downstream MEHGT model receives inputs with high signal-to-noise ratio, thereby reducing the aleatoric uncertainty in stock trend prediction.

## D.2 EDGE-ENHANCED ATTENTION MECHANISM

Standard Heterogeneous Graph Transformers (HGT) compute attention scores purely based on node feature similarity ($q_t k_s^T$). This formulation implies *relational isotropy*: the existence of an edge guarantees a valid information channel, regardless of the edge's semantic quality. This assumption fails in financial graphs where edges extracted from news may be noisy or indicate the "severance" of ties.

**Mathematical Formulation of MEHGT.** As implemented in our `message` function, MEHGT introduces a multiplicative modulation mechanism. To articulate this, we first define the unnormalized attention score $e_{ts}$ between target node $t$ and source node $s$:

$$e_{ts} = \frac{(q_t k_s^T) \cdot \Psi(\mathbf{e}_{ts})}{\sqrt{d}} \tag{14}$$

where $\mathbf{e}_{ts} \in \mathbb{R}^{d_{edge}}$ is the edge feature vector, and $\Psi(\cdot)$ denotes the scalar projection function (implemented as summation `sum(dim=-1)` in our code). The final attention coefficient $\alpha_{ts}$ is obtained via normalization:

$$\alpha_{ts} = \frac{\exp(e_{ts})}{\sum_{s' \in \mathcal{N}(t)} \exp(e_{ts'})} \tag{15}$$

**Proposition 1 (Differentiable Topology Refinement).** *The multiplicative interaction between node similarity and edge scalar $\Psi(\mathbf{e}_{ts})$ transforms the attention mechanism into a differentiable gate, enabling the model to dynamically refine the graph topology during training.*

*Proof Analysis.* Consider the gradient flow with respect to the source node features $h_s$. Let $\gamma_{ts} = \Psi(\mathbf{e}_{ts})$ be the learnable scalar gate derived from the edge.

- **Additive vs. Multiplicative:** Standard methods often use additive bias ($q_t k_s^T + \gamma_{ts}$). In an additive scheme, a negative edge feature only shifts the mean of the logits but cannot strictly "shut down" a channel if the node similarity $q_t k_s^T$ is sufficiently large.

- **Gating Mechanism:** In MEHGT, $\gamma_{ts}$ acts as a gain controller. If FinEX extracts a noisy or irrelevant event, the edge encoder can learn to output a large negative scalar for $\Psi(\mathbf{e}_{ts})$ (or $\Psi(\mathbf{e}_{ts}) \to 0$ depending on activation). In the case of $\gamma_{ts} \to -\infty$, the term $\exp(e_{ts}) \to 0$, effectively removing the edge $(s, t)$ from the computational graph during aggregation.

Thus, MEHGT does not merely aggregate information over a fixed graph; it learns a *soft adjacency matrix* $\tilde{A}$ where $\tilde{A}_{ts}$ is modulated by the semantic validity of the edge, effectively performing structural denoising.

### D.3 Multimodal Alignment and Orthogonal Fusion

Financial graphs are inherently multimodal, containing edges driven by numerical correlations ($E_{corr}$) and textual events ($E_{text}$). A core challenge is the *distribution shift*: numerical embeddings and semantic embeddings often lie on disjoint manifolds.

**Subspace Alignment via Edge-Type Projection.** In our forward propagation, we explicitly apply a type-specific linear transformation ('edge_type_transforms') before aggregation. Let $m_{ts}^{(r)}$ be the message from relation type $r$. The aligned message $\tilde{m}_{ts}^{(r)}$ is:

$$\tilde{m}_{ts}^{(r)} = m_{ts}^{(r)} W_{proj}^{(r)} + b^{(r)} \tag{16}$$

Geometrically, $W_{proj}^{(r)}$ rotates and scales the semantic subspace of relation $r$ to align with the target node's feature space. This ensures that when we perform the summation $\sum_r \tilde{m}_{ts}^{(r)}$, we are adding vectors that share a common semantic basis, preventing destructive interference between modalities.

**Proposition 2 (Variance Reduction in Joint Estimation).** *The fusion of aligned heterogeneous messages acts as a joint estimator that reduces the aleatoric uncertainty of the node representation.*

*Analysis.* The final node update in MEHGT is governed by a residual connection of fused messages:

$$h_t' = \text{Skip}(h_t, \sum_{r \in \mathcal{R}} \sum_{s \in \mathcal{N}_r(t)} \tilde{m}_{ts}^{(r)}) \tag{17}$$

From a statistical learning perspective, let the true signal for stock $t$ be $S_t$. Each relation type $r$ provides a noisy observation $O_r = S_t + \epsilon_r$.

- **Orthogonality of Noise:** The noise in price correlations ($\epsilon_{price}$) typically stems from market microstructure (high-frequency fluctuation), while noise in text ($\epsilon_{text}$) stems from linguistic ambiguity. These noise sources are statistically independent (orthogonal).

- **Fusion Gain:** By summing over $R$ aligned views, the variance of the aggregated noise term scales by $\frac{1}{R^2} \sum \text{Var}(\epsilon_r)$, which is strictly lower than the variance of any single modality.

Therefore, the architectural choice of "Projection followed by Summation" is not arbitrary; it is the theoretical requirement for maximizing the Mutual Information $I(Y; H_{graph})$ under multimodal noise conditions.

### D.4 Computational Complexity Analysis

Despite the enhanced expressivity and multimodal fusion capabilities, MEHGT maintains efficient scalability suitable for financial applications.

Let $|V|$ be the number of nodes and $|E|$ be the number of edges in the heterogeneous graph $G_t$. Let $d$ denote the hidden dimension and $h$ be the number of attention heads. The computational cost is dominated by the linear projections and the attention computation.

- **Linear Projections:** The cost for computing Query, Key, Value, and Message projections for all nodes is $O(|V| \cdot d^2)$.

- **Edge-Enhanced Attention:** The cost for computing Eq. (5) and aggregating messages for all edges is $O(|E| \cdot h \cdot d)$.

The additional cost introduced by the edge feature transformation $f(X_{edge})$ is $O(|E| \cdot d)$, which is linear with respect to the number of edges. Since financial graphs are typically sparse (i.e., $|E| \ll |V|^2$), the overall complexity of MEHGT is $O(|V| \cdot d^2 + |E| \cdot d)$. This is comparable to standard GAT models and significantly more efficient than fully connected Transformer architectures ($O(|V|^2)$), ensuring that MEHGT-LKG can be trained effectively on large-scale dynamic market data.

## E Implement Details

All experiments were conducted on Linux (CentOS) operating systems with Python 3.11. The FinEX model was fine-tuned on NVIDIA A100 GPUs with a multi-GPU distributed training framework (NVLink is better). And we use python-based pipeline to construct FEKGs and a series of dynamic multimodal heterogeneous graphs, and use Neo4j for better storage, querying and visualization. The MEHGT-LKG model and baselines were trained on NVIDIA RTX 4090 GPUs, and all graph neural networks are implemented with the PyTorch Geometric (PyG) library. The environment included sufficient system memory to support large-scale training of heterogeneous and multimodal architectures.

FinEX was trained for up to 500 epochs, with learning rate 5e-05. And other model including MEHGT and baselines was trained for up to 5000 epochs, with early stopping on the validation loss using a patience of 20 epochs to mitigate overfitting. The learning rate was initialized to 0.0001. In MEHGT, the temporal scales were set to 5, 10, and 15, and the Edge-enhanced heterogeneous transformer consists three layers. All models were trained 10 times with different random seeds, and we report the average performance.

The details of hardware and settings are shown in Table 6.

Table 6: Detailed Implementation Specifications and Reproducibility Settings.

| Configuration | FinEX Agent | MEHGT & Baselines |
|---|---|---|
| GPU hardware | 8*NVIDIA A100 (40G) | 2*NVIDIA RTX 4090 (24G) |
| CPU environment | Intel Xeon Platinum (128 Cores) | Intel Xeon Platinum (128 Cores) |
| System memory | 512 GB RAM | 128 GB RAM |
| Batch size | 4 | 256 |
| Training time | ∼168 hours (Total) | ∼3 hours (Total) |
| Peak VRAM usage | 39 GB | 14 GB |
| CUDA version | 12.2 | 12.3 |
| Python version | 3.10 | 3.11 |
| Accelerator | Deepspeed Zero-2 | / |
| Precision | FP16 | FP16 |
| Learning rate | 5e-05 | 1e-04 |
| Packages | torch, transformers, torch_geometric, safetensors, deepspeed, scikit_learn | |

# F SUPPLEMENTARY EXPERIMENT

## F.1 COMPUTATIONAL EFFICIENCY COMPARISON

To provide a comprehensive performance evaluation, we record the training and inference time, GPU memory, and recommended hardware for all models, as detailed in Table 7. It is observed that Time-series models, benefiting from their lightweight architectures, consume minimal memory and consequently achieve faster training and inference speeds. Graph-based models generally demand higher memory overhead due to their complex structures and larger parameters. Regarding our proposed MEHGT-LKG, although the integration of the sophisticated Edge-enhanced Mechanism for multimodal data results in relatively higher memory usage, its training time remains comparable to other graph-based baselines. Crucially, during the inference phase, the model's latency is minimal, showing a negligible difference compared to time-series baselines and graph-based lines. Given its superior predictive performance, we argue that MEHGT-LKG maintains a significant competitive advantage despite the moderate computational cost.

In particular, we also present the computational costs and hardware requirements for FinEX to facilitate future reproduction and accessibility. While the LLM fine-tuning indeed demands significant computational resources, the deployment and inference stages are highly efficient. By leveraging the Flash-attention framework and 8-bit quantization, FinEX achieves an impressive inference speed of 125 tokens/s on an A100 GPU. Consequently, with such minimal latency. Thus, FinEX can rapidly extract financial knowledge and integrate with MEHGT efficiently through multimodal heterogeneous graphs, enhancing the predictive performance and result interpretability

Table 7: Comparison of computational efficiency during training and inference.

| Category | Methods | Training Time (s/epoch) | Inference Time (s/sample) | GPU Memory (GB) | Recommend GPU |
|---|---|---|---|---|---|
| **Time-series models** | Informer | 0.53 | 0.0008 | 0.89 | 1*RTX 4090 |
| | TCN | 0.32 | 0.0007 | 0.45 | 1*RTX 4090 |
| | CNN-LSTM | 0.34 | 0.0009 | 0.64 | 1*RTX 4090 |
| | Time-MoE | / | 0.0018 | 1.28 | 1*RTX 4090 |
| | KRONOS | 0.95 | 0.0014 | 7.85 | 1*RTX 4090 |
| **Graph-based models** | GAT | 1.28 | 0.0015 | 5.86 | 1*RTX 4090 |
| | HGT | 1.46 | 0.0014 | 8.42 | 1*RTX 4090 |
| | MAC | 3.25 | 0.0018 | 10.98 | 1*RTX 4090 |
| | MDGNN | 3.76 | 0.0018 | 12.56 | 1*RTX 4090 |
| | COGRASP | 0.55 | 0.0011 | 8.63 | 1*RTX 4090 |
| | ENHANCER | 1.09 | 0.0012 | 10.87 | 1*RTX 4090 |
| **Ours** | FinEX | 135.47 | 50 tokens/s | 39.58 | 8*A100-40G/80G in fine-tuning 1*A100-40G/80G in inferencing |
| | MEHGT-LKG | 1.395 | 0.0012 | 11.57 | 1*RTX 4090 |

## F.2 Main Comparison Results on other datasets

The performance comparison between our model and the baselines on other datasets is summarized below. The conclusions drawn from these results are consistent with those presented in the main text. The details are shown in Table 8 and Table 9.

## F.3 Additional Results of Market Trading Simulation

The return curves on additional datasets are shown in the figure below. The analysis based on these curves leads to conclusions consistent with those in the main text. The details are shown in Figure 10 and Table 10

## F.4 Performance under different conditions

Figure 11 depicts the simulated investment performance of MEHGT-LKG over the backtesting period. We annotate key time intervals and major events to further analyze how MEHGT-LKG behaves under different market conditions. Overall, MEHGT-LKG exhibits a more adaptive response to different conditions, especially when market conditions change due to major news shocks and structural shifts in volatility.

First, in late May 2023, positive news about NVIDIA GPUs drove market attention and capital into the AI sector. Compared with the Buy-and-Hold and ENHANCER strategies, MEHGT-LKG captured this upward momentum more effectively, and cumulative capital curve steeper rise. By tracing back the multimodal heterogeneous graphs constructed for the corresponding time window, we observe that this news event was extracted by FinEX as an event triple and written into the graph. This shows that FinEX can sensitively detect and extract key financial events. And on basis of multimodal heterogeneous graphs, MEHGT-LKG injects the corresponding event-semantic edge features into heterogeneous attention computation and message passing. This strengthens the model's predictive signals regarding on the stock, thereby allowing it to capture this round of investment opportunity more promptly.

Second, when negative shocks hit the market (e.g., news about potential U.S. restrictions on exporting AI chips to China), the baseline strategies experience noticeably sharper drawdowns. In contrast, MEHGT-LKG exhibits a milder decline and recovers most of the losses within a shorter period. By tracing back the multimodal heterogeneous graphs constructed for the corresponding time window, we observe that this news also has been extracted by FinEX as an event triple and injected into the graph. This indicates that the model's edge-enhanced attention mechanism can also detect downside risk triggered by major events and partially hedge against drawdowns through timely position adjustment.

Finally, during the technical rebound and bottom-fishing phase in the market, MEHGT-LKG identifies bottoming signals earlier and increases investment accordingly. Its equity curve turns upward faster than those of the baselines and delivers more persistent excess returns. This suggests that the

Table 8: Prediction performance of different methods across selected stock datasets (1).

| Methods | Inspur (000977) | | | | | | CATL (300750) | | | | | |
|---|---|---|---|---|---|---|---|---|---|---|---|---|
| | ACC | MCC | Precision | Recall | F1 | AUC | ACC | MCC | Precision | Recall | F1 | AUC |
| *Time-series models* | | | | | | | | | | | | |
| Informer | 0.6104 | 0.2195 | 0.6154 | 0.5333 | 0.5714 | 0.5894 | 0.6104 | 0.2372 | 0.5357 | 0.6818 | 0.6000 | 0.5808 |
| TCN | 0.5844 | 0.1669 | 0.5846 | 0.5067 | 0.5429 | 0.5515 | 0.5909 | 0.1800 | 0.5200 | 0.5909 | 0.5532 | 0.5913 |
| CNN-LSTM | 0.6234 | 0.2462 | 0.6349 | 0.5333 | 0.5797 | 0.6245 | 0.5455 | 0.1913 | 0.4828 | **0.8485** | 0.6154 | 0.5248 |
| Time-MoE | 0.6164 | 0.2298 | 0.6094 | 0.5571 | 0.5821 | 0.5932 | 0.6234 | 0.2104 | 0.5833 | 0.4242 | 0.4912 | 0.5559 |
| KRONOS | 0.6233 | 0.2449 | 0.6087 | 0.6000 | 0.6043 | 0.6235 | 0.6233 | 0.2400 | 0.5588 | 0.6032 | 0.5802 | 0.6196 |
| *Graph-based models* | | | | | | | | | | | | |
| GAT | 0.6169 | 0.2865 | 0.5690 | **0.8800** | 0.6911 | 0.6019 | 0.6234 | 0.2439 | 0.5541 | 0.6212 | 0.5857 | 0.5754 |
| HGT | 0.6169 | 0.2630 | 0.5755 | 0.8133 | 0.6740 | 0.5516 | 0.6364 | 0.2737 | 0.5658 | 0.6515 | 0.6056 | 0.6030 |
| MAC | 0.6039 | 0.2171 | 0.5761 | 0.7067 | 0.6347 | 0.5954 | 0.6234 | 0.2104 | 0.5833 | 0.4242 | 0.4912 | 0.5559 |
| MDGNN | 0.6299 | 0.2605 | 0.6500 | 0.5200 | 0.5778 | 0.6226 | **0.6558** | **0.2960** | **0.6000** | 0.5909 | 0.5954 | 0.6288 |
| COGRASP | 0.5974 | 0.3111 | 0.6257 | 0.7467 | 0.6437 | 0.6243 | 0.6234 | 0.2604 | 0.5833 | 0.6242 | 0.5912 | 0.6259 |
| ENHANCER | **0.6364** | 0.2956 | 0.5941 | 0.7060 | 0.6818 | 0.6294 | 0.6301 | 0.2461 | 0.5714 | 0.5714 | 0.5714 | 0.6213 |
| **MEHGT-LKG (ours)** | 0.6234 | **0.3718** | **0.6883** | 0.7713 | **0.7184** | **0.6322** | 0.6039 | 0.2748 | 0.5243 | 0.8182 | **0.6391** | **0.6307** |
| | IFLYTEK (002230) | | | | | | EVE (300014) | | | | | |
| *Time-series models* | | | | | | | | | | | | |
| Informer | 0.5779 | 0.1962 | 0.5288 | **0.7746** | 0.6286 | 0.5280 | 0.6429 | 0.2517 | 0.4769 | 0.5962 | 0.5299 | 0.5850 |
| TCN | 0.5974 | 0.1883 | 0.5652 | 0.5493 | 0.5571 | 0.5305 | 0.5909 | 0.1922 | 0.4286 | 0.6346 | 0.5116 | 0.5786 |
| CNN-LSTM | 0.6234 | 0.2438 | 0.5890 | 0.6056 | 0.5972 | 0.5843 | 0.5260 | 0.2302 | 0.4054 | 0.8654 | 0.5521 | 0.5754 |
| Time-MoE | 0.6164 | 0.2407 | 0.5584 | 0.6615 | 0.6056 | 0.6321 | 0.5616 | 0.2048 | 0.4396 | 0.7547 | 0.5556 | 0.6182 |
| KRONOS | 0.6507 | 0.2854 | 0.6250 | 0.5385 | 0.5785 | 0.6104 | 0.5685 | 0.2343 | 0.4468 | 0.7925 | 0.5714 | 0.6129 |
| *Graph-based models* | | | | | | | | | | | | |
| GAT | 0.6234 | 0.2456 | 0.5867 | 0.6197 | 0.6027 | 0.6129 | 0.6039 | 0.2567 | 0.4471 | 0.7308 | 0.5547 | **0.6655** |
| HGT | 0.6104 | 0.2544 | 0.5556 | **0.7746** | 0.6471 | 0.6359 | 0.6494 | 0.2858 | 0.4857 | 0.6538 | 0.5574 | 0.6158 |
| MAC | 0.6234 | 0.2382 | 0.6000 | 0.5493 | 0.5735 | 0.5959 | 0.4545 | 0.2350 | 0.3806 | **0.9808** | 0.5484 | 0.5430 |
| MDGNN | 0.6299 | 0.2562 | **0.6842** | 0.5662 | 0.5771 | 0.5418 | 0.6169 | 0.2383 | 0.4533 | 0.6538 | 0.5354 | 0.5911 |
| COGRASP | 0.6234 | 0.2956 | 0.6267 | 0.6397 | 0.6127 | 0.6129 | 0.6245 | 0.2635 | 0.5204 | 0.7520 | 0.5431 | 0.6303 |
| ENHANCER | 0.6507 | 0.3038 | 0.6250 | 0.6553 | 0.6543 | 0.6104 | 0.6678 | 0.2771 | 0.5225 | 0.6570 | **0.6325** | 0.6312 |
| **MEHGT-LKG (ours)** | **0.6818** | **0.3681** | 0.6375 | 0.7183 | **0.6755** | **0.6930** | **0.6883** | **0.3166** | 0.5357 | 0.5769 | 0.5556 | 0.6640 |
| | Zhongji Innolight (300308) | | | | | | Sungrow (300274) | | | | | |
| *Time-series models* | | | | | | | | | | | | |
| Informer | 0.6558 | 0.3155 | 0.6857 | 0.6076 | 0.6443 | 0.6297 | 0.6234 | 0.2489 | 0.5584 | 0.6418 | 0.5972 | 0.5891 |
| TCN | 0.6169 | 0.2369 | 0.6000 | **0.7595** | 0.6704 | 0.5690 | 0.5974 | 0.1810 | 0.5373 | 0.5373 | 0.5373 | 0.5963 |
| CNN-LSTM | 0.5779 | 0.1538 | 0.5795 | 0.6456 | 0.6108 | 0.4928 | 0.6234 | 0.2874 | 0.5474 | 0.7761 | **0.6420** | 0.6056 |
| Time-MoE | 0.6027 | 0.2062 | 0.5949 | 0.6438 | 0.6184 | 0.6356 | 0.6027 | 0.2161 | 0.5385 | 0.6256 | 0.6017 | 0.6109 |
| KRONOS | 0.6233 | 0.2477 | 0.6364 | 0.5753 | 0.6043 | 0.6132 | 0.6301 | 0.2544 | 0.5735 | 0.6094 | 0.5909 | 0.6425 |
| *Graph-based models* | | | | | | | | | | | | |
| GAT | 0.6104 | 0.2217 | 0.5979 | 0.7342 | 0.6591 | 0.6175 | 0.6494 | 0.3085 | 0.5802 | 0.7015 | 0.6351 | 0.6447 |
| HGT | 0.6169 | 0.3076 | 0.5758 | 0.9620 | **0.7204** | 0.6046 | 0.6558 | 0.2910 | 0.6207 | 0.5373 | 0.5760 | 0.5656 |
| MAC | 0.5974 | 0.2342 | 0.7073 | 0.3671 | 0.4833 | 0.5332 | 0.6169 | 0.2144 | 0.5645 | 0.5224 | 0.5426 | 0.5805 |
| MDGNN | 0.6364 | 0.2992 | **0.7255** | 0.4684 | 0.5692 | 0.6117 | 0.6039 | 0.2665 | 0.5294 | **0.8060** | 0.6391 | 0.5656 |
| COGRASP | 0.6009 | 0.2787 | 0.6918 | 0.7124 | 0.6347 | 0.6254 | 0.5844 | 0.2919 | 0.5961 | 0.6289 | 0.6352 | 0.6126 |
| ENHANCER | 0.6429 | 0.2855 | 0.6538 | 0.6456 | 0.6497 | 0.6370 | 0.6310 | 0.2644 | 0.5735 | 0.6094 | 0.5909 | 0.6254 |
| **MEHGT-LKG (ours)** | **0.6688** | **0.3375** | 0.6591 | 0.7324 | 0.6946 | **0.6485** | **0.6753** | **0.3337** | 0.6393 | 0.5821 | 0.6094 | **0.6523** |

model not only reacts relatively quickly to sudden shocks, but also leverages correlation edges and capital-flow signals to detect changes in market conditions, thereby achieving smoother and more robust performance over longer horizons.

Overall, MEHGT-LKG is able to adapt its trading behavior across different market conditions (rising, falling, and regime-shift periods). Building on its strong predictive performance, it strikes a better balance between return and drawdown, and delivers comparatively stable excess returns.

### F.5 ADDITIONAL RESULTS OF ABLATION STUDY

Here, we additionally report the prediction performance of remain datstes in Table 11, and the return performance of the ablation experiments. The details are shown in Figure 12.

### F.6 SPEARMAN ANALYSIS

A Spearman correlation heatmap is constructed based on the closing prices of selected key stocks, as illustrated in Figure 13. The intensity of the color reflects the strength of the correlation: stronger positive correlations are depicted in red, whereas stronger negative correlations are shown in blue.

The graph shows that within sectors such as artificial intelligence and new energy, stocks often exhibit strong internal correlations. For example, the Spearman coefficient between Kunlun Tech

Table 9: Prediction performance of different methods across selected stock datasets (2)

| Methods | Sugon (603019) | | | | | | BYD (002594) | | | | | |
|---|---|---|---|---|---|---|---|---|---|---|---|---|
| | ACC | MCC | Precision | Recall | F1 | AUC | ACC | MCC | Precision | Recall | F1 | AUC |
| *Time-series models* | | | | | | | | | | | | |
| Informer | 0.5714 | 0.1791 | 0.6667 | 0.3291 | 0.4407 | 0.5750 | 0.6623 | 0.3079 | **0.6000** | 0.6000 | 0.6000 | 0.6088 |
| TCN | 0.5649 | 0.1659 | 0.6579 | 0.3165 | 0.4274 | 0.5445 | 0.5584 | 0.1545 | 0.4839 | 0.6923 | 0.5696 | 0.5167 |
| CNN-LSTM | 0.5779 | 0.1883 | 0.6667 | 0.3544 | 0.4628 | 0.5789 | 0.6234 | 0.2498 | 0.5455 | 0.6462 | 0.5915 | 0.5749 |
| Time-MoE | 0.6233 | 0.2471 | 0.6349 | 0.5556 | 0.5926 | 0.6368 | 0.6364 | 0.2207 | 0.5738 | 0.5385 | 0.5556 | 0.62621 |
| Kronos | 0.6370 | 0.2761 | 0.6203 | 0.6806 | 0.6490 | 0.6509 | 0.6039 | 0.2458 | 0.5217 | 0.7385 | 0.6115 | 0.5854 |
| *Graph-based models* | | | | | | | | | | | | |
| GAT | 0.6169 | 0.2559 | 0.5862 | 0.8608 | 0.6974 | 0.6116 | 0.6429 | 0.3098 | 0.5581 | 0.7385 | 0.6358 | 0.6659 |
| HGT | 0.6234 | 0.2540 | 0.6615 | 0.5443 | 0.5972 | 0.6571 | 0.6558 | 0.3176 | 0.5769 | 0.6923 | 0.6294 | 0.6352 |
| MAC | 0.6234 | 0.2601 | **0.6780** | 0.5063 | 0.5797 | 0.6322 | 0.6364 | 0.2348 | 0.5918 | 0.4462 | 0.5088 | 0.6408 |
| MDGNN | 0.6169 | 0.2324 | 0.6190 | 0.6582 | 0.6380 | 0.5970 | 0.6104 | 0.3268 | 0.5225 | **0.8923** | 0.6381 | 0.6193 |
| COGRASP | 0.6104 | 0.2236 | 0.6338 | 0.5696 | 0.6000 | 0.6270 | 0.5366 | 0.3174 | 0.5666 | 0.6769 | 0.6286 | 0.6366 |
| ENHANCER | 0.6299 | 0.2590 | 0.6375 | 0.6456 | 0.6415 | 0.6603 | 0.6039 | 0.3006 | 0.4921 | 0.6232 | 0.6292 | 0.6570 |
| MEHGT-LKG (ours) | **0.6429** | 0.3162 | 0.6034 | **0.8861** | **0.7179** | **0.6693** | **0.6688** | 0.3483 | 0.5875 | 0.7231 | **0.6483** | **0.6746** |

| | Kunlun Tech (300418) | | | | | | LONGi (601012) | | | | | |
|---|---|---|---|---|---|---|---|---|---|---|---|---|
| *Time-series models* | | | | | | | | | | | | |
| Informer | 0.6104 | 0.2102 | 0.5672 | 0.5507 | 0.5588 | 0.5719 | 0.6688 | 0.2292 | 0.5385 | 0.3889 | 0.4516 | 0.6128 |
| TCN | 0.5909 | 0.1640 | **0.6875** | 0.1594 | 0.2588 | 0.4223 | 0.6429 | 0.1535 | 0.4857 | 0.3148 | 0.3820 | 0.5828 |
| CNN-LSTM | 0.6104 | 0.2005 | 0.5818 | 0.4638 | 0.5161 | 0.5782 | 0.4156 | 0.1629 | 0.3732 | **0.9815** | 0.5408 | 0.4723 |
| Time-MoE | 0.5411 | 0.2149 | 0.5369 | 0.6333 | 0.5702 | 0.6129 | 0.6234 | 0.2387 | 0.4722 | 0.6296 | 0.5397 | 0.6343 |
| KRONOS | 0.6233 | 0.2463 | 0.5692 | 0.5781 | 0.6029 | 0.6029 | 0.6164 | 0.2163 | 0.4848 | 0.5926 | 0.5333 | 0.6190 |
| *Graph-based models* | | | | | | | | | | | | |
| GAT | 0.6104 | 0.2628 | 0.5455 | **0.7826** | 0.6429 | **0.6477** | 0.5714 | 0.2548 | 0.4400 | 0.8148 | 0.5714 | 0.5517 |
| HGT | 0.6494 | 0.2848 | 0.6230 | 0.5507 | 0.5846 | 0.6462 | 0.5649 | 0.2802 | 0.4393 | 0.8704 | 0.5639 | 0.6289 |
| MAC | 0.6169 | 0.2311 | 0.5676 | 0.6087 | 0.5874 | 0.5877 | 0.6234 | 0.2387 | 0.4722 | 0.6296 | 0.5397 | 0.6343 |
| MDGNN | **0.6623** | 0.3264 | 0.6104 | 0.6812 | **0.6438** | 0.6368 | 0.6883 | 0.3042 | **0.5600** | 0.5185 | 0.5385 | 0.6298 |
| COGRASP | 0.6169 | 0.2676 | 0.5532 | 0.7636 | 0.6380 | 0.5886 | 0.6234 | 0.2687 | 0.4722 | 0.6296 | 0.5697 | 0.6343 |
| ENHANCER | 0.6299 | 0.2589 | 0.5262 | 0.5556 | 0.5128 | 0.5394 | 0.6104 | 0.2193 | 0.4595 | 0.6296 | 0.5312 | 0.6106 |
| MEHGT-LKG (ours) | 0.6234 | 0.2604 | 0.5647 | 0.6957 | 0.6234 | 0.6302 | **0.6883** | 0.3275 | 0.5517 | 0.5926 | **0.5714** | **0.7135** |

| | Jingjia Micro (300474) | | | | | | Tongwei (600438) | | | | | |
|---|---|---|---|---|---|---|---|---|---|---|---|---|
| *Time-series models* | | | | | | | | | | | | |
| Informer | 0.6039 | 0.2187 | 0.5714 | 0.7027 | 0.6303 | 0.5889 | 0.6234 | 0.1943 | 0.5400 | 0.4355 | 0.4821 | 0.5852 |
| TCN | 0.5260 | 0.1842 | 0.5034 | **0.9865** | **0.6667** | 0.5446 | 0.5519 | 0.2249 | 0.4690 | 0.8548 | **0.6057** | 0.5521 |
| CNN-LSTM | 0.6234 | 0.2456 | 0.6081 | 0.6081 | 0.6081 | 0.5703 | 0.5260 | 0.2140 | 0.4553 | 0.8032 | 0.6054 | 0.5603 |
| Time-MoE | 0.6027 | 0.2002 | 0.5873 | 0.5362 | 0.5606 | 0.5711 | 0.5890 | 0.2000 | 0.5062 | 0.6321 | 0.5775 | 0.6016 |
| KRONOS | 0.6233 | 0.2462 | 0.5972 | 0.6232 | 0.6099 | 0.6426 | 0.6027 | 0.2178 | 0.5195 | 0.6557 | 0.5297 | 0.6120 |
| *Graph-based models* | | | | | | | | | | | | |
| GAT | 0.6234 | 0.2465 | 0.6053 | 0.6216 | 0.6133 | 0.6392 | 0.6039 | 0.2171 | 0.5063 | 0.6452 | 0.5674 | 0.6048 |
| HGT | 0.6234 | 0.2551 | 0.5909 | 0.7027 | 0.6420 | 0.6033 | 0.6234 | 0.2255 | 0.5303 | 0.5645 | 0.5469 | 0.5640 |
| MAC | 0.6234 | 0.2491 | 0.6600 | 0.4459 | 0.5323 | 0.5867 | 0.5390 | 0.2137 | 0.4615 | **0.8710** | 0.6034 | 0.5903 |
| MDGNN | 0.6299 | 0.2716 | 0.5934 | 0.7297 | 0.6545 | 0.5912 | **0.6429** | 0.2269 | **0.5814** | 0.4032 | 0.4762 | 0.5403 |
| COGRASP | 0.6234 | 0.2475 | 0.6026 | 0.6184 | 0.6164 | 0.5867 | 0.6027 | 0.2478 | 0.5495 | 0.6557 | 0.5797 | 0.6027 |
| ENHANCER | 0.6425 | 0.2812 | 0.6216 | 0.6632 | 0.6240 | 0.5867 | 0.6299 | 0.2518 | 0.5325 | 6613 | 0.5899 | 0.6008 |
| MEHGT-LKG (ours) | **0.6623** | 0.3226 | **0.6618** | 0.6018 | 0.6338 | **0.6459** | 0.6429 | 0.2518 | 0.5593 | 0.5323 | 0.5455 | **0.6197** |

and Zhongji Innolight is 0.89, and between CATL and EVE is 0.87. These high values indicate shared industry factors or capital flows, leading to synchronized price movements. In contrast, cross-sector pairs, especially between AI and new energy, tend to show negative correlations. For instance, Kunlun Tech and Tongwei have a coefficient of –0.68, reflecting divergent trends and a clear sector rotation effect, where capital shifts between the two in a see-saw pattern.

Interestingly, Inspur shows the highest cumulative Spearman correlation within its sector, totaling 3.79. Our model correspondingly achieves the best prediction performance for Inspur, with an MCC of 0.3718. This indicates that stronger correlations enhance the value of information from other stocks. During training, the model captures intrinsic price formation patterns via message passing along the Key Stocks–correlation–Key Stocks edges, leading to the best MCC for Inspur in stock trend prediction.

## F.7 INTERPRETABILITY

To investigate how the multi-head attention mechanism allocates importance across different edge types in the heterogeneous graph, we compute the average attention weights for each edge type based on the test set. Specifically, we extract the attention coefficients from each layer of MEHGTConv,

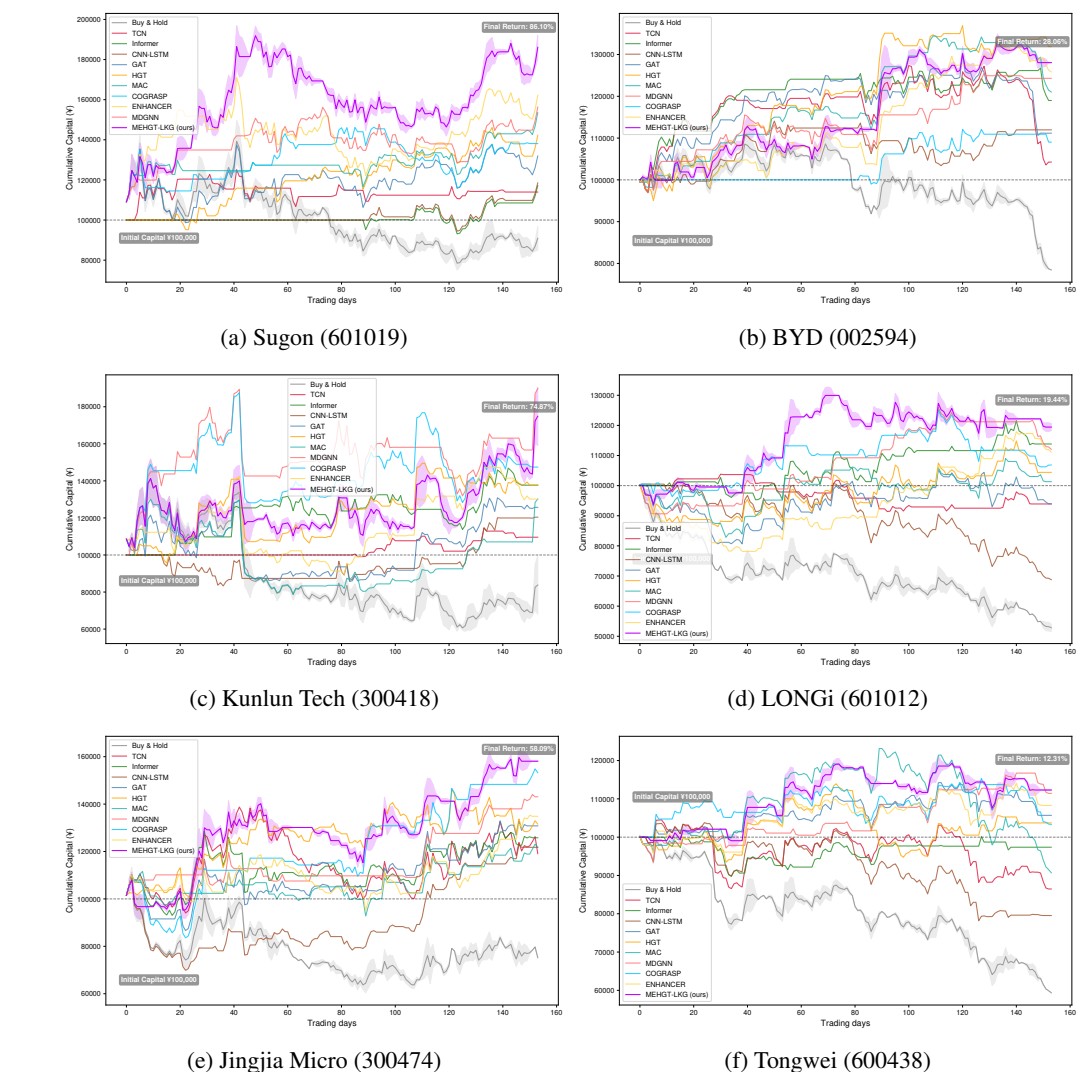

(a) Sugon (601019)

(b) BYD (002594)

(c) Kunlun Tech (300418)

(d) LONGi (601012)

(e) Jingjia Micro (300474)

(f) Tongwei (600438)

Figure 10: Simulated trading performance of all models during backtesting.

aggregate them across all heads for each edge, and then compute the average attention coefficients for each edge type over all test datasets. The final results are visualized in Figure 14.

From the graph, the proposed model shows distinct attention distributions across edge types. The edge type Key Stocks—correlation—Key Stocks receives the highest attention (0.6716), highlighting the role of stock co-movements in message aggregation. Valuable information is effectively transmitted through these structured correlations, enhancing trend prediction. For the four edge types enriched with LLM-driven semantics, MEHGT-LKG also assigns high attention, indicating that FinEX-extracted financial events help relay relevant information to target stock nodes. These semantic edge types complement MEHGT-LKG's edge feature processing, jointly improving information aggregation and prediction accuracy.

# G    RELATED WORK

## G.1    NLP-DRIVEN METHODS FOR FINANCIAL MODELING.

Knowledge-driven methods have been widely applied in finance, leveraging huge structured and unstructured knowledge to improve stock prediction, investment decision making, risk management, and financial analysis. Traditional knowledge-driven methods focus mainly on the extraction of features from financial texts. For example, Schumaker & Chen (2009) explored how breaking

Table 10: Profitability performance of different methods.

| Stocks I | Methods | Profitability performance | | | Stocks II | Methods | Profitability performance | | |
|---|---|---|---|---|---|---|---|---|---|
| | | CRR | MDD | Sharpe | | | CRR | MDD | Sharp |
| Inspur 000977 | Informer | 39.1375 | 0.2496 | 1.4859 | CATL 300750 | Informer | -0.3439 | 0.1881 | 0.1048 |
| | TCN | 21.2760 | 0.2696 | 0.7683 | | TCN | -46.1522 | 0.5023 | -1.3684 |
| | CNN-LSTM | 28.0206 | 0.2079 | 1.2795 | | CNN-LSTM | -50.3170 | 0.5062 | -1.5319 |
| | Time-MoE | 27.7812 | 0.2962 | 1.0681 | | Time-MoE | 3.3834 | 0.1049 | 0.435 |
| | KRONOS | 48.1207 | 0.2002 | 1.6502 | | KRONOS | 14.7469 | 0.0932 | 0.7051 |
| | GAT | 48.7005 | 0.2623 | 1.2811 | | GAT | 3.2661 | 0.1636 | 0.3350 |
| | HGT | 46.0885 | 0.2546 | 1.4366 | | HGT | 15.5525 | 0.1136 | 1.1398 |
| | MAC | 28.2165 | 0.2973 | 1.0461 | | MAC | -4.4937 | 0.1630 | -0.3461 |
| | MDGNN | 58.2471 | 0.1530 | 2.0843 | | MDGNN | 20.0766 | 0.0945 | 1.4430 |
| | COGRASP | 36.7374 | 0.3299 | 1.0845 | | COGRASP | 9.0183 | 0.1794 | 0.6954 |
| | ENHANCER | 53.8005 | 0.3069 | 1.344 | | ENHANCER | 11.1375 | 0.1896 | 0.8205 |
| | MEHGT-LKG (ours) | 104.5015 | 0.1989 | 2.4844 | | MEHGT-LKG (ours) | 20.8393 | 0.0838 | 1.7462 |
| Sugon 603019 | Informer | 17.1574 | 0.1254 | 1.3779 | BYD 002594 | Informer | 18.9981 | 0.0612 | 1.8414 |
| | TCN | 13.9591 | 0.1467 | 0.8461 | | TCN | 4.2585 | 0.1868 | 0.4635 |
| | CNN-LSTM | 18.5299 | 0.1254 | 1.4795 | | CNN-LSTM | 11.9645 | 0.0937 | 1.2065 |
| | Time-MoE | 20.9031 | 0.2009 | 1.584 | | Time-MoE | 8.7657 | 0.0883 | 0.9695 |
| | KRONOS | 34.6748 | 0.2884 | 1.4372 | | KRONOS | 14.1406 | 0.1105 | 1.0897 |
| | GAT | 31.9407 | 0.2702 | 0.8620 | | GAT | 11.1698 | 0.1508 | 0.9874 |
| | HGT | 42.3104 | 0.0978 | 2.0081 | | HGT | 31.8891 | 0.0532 | 2.4177 |
| | MAC | 53.5646 | 0.1014 | 2.0462 | | MAC | 21.0417 | 0.0996 | 1.8541 |
| | MDGNN | 56.3198 | 0.1776 | 1.8907 | | MDGNN | 11.5724 | 0.0820 | 0.8191 |
| | COGRASP | 38.1234 | 0.2063 | 1.1945 | | COGRASP | 9.0316 | 0.0432 | 1.1191 |
| | ENHANCER | 62.2121 | 0.2863 | 1.6322 | | ENHANCER | 25.8289 | 0.0683 | 1.9545 |
| | MEHGT-LKG (ours) | 86.1013 | 0.2372 | 2.1732 | | MEHGT-LKG (ours) | 28.0607 | 0.0694 | 2.1129 |
| IFLYTEK 002230 | Informer | 6.1512 | 0.3228 | 0.4445 | LONGi 601012 | Informer | 2.4555 | 0.0894 | 0.3191 |
| | TCN | 1.5595 | 0.1628 | 0.2556 | | TCN | -6.0843 | 0.1195 | -0.7286 |
| | CNN-LSTM | 35.9297 | 0.1690 | 1.5652 | | CNN-LSTM | -31.2370 | 0.3173 | -1.6541 |
| | Time-MoE | 23.4226 | 0.1832 | 1.143 | | Time-MoE | 0.1079 | 0.1713 | -0.021 |
| | KRONOS | 45.5197 | 0.1635 | 1.8705 | | KRONOS | 6.8410 | 0.1416 | 0.3105 |
| | GAT | 14.2939 | 0.2822 | 0.6382 | | GAT | -6.0351 | 0.1935 | -0.1572 |
| | HGT | 26.8404 | 0.2601 | 1.0763 | | HGT | 4.4006 | 0.1391 | 0.3635 |
| | MAC | 12.2738 | 0.2326 | 0.6969 | | MAC | 1.3198 | 0.1757 | 0.1905 |
| | MDGNN | 43.2621 | 0.1283 | 1.7303 | | MDGNN | 12.0417 | 0.1105 | 0.9541 |
| | COGRASP | 34.8971 | 0.1374 | 1.8975 | | COGRASP | 6.8576 | 0.1512 | 0.519 |
| | ENHANCER | 48.3968 | 0.2098 | 1.434 | | ENHANCER | 11.1476 | 0.2182 | 0.7545 |
| | MEHGT-LKG (ours) | 74.0035 | 0.1742 | 2.4209 | | MEHGT-LKG (ours) | 19.4352 | 0.0967 | 1.2398 |
| Kunlun Tech 300418 | Informer | 37.6844 | 0.1421 | 1.5573 | EVE 300014 | Informer | 6.0840 | 0.1264 | 0.4937 |
| | TCN | 9.5881 | 0.0736 | 1.1430 | | TCN | -11.1429 | 0.1412 | -0.6905 |
| | CNN-LSTM | 20.4161 | 0.1673 | 0.9239 | | CNN-LSTM | -12.9285 | 0.1926 | -0.5445 |
| | Time-MoE | 20.4387 | 0.3071 | 0.7905 | | Time-MoE | -11.9021 | 0.2810 | -1.1207 |
| | KRONOS | 25.2375 | 0.2891 | 0.9332 | | KRONOS | 0.2922 | 0.1855 | -0.3393 |
| | GAT | 25.7172 | 0.3392 | 0.6937 | | GAT | 3.1514 | 0.1957 | 0.3191 |
| | HGT | 37.6609 | 0.2443 | 1.0303 | | HGT | 7.9989 | 0.1375 | 0.7413 |
| | MAC | 30.2156 | 0.3931 | 0.9842 | | MAC | -20.3043 | 0.2921 | -0.8207 |
| | MDGNN | 89.8132 | 0.3036 | 1.9303 | | MDGNN | 3.3344 | 0.2460 | 0.3651 |
| | COGRASP | 47.4476 | 0.3166 | 1,035 | | COGRASP | 9.3478 | 0.0551 | 1.1725 |
| | ENHANCER | 30.1954 | 0.3717 | 0.7710 | | ENHANCER | 19.1132 | 0.1106 | 1.4175 |
| | MEHGT-LKG (ours) | 74.8732 | 0.2419 | 1.4239 | | MEHGT-LKG (ours) | 25.2459 | 0.1416 | 1.4747 |
| Zhongji Innolight 300308 | Informer | 84.3662 | 0.1529 | 2.4685 | Tongwei 600438 | Informer | -2.6245 | 0.1024 | -0.2461 |
| | TCN | 65.6934 | 0.3317 | 1.7605 | | TCN | -13.5322 | 0.1728 | -0.7652 |
| | CNN-LSTM | 4.4516 | 0.3619 | 0.3921 | | CNN-LSTM | -20.4624 | 0.2470 | -1.2588 |
| | Time-MoE | 100.7028 | 0.2621 | 2.529 | | Time-MoE | -9.2933 | 0.1674 | -1.5725 |
| | KRONOS | 144.3722 | 0.1864 | 2.7765 | | KRONOS | 9.5537 | 0.1172 | 0.3405 |
| | GAT | 124.2828 | 0.3976 | 2.3478 | | GAT | 5.6952 | 0.0766 | 0.5508 |
| | HGT | 168.6253 | 0.3139 | 2.3542 | | HGT | 3.7162 | 0.1684 | 0.3969 |
| | MAC | 92.7296 | 0.1053 | 3.0320 | | MAC | -9.3012 | 0.2635 | -0.5096 |
| | MDGNN | 232.7633 | 0.1238 | 4.2321 | | MDGNN | 11.4228 | 0.0669 | 1.1255 |
| | COGRASP | 58.8023 | 0.1427 | 1.8465 | | COGRASP | 3.3198 | 0.1044 | 0.3300 |
| | ENHANCER | 191.8602 | 0.3310 | 2.361 | | ENHANCER | 8.2722 | 0.0823 | 0.6675 |
| | MEHGT-LKG (ours) | 274.4909 | 0.2735 | 3.8083 | | MEHGT-LKG (ours) | 12.3115 | 0.0978 | 1.1192 |
| Jingjia Micro 300474 | Informer | 25.7934 | 0.2017 | 1.0557 | Sungrow 300274 | Informer | 6.3613 | 0.2230 | 0.4588 |
| | TCN | 19.0871 | 0.2782 | 0.7763 | | TCN | 12.7821 | 0.1289 | 0.5032 |
| | CNN-LSTM | 21.7649 | 0.3655 | 0.9461 | | CNN-LSTM | 33.4434 | 0.1288 | 1.2017 |
| | Time-MoE | 15.2226 | 0.1828 | 0.8595 | | Time-MoE | 5.1654 | 0.1547 | 0.4335 |
| | KRONOS | 19.0426 | 0.1685 | 1.1811 | | KRONOS | 17.5274 | 0.2076 | 1.2285 |
| | GAT | 30.8164 | 0.2158 | 1.1493 | | GAT | 42.7223 | 0.1440 | 1.6319 |
| | HGT | 31.5752 | 0.1256 | 1.2969 | | HGT | 31.9033 | 0.1553 | 1.1684 |
| | MAC | 21.8109 | 0.1685 | 1.1811 | | MAC | 5.2465 | 0.1997 | 0.0889 |
| | MDGNN | 40.5920 | 0.1640 | 1.6081 | | MDGNN | 28.9026 | 0.1380 | 1.2700 |
| | COGRASP | 27.7979 | 0.1503 | 1.0455 | | COGRASP | 25.8189 | 0.0686 | 1.1883 |
| | ENHANCER | 34.8507 | 0.1998 | 1.3185 | | ENHANCER | 32.0946 | 0.1493 | 1.0832 |
| | MEHGT-LKG (ours) | 58.0853 | 0.1776 | 1.9637 | | MEHGT-LKG (ours) | 51.1442 | 0.1016 | 1.7827 |

news influences stock prices, employing various textual news representation techniques. Khadjeh Nassirtoussi et al. (2015) extracted sentiment information from breaking financial news, and further utilized them to predict stock marker accurately. Nam & Seong (2019) focused on identifying events from financial texts, and proposed a novel stock prediction model. Zhou et al. (2024) explored the impact of news sources on stock prices, and further designed a deep learning prediction model based on this novel feature.

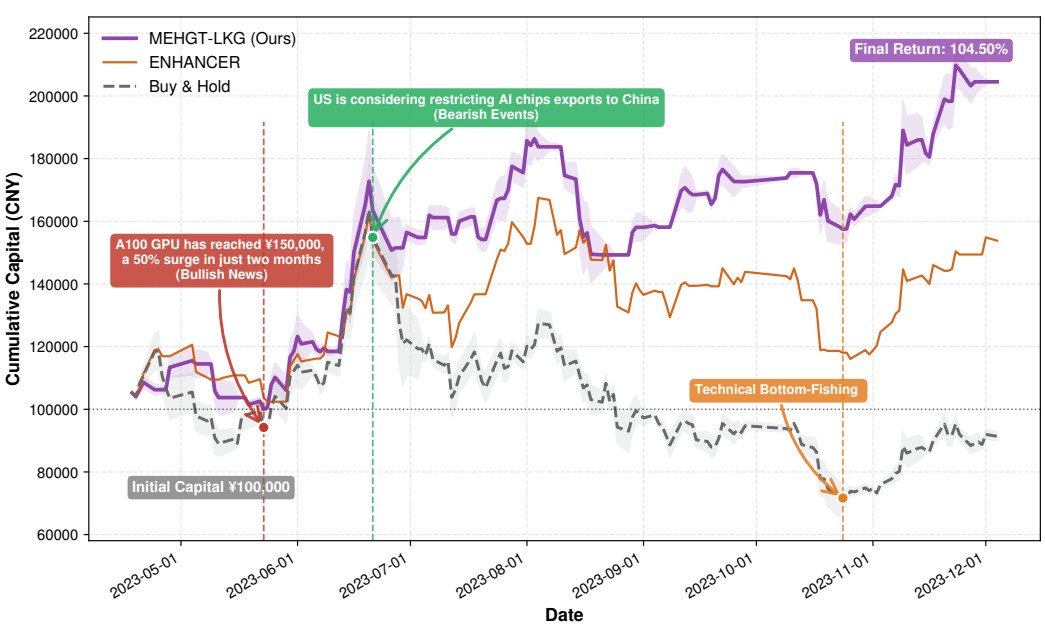

Figure 11: Investment analysis in different conditions.

Table 11: Comparison results of ablation analysis across other stock datasets.

| Methods | Sugon (603019) | | | | | | BYD(002594) | | | | | |
|---|---|---|---|---|---|---|---|---|---|---|---|---|
| | ACC | MCC | Precision | Recall | F1 | AUC | ACC | MCC | Precision | Recall | F1 | AUC |
| w/o events | **0.6429** | 0.2908 | **0.6765** | 0.5823 | 0.6259 | 0.6073 | 0.6558 | 0.3059 | 0.5833 | 0.6462 | 0.6131 | 0.6321 |
| w/o edge feats | 0.6299 | 0.2754 | 0.6250 | 0.7595 | 0.6857 | 0.6384 | 0.6429 | 0.3051 | 0.5476 | 0.7077 | 0.6174 | 0.6450 |
| w/o indicators | 0.6234 | 0.2468 | 0.6364 | 0.6203 | 0.6282 | 0.6285 | 0.6169 | 0.2974 | 0.5300 | **0.8154** | 0.6424 | 0.5898 |
| MEHGT-LKG | **0.6429** | **0.3162** | 0.6034 | **0.8861** | **0.7179** | **0.6693** | **0.6688** | **0.3483** | 0.5875 | 0.7231 | **0.6483** | **0.6746** |
| | Kunlun Tech (300418) | | | | | | LONGi(601012) | | | | | |
| w/o events | **0.6494** | **0.2845** | **0.6829** | 0.4058 | 0.5091 | 0.6049 | 0.6299 | 0.2563 | 0.4795 | **0.6481** | 0.5512 | 0.6128 |
| w/o edge feats | 0.6104 | 0.2356 | 0.5429 | 0.6129 | 0.5758 | 0.6085 | 0.6623 | 0.2912 | 0.5172 | 0.5556 | 0.5357 | 0.6650 |
| w/o indicators | 0.6169 | 0.2393 | 0.5625 | 0.6522 | 0.6040 | 0.5915 | 0.6753 | 0.2637 | 0.5435 | 0.4630 | 0.5000 | 0.6294 |
| MEHGT-LKG | 0.6234 | 0.2604 | 0.5647 | **0.6957** | **0.6234** | **0.6302** | **0.6883** | **0.3275** | 0.5517 | 0.5926 | **0.5714** | **0.7135** |
| | Jingjia Micro (300474) | | | | | | Tongwei (600438) | | | | | |
| w/o events | 0.6234 | 0.2486 | 0.6000 | **0.6486** | 0.6234 | 0.5596 | **0.6558** | 0.2610 | **0.6098** | 0.4032 | 0.4854 | 0.5826 |
| w/o edge feats | 0.6494 | 0.2945 | 0.6267 | 0.5802 | 0.6026 | 0.6388 | 0.6299 | 0.2245 | 0.5254 | 0.5000 | 0.5124 | 0.6050 |
| w/o indicators | 0.6364 | 0.2710 | 0.6250 | 0.6081 | 0.6164 | 0.6030 | **0.6558** | **0.2610** | 0.5957 | 0.4516 | 0.5138 | **0.6177** |
| MEHGT-LKG | **0.6623** | **0.3226** | **0.6618** | 0.6018 | **0.6338** | **0.6459** | 0.6429 | 0.2518 | 0.5593 | **0.5323** | **0.5455** | 0.6197 |

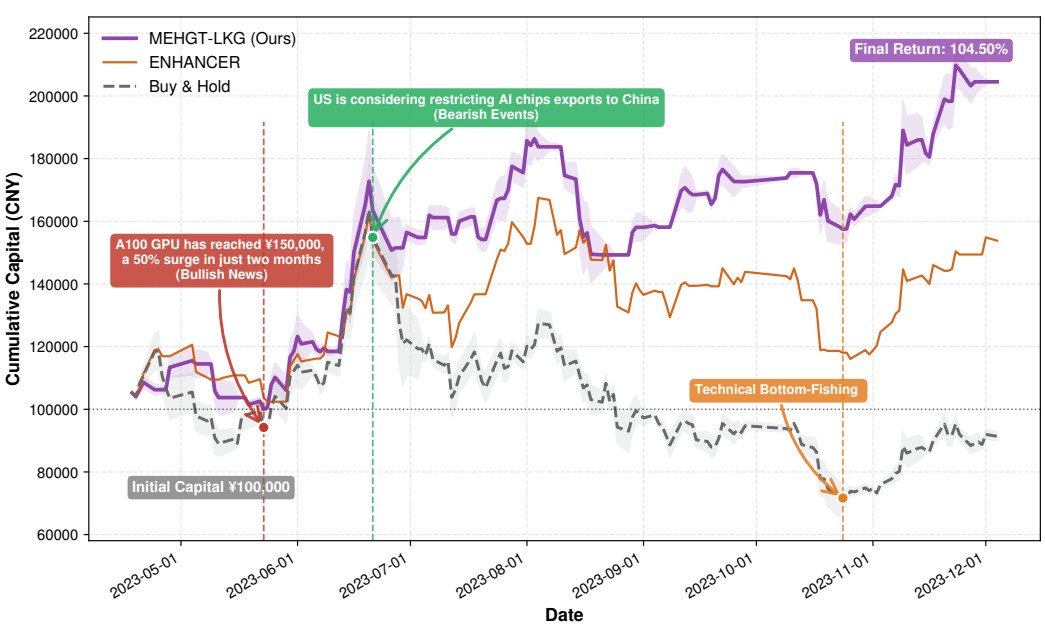

(a) CRR      (b) MDD      (c) Sharp

Figure 12: Profitability performance of ablation analysis across all stock datasets.

As knowledge graph technology evolves, it has become increasingly effective at systematizing both structured and unstructured knowledge, and more scholars are using knowledge graphs to solve var-

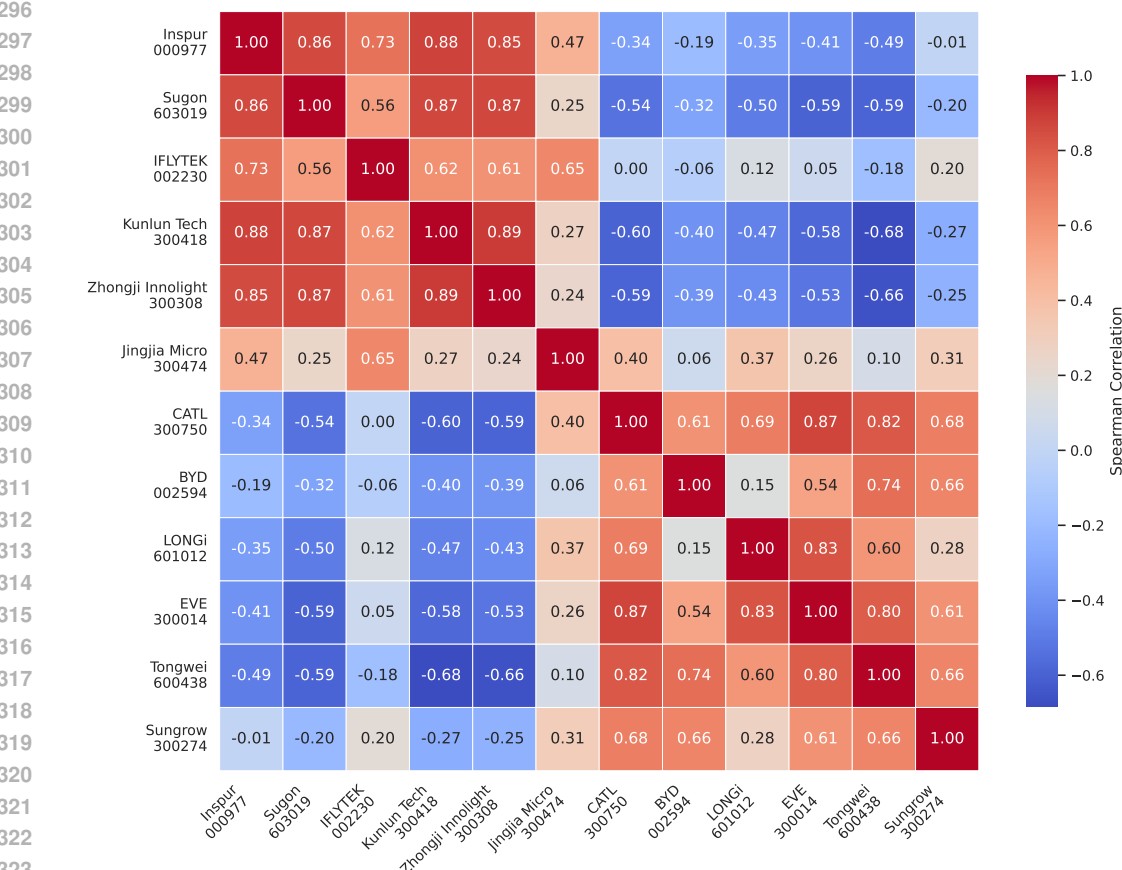

Figure 13: Spearman correlation heatmap of selected stocks.

ious financial tasks. For instance, Wang et al. (2022a) constructed a knowledge graph for online lending fraud detection through address disambiguation and mining implicit relations. Cheng et al. (2022) extracted structured event triples from financial texts to build financial knowledge graphs, and proposed a multimodal GNN to forecast stock trends. Haque & Tozal (2023) introduced a knowledge graph-based method to model the relationships between medical codes, offering a graph-driven solution for fraud detection in health insurance. Wang et al. (2023) constructed a stock knowledge graph based on the fundamental information of listed companies to represent semantic and relational information among stocks, and utilized GCNs to enhance stock trend prediction performance. Song et al. (2024) formulated a knowledge graph from enterprise relational data, and proposed a multi-structure cascaded graph neural network framework (MS-CGNN) for enterprise credit risk assessment. Cai & Xie (2024) designed a two-layer knowledge graph with a semantic layer modeling financial subordination and a syntactic layer capturing articulation relations, supporting accurate and interpretable fraud detection.

Notably, in recent years, large language models (LLMs) renowned for their exceptional semantic understanding and knowledge-reasoning capabilities, have been employed in some domains for knowledge graph construction. In particular, Cheng et al. (2024) constructed a Chinese financial event knowledge graph by extracting relational triples using an LLM-based module with iterative verification, and developed a GAT-based model to improve graph completeness. Li & Sanna Passino (2024) extracted dynamic entity relations from financial texts through fine-tuning LLM, and constructed a dynamic financial knowledge graph, enabling effective predict stocks trend prediction via an attention-based GNN. Yan et al. (2025) proposed KnowNet, a system that constructs knowledge graphs by extracting entity-relation triples from LLM outputs and aligning them with validated external KGs to enhance accuracy in health information retrieval. Wang et al. (2025) introduced

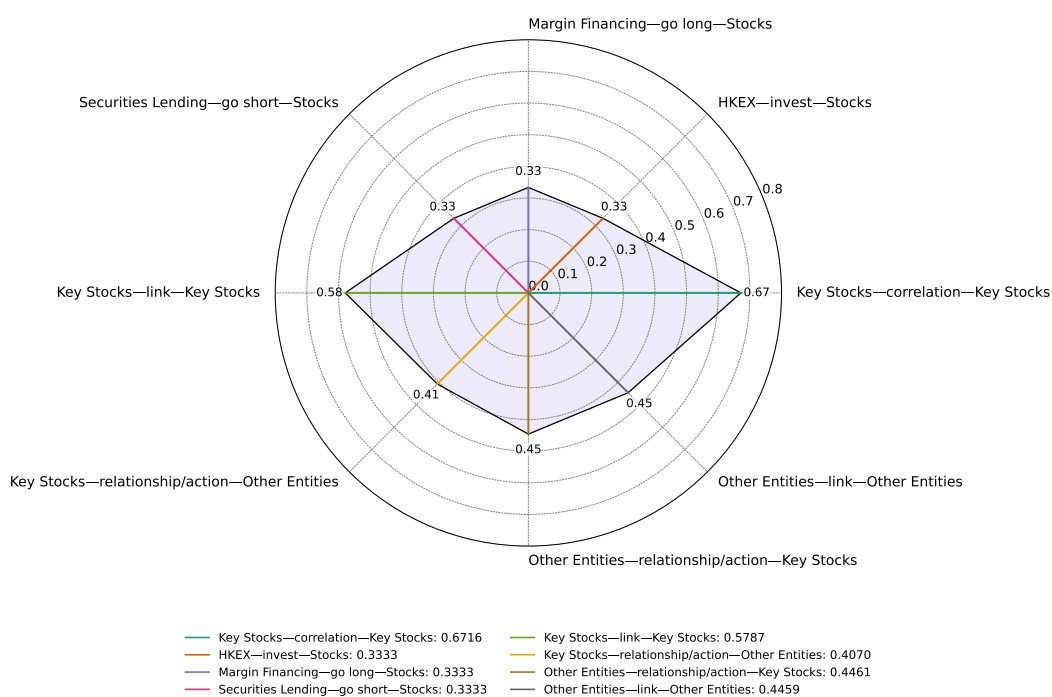

Figure 14: Average attention weights for different edge types.

a novel LLM-based model to for efficient named entity recognition, enhancing knowledge graph construction across diverse domains.

In summary, knowledge-driven methods have been widely applied in financial domain and have demonstrated strong performance across various tasks. In recent years, some researchers attempted to introduce LLM into the construction of financial knowledge graphs. However, existing approaches mainly rely on prompt engineering or API-based queries, and still suffer from hallucinations, output randomness, and poor structural consistency in generating structured knowledge representations. These limitations lead to low efficiency, limited accuracy, and weak controllability in the process of knowledge graph construction. Hence, to address these issues, this study builds a high-quality, well-formatted instruction dataset based on multi-source financial texts such as news and research reports, with guidance from domain experts. A fine-tuned LLM-based module, named Event and Relationship Extraction Agent, is proposed to extract key events and corresponding structured knowledge (including triples and pairs) from long-form financial texts. This method mitigates hallucination and uncertainty during generation, significantly improving the efficiency and reliability of structured information extraction. It lays a solid foundation for high-quality financial knowledge graph construction and the subsequent modeling of knowledge-infused multimodal heterogeneous graphs.

### G.2 STOCK PREDICTION WITH GRAPH LEARNING.

Essentially, the price volatility of stock is caused by its own market signal and the interference of related enterprises. Compared with traditional time series methods, graph learning can effectively integrate features from neighboring nodes across diverse relation types. This enriches the modeling process with structural complexity and better captures the dependencies among stocks, making it increasingly popular in stock prediction tasks. Chen et al. (2018) proposed a GCN-based stock prediction model incorporating related corporations' information, effectively capturing inter-company relations to enhance forecasting accuracy. Cheng & Li (2021). measured spillover sensitivity with respect to firm attributes and volume signals, and further designed attribute-driven graph attention network (AD-GAT) to capture dynamic influence strengths for more accurate trend forecasting. Hsu et al. (2021) explored stock and sector interactions from time series data, and subsequently proposed FinGAT to capture latent relations and improve profitable stock recommendation without

predefined graphs. Chen et al. (2021) introduced a GC–CNN framework that integrates an improved GCN and a Dual-CNN structure to jointly model stock-level and market-level information, demonstrating superior performance in stock trend prediction compared to existing methods. Cheng et al. (2022) proposed MAGNN, a multi-modality graph neural network for financial time series prediction. By modeling heterogeneous sources with a two-phase attention mechanism, the model captures both intra- and inter-modality dependencies, offering interpretable and accurate forecasting results. Wang et al. (2022b) designed HATR-I, a hierarchical temporal-relational model combining dilated convolutions and dual attention over multiplex graphs to capture fine-grained stock dynamics and inter-stock relations, achieving superior performance on real-world datasets. Ma et al. (2023a) considered the dynamic nature of stock relationships and integrated motif-based similarity into graph modeling, further proposing a dynamic graph LSTM to improve trend prediction and trading profitability. To address the limitations of traditional sentiment-based methods, Ma et al. (2023b) developed a Multi-source Aggregated Classification (MAC) model, which combines numerical features, sentiment embeddings, and related stock information via GCNs, significantly improving stock price movement prediction. Zhang et al. (2025) fused sentiment, transaction, and textual data via tensor-based early fusion, and further proposed a novel dynamic attribute-driven graph attention network incorporating sentiment (AGATS), thereby enhancing stock prediction and trading performance.

Most existing methods rely on homogeneous graph learning, treating all nodes as the same type and ignoring semantic differences between entities. However, in financial markets, different types of entities (e.g., listed companies, industries, market themes, and financial events) influence stock prices in different ways. In fact, heterogeneous graph neural networks (HGNNs), owing to their capacity to model multiple types of nodes and relations, have been widely applied in various domains such as medical diagnosis and intelligent transportation. Hence, an increasing number of studies have introduced HGNNs into financial scenarios to explicitly capture the complex structural relationships among diverse types of entities and edges. This approach enables a more comprehensive representation of complex interactions among financial entities, improving the modeling and prediction of market behavior. For example, Tan et al. (2022) transformed relational market factors into heterogeneous node variables and further proposed a novel conditional heterogeneous graph neural network (FinHGNN) to predict stock price. Ma et al. (2024) fused multi-relation spillover effects and jointly modeled return and volatility, and then proposed a heterogeneous graph attention multi-task model (HGA-MT) to enhance risk-aware stock ranking for portfolio optimization. By leveraging semantic signals from earnings calls, Liu et al. (2024b) designed ECHO-GL, a novel heterogeneous graph-based model that enhances stock movement prediction with dynamic relation modeling and post earnings announcement drift processes. Liu et al. (2024a) captured cross-temporal interventions and heterogeneous stock interactions, and further proposed a dual-dynamic attention model (HD2AT) for enhancing portfolio-level financial forecasting. Qian et al. (2024) extracted multi-source stock relations and temporal dynamics, and designed the Multi-relational Dynamic Graph Neural Network (MDGNN) to jointly model evolving inter-stock dependencies for accurate stock movement prediction.

These graph learning-based models have achieved promising results for stock prediction and investment. And, increasing studies have adopted HGNNs to model complex relationships among different types of entities in financial markets, aiming to enhance market behavior understanding and modeling. However, most existing HGNN-based methods fail to effectively process edge features such as Financial event relationships during training, leading to insufficient utilization of multi-source heterogeneous financial information. To address this limitation, we propose MEHGT model, a Multimodal Edge-enhanced Heterogeneous Graph Transformer. The proposed model embeds edge attributes directly into the computation of attention scores within the multi-head attention mechanism, allowing relation information to influence both message passing and representation learning. This further facilitates comprehensive understanding of multimodal heterogeneous graph representation and stock trend forecasting precisely.

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
