# OpenReview forum: "MEHGT-LKG: Multimodal Edge-enhanced Heterogeneous Graph Transformer with LLM-driven Knowledge Graph for Stock Trend Prediction"
_ICLR.cc/2026/Conference — Submitted to ICLR 2026_

### Official Review · Reviewer_vvBP · 2025-10-24

**Soundness:** 2
**Presentation:** 2
**Contribution:** 3
**Rating:** 4
**Confidence:** 4

**Summary:**

This paper presents a study focused on stock trend prediction by constructing and leveraging a heterogeneous knowledge graph centered on financial events. Its main contributions include fine-tuning a LLM for financial event extraction from diverse text sources and proposing a novel method, MEHGT, which explicitly incorporates edge features into both the attention mechanism and message passing process within the graph structure. The work outlines a comprehensive pipeline from information extraction and graph construction to representation learning.

**Strengths:**

1. The paper focuses on the commonly overlooked feature information of edges in heterogeneous knowledge graphs and utilizes it to address the reliance on financial events and capital flow information in stock trend prediction.

2. The paper fine-tuned an LLM specifically designed for extracting financial events from multiple textual sources, which appears to contribute to data processing in the financial field.

3. The paper proposes the MEHGT method, which fully leverages edge information in heterogeneous graphs, providing significant supplementary information for predicting stock market trends.

4. The workflow proposed in the paper encompasses a complete process of information extraction, graph construction, and representation prediction, offering practical reference value for real-world applications.

**Weaknesses:**

Though the paper demonstrates a complete process for the stock movement prediction task, the following weaknesses should be concerned for further improvements.

1. The paper constructs a financial event-centric knowledge graph based on FinEX, but lacks more detailed descriptions such as statistical information about this knowledge graph. Additionally, if the events extracted by FinEX can be tuples of dual entities or single entities, how are they uniformly stored in the graph database?

2. The paper claims to have constructed a multimodal heterogeneous graph, but the multimodality is only reflected in the multiple sources of information, rather than in the different modalities of node and edge representations in the graph. Moreover, the MEHGT method does not include specialized processing or encoding for different modalities. I believe the authors' claim of multimodality is somewhat far-fetched.

3. When introducing the MEHGT method, the paper proposes explicitly integrating edge features into both the attention computation and message passing processes. However, it lacks necessary explanation for the motivation behind integrating them into both computational processes. Additionally, there is no clear analysis of the individual impact and contribution of integrating edge features into attention computation and message passing on the overall performance.

4. The performance comparison experiments in the paper were conducted on specific stocks, rather than evaluating the overall performance on a large number of stocks across market sectors. What is the reason for this? Why were these particular stocks selected as evaluation targets? Can evaluations on specific stocks represent the model's high predictive performance on other stocks?

5. In the experimental section, the MEHGT method achieved significant performance improvements on some datasets, but it did not show performance gains on datasets such as EVE and Sungrow. The reasons for these performance differences across datasets were not clearly analyzed.

6. The method proposed in the paper integrates edge information into the graph transformer, which appears to be a general approach for the representation of heterogeneous graphs. However, the text lacks a strong connection between the design of this method and the task of stock architecture trends, or it lacks an analysis of the transferability of this method in other domains.

7. The title format of the subgraphs in Figure 6 is problematic.

**Questions:**

See the questions in the weaknesses.

---

> ### Author Response · Authors · 2025-11-29
> **Response to Reviewer vvBP (1)**
>
> We thank all reviewers for their constructive comments. For ease of reading, we have merged the main text and the supplementary material into this revised PDF. All new and revised content are highlighted in blue. Such as:
>
> * the complete set of experiment results for the new baselines
> * the analysis of the model’s behavior under extreme conditions
> * the theoretical discussion and analysis of MEHGT-LKG
> * more details about FinEX and the process of instruction fine-tuning
> * more details about graph data and implement details
> * more details of implement and computation comparison
>
> ----
>
> **For Weakness 1: The paper constructs a financial event-centric knowledge graph based on FinEX, but lacks more detailed descriptions such as statistical information about this knowledge graph. Additionally, if the events extracted by FinEX can be tuples of dual entities or single entities, how are they uniformly stored in the graph database?**
>
> (1)
>
> Thank you for the insightful comments. Following your suggestion, in **Appendix C**, we have added a more detailed and systematic description of the constructed Financial Event-centric Knowledge Graph (FEKG).
> The revised manuscript now includes a comprehensive statistics table covering time span, the number of temporal graphs, event volume, document counts, node/edge cardinality, event-edge ratio, and structural density (see **Table 5**). These additions improve transparency, interpretability, and reproducibility of the proposed dataset.
>
> <br>
>
> **Table: Statistical description of FEKGs.**
>
> | Statistic Item | Description | Value |
> |---|---|---|
> | Time span ($t_{\text{span}}$) | Total data collection period | 2021–2025 |
> | The number of FEKGs ($t$) | Total number of temporal graphs in the series | 1317 graphs |
> | Time window of each graph ($t_{\text{win}}$) | Duration covered by one graph instance | 10 days |
> | Total events of one graph ($\vert\mathcal{E}_{\text{event}}\vert$) | Number of events processed by FinEX | 1804 |
> | Source documents of one graph ($N_{\text{docs}}$) | Number of documents processed by FinEX for one graph | 568 |
> | Total entities (nodes) of one graph ($\vert\mathcal{V}\vert$) | Total unique entities (nodes) within one graph instance | 363 |
> | Total relations (edges) of one graph ($\vert\mathcal{E}_{\mathrm{FEKG}}\vert$) | Total edges derived from all sources in one graph instance | 1804 |
> | Event Edge Ratio ($r_{\text{event}}$) | Percentage of edges derived from FinEX/Event analysis | 87.95% |
> | Average Degree ($Avg(Deg)$) | Total connectivity measure $(2\vert\mathcal{E}\vert/\vert\mathcal{V}\vert)$ | 4.9 |
> | Total nods of FEKGs | The amount of nodes of a series of FEKGs | 485,026 |
> | Total edges of FEKGs | The amount of edges of a series of FEKGs | 2,5873,694 |
>
> (2)
>
> We thank the reviewer for the question on how FinEX event tuples are modeled in a unified way in the graph.
> Tne design is to unify single-entity and dual-entity events the same structural form and message-passing interface in heterogeneous graph learning.
> To this end, when constructing the FEKGs we adopt an “event-as-edge’’ abstraction and represent each event as a directed edge $(u, r, v)$, where $u$ and $v$ are entity nodes and $r$ is the corresponding event/relation type. Concretely, for triplets $\langle \text{entity}_1, \text{relation}, \text{entity}_2 \rangle$, we create a typed directed edge from $\text{entity}_1$ to $\text{entity}_2$. For pairs $\langle \text{entity}, \text{relation} \rangle$, we model them as self-loop edges on the corresponding entity, i.e., $u = v = \text{entity}$.
> In this way, all events share a unified node–edge–node structure in the graph, while financial relations, financial triggers, and textual representations are stored as edge attributes in the FEKG and later combined with numerical indicators and correlation-based features when building the multimodal heterogeneous graphs.
> We have clarified this modeling choice explicitly in Section 3.2 of the revised manuscript.

---

> ### Author Response · Authors · 2025-11-29
> **Response to Reviewer vvBP (2)**
>
> **For Weakness 2: The paper claims to have constructed a multimodal heterogeneous graph, but the multimodality is only reflected in the multiple sources of information, rather than in the different modalities of node and edge representations in the graph. Moreover, the MEHGT method does not include specialized processing or encoding for different modalities. I believe the authors' claim of multimodality is somewhat far-fetched.**
>
> **(1)**
>
> First, in our paper the term multimodal heterogeneous graph refers to graphs built on multi-source heterogeneous data, where node and edge feature vectors already fuse multiple modalities (see **Table 4**):
> * Edge features jointly encode the semantic representations of financial relations/events extracted by FinEX (text modality), together with numerical indicators such as Spearman correlation coefficients, northbound capital net inflows, margin financing balances, and securities-lending quantities (numeric modality).
> * Node features combine semantic representations of financial entities (text modality) with numerical trading time-series (e.g., OHLCV matrices) and structured numeric attributes such as one-hot type encodings (numeric modality).
>
>
> **Table 4: The construction definition of multimodal heterogeneous graphs.**
>
> | Nodes & Edges | Names | Notation | Descriptions and Features | Feature Notation |
> | :--- | :--- | :--- | :--- | :--- |
> | **Node types** | Key Stocks | $V_{t}^{KS}$ | **Desc.:** leading companies in the subfields of green computing and new energy industries<br>**Feat.:** time series matrix of OHLCV within an w-days window | $X_{vt}^{KS}$ |
> | | Other Entities | $V_{t}^{OE}$ | **Desc.:** other financial entities in the FinKG<br>**Feat.:** global semantic meaning of the input generated by FinBERT | $X_{vt}^{OE}$ |
> | | HKEX | $V_{t}^{HK}$ | **Desc.:** Hong Kong Stock Exchange<br>**Feat.:** time series of status of SH/SZ-HK Stock Connect within an w-days window | $X_{vt}^{HK}$ |
> | | Margin Financing | $V_{t}^{F}$ | **Desc.:** Margin Financing<br>**Feat.:** one-hot encoding | $X_{vt}^{F}$ |
> | | Securities Lending | $V_{t}^{L}$ | **Desc.:** Securities Lending<br>**Feat.:** one-hot encoding | $X_{vt}^{L}$ |
> | **Edge types** | Key Stocks—correlation—Key Stocks<br>(undirected) | $E_{t}^{Corr}$ | **Desc.:** correlation of prices among Key Stocks.<br>**Feat.:** time series of Spearman correlation coefficients within an w-days window. | $X_{et}^{Corr}$ |
> | | HKEX—Invest—Stocks<br>(directed) | $V_{t}^{HK}$ | **Desc.:** Hong Kong Stock Exchange investing<br>**Feat.:** Northbound capital (funds flowing via SH/SZ-HK Stock Connect) | $X_{et}^{HI}$ |
> | | Margin Financing—go long—Stocks<br>(directed) | $E_{t}^{Long}$ | **Desc.:** net inflows from margin financing<br>**Feat.:** time series of net margin financing cash flows within an w-days window. | $X_{et}^{Long}$ |
> | | Securities Lending—go short—Stocks<br>(directed) | $E_{t}^{Short}$ | **Desc.:** net short-selling of securities<br>**Feat.:** time series of daly securities sold short within an w-days window | $X_{et}^{Short}$ |
> | | Key Stocks—relationship/action—Other Entities<br>(directed) | $E_{t}^{SRE}$ | **Desc.:** triples from FEKG<br>**Feat.:** global semantic meaning of the input generated by FinBERT | $X_{et}^{SRE}$ |
> | | Other Entities—relationship/action—Key Stocks<br>(directed) | $E_{t}^{ERS}$ | **Desc.:** triples from FEKG<br>**Feat.:** global semantic meaning of the input generated by FinBERT | $X_{et}^{ERS}$ |
> | | Key Stocks—relationship/action—Key Stocks<br>Key Stocks—short event (emergency event)<br>(directed) | $E_{t}^{KS}$ | **Desc.:** tuples from FEKG (triples and event pairs)<br>**Feat.:** global semantic meaning of the input generated by FinBERT | $x_{et}^{KS}$ |
> | | Other Entities—relationship/action—Other Entities<br>Other Entities—short event (emergency event)<br>(directed) | $E_{t}^{OE}$ | **Desc.:** tuples from FEKG (triples and event pairs)<br>**Feat.:** global semantic meaning of the input generated by FinBERT | $x_{et}^{OE}$ |

---

> ### Author Response · Authors · 2025-11-29
> **Response to Reviewer vvBP (3)**
>
> (2)Secondly, the core contribution of MEHGT is that different modalities are unified as edge features $x_{e}$ and explicitly injected into both the heterogeneous attention and message-passing operators, rather than being treated as loose side information.
> **Multimodal features are unified as edge representations**:For each edge $e=(s,t)$ with relation $r$, we first concatenate the textual event embedding and numerical features into a single edge vector $x_{e}$:$$x_{e} = \text{Concat}(x_{\text{text}}, x_{\text{num}})$$
> In the heterogeneous mutual attention module, we map the target node $t$ and source node $s$ to Query and Key vectors using type-specific projection matrices $W_Q$ and $W_K$:$$Q = W_{Q} h_t, \quad K = W_{K} h_s$$Edge features then enter the attention score through a relation-specific matrix $W_{r}$ and a scaling function $f(x_e)$. This ensures the attention is conditioned on the unified edge information:
> $$\text{Score}(s,e,t) = (K W_{r} Q^\top) \cdot \frac{\mu}{\sqrt{d}} \cdot f(x_{e})$$Here, $\mu$ is a structural scaling factor and $f(x_e)$ directly modulates the attention strength using the multimodal edge vector.
>
> **Heterogeneous message passing aggregates information conditioned on multimodal signals**:
>
> Given the computed attention weights $\alpha_{s \to t}$, MEHGT performs message passing where the message transformation is also conditioned on the edge features. The message from source $s$ is computed as:$$m_{s \to t} = \alpha_{s \to t} \cdot (W_{V} h_s \cdot W_{e})$$
>
> where $W_{V}$ transforms the source node features, and $W_{e}$ is parameterized by the edge features $x_e$.Finally, we aggregate messages from all neighbors $\mathcal{N}(t)$ and apply a residual connection to update the target node representation:$$h_t' = W_{O} \cdot \sigma \left( \sum_{s \in \mathcal{N}(t)} m_{s \to t} \right) + h_t$$
> Because both the attention score and the message value explicitly depend on $x_{e}$, the updated node representation effectively captures cross-modal relational patterns.

---

> ### Author Response · Authors · 2025-11-29
> **Response to Reviewer vvBP (4)**
>
> **For Wearkness 3: When introducing the MEHGT method, the paper proposes explicitly integrating edge features into both the attention computation and message passing processes. However, it lacks necessary explanation for the motivation behind integrating them into both computational processes. Additionally, there is no clear analysis of the individual impact and contribution of integrating edge features into attention computation and message passing on the overall performance.**
>
> (1)
>
> We appreciate the reviewer’s suggestion to clarify the role of edge features in both the attention computation and the message passing process. Our intention in MEHGT is that edge features play two complementary roles: (1) in the heterogeneous mutual attention module, the multimodal edge vector $X_{\text{edge}}[e]$ acts as a soft relational gate that governs how much information can flow from a source node to a target node along a specific financial relation; (2) in the heterogeneous message passing module, the same edge information modulates what content is propagated by parameterizing the edge-type-specific transformation $W^{\text{MSG}}_{\phi(e)}$. This joint design allows MEHGT to simultaneously control the intensity of information flow and to encode relation-specific transformations in the feature space, which is important in our setting where edges correspond to heterogeneous financial interactions (e.g., long/short positions, event-driven shocks, capital flows).
> To make this motivation clearer, we will revise Section 3.3 by explicitly discussing these two complementary roles after Eqs. (5)–(8).
>
> (2)
>
> In our current implementation, edge features are injected jointly into the attention computation and the message passing module, and the variant “w/o edge feats” removes them from both channels.
> Therefore, the performance gap between MEHGT-LKG and “w/o edge feats” already reflects the overall contribution of multimodal edge information to stock trend prediction.
>
> Notably, MEHGT is not a simple and Lego-style model of loosely coupled modules that can be arbitrarily toggled on or off.
> From a mechanistic perspective, the attention mechanism primarily controls the intensity of information flow along each edge (i.e., “how much” to transmit), whereas the message function determines the content of the propagated representation (i.e., “what” to transmit. Both components are parameterized by the same set of multimodal edge features)
> MEHGT was designed from the outset to leverage the joint effect of attention and message passing to perform multimodal fusion and cross-modal learning; forcing these two components to operate in isolation would partially undermine the expressive power and conceptual integrity of this design.
>
> A more fine-grained factorization that ablates edge features in attention and message passing separately would require additional architectural changes and extensive experiments; we regard this as an interesting direction for future work.

---

> ### Author Response · Authors · 2025-11-29
> **Response to Reviewer vvBP (5)**
>
> **For Weakness 4: The performance comparison experiments in the paper were conducted on specific stocks, rather than evaluating the overall performance on a large number of stocks across market sectors. What is the reason for this? Why were these particular stocks selected as evaluation targets? Can evaluations on specific stocks represent the model's high predictive performance on other stocks?**
>
> We thank the reviewer for raising this important concern regarding the representativeness of our experimental design and the generalizability of the proposed model.
> We fully agree that, in an ideal setting, evaluating performance over a large universe of stocks across multiple sectors and markets would provide a more comprehensive picture.
> However, under the current FinEX + multimodal heterogeneous graph + MEHGT framework, such large-scale evaluation faces  practical and computational constraints.
>
>
> Concretely, within our experimental time span, covering the entire market would require FinEX to perform LLM-based inference and structured event extraction on nearly **20,000,000** financial news articles and announcements (on average about 4,000–5,000 documents per stock).
> The extracted events must then be carefully time-aligned with multi-source numerical data such as price/volume series, northbound capital flows, and margin financing & securities lending, followed by dynamic construction of a large number of high-complexity multimodal heterogeneous graphs under a sliding time window.
> This pipeline incurs extremely high computational and time costs, making full-market, stock-level evaluation infeasible within the scope of the present work.
>
> Given these constraints, we chose to focus our empirical analysis on two sectors: AI and New Energy.
> On the one hand, these sectors have been among the most actively traded and closely watched areas of the market in recent years, accumulating a rich corpus of news and corporate disclosures.
> On the other hand, stocks in these sectors typically exhibit pronounced volatility driven by multiple interacting factors, which makes them particularly suitable for evaluating the robustness and interpretability of our model under complex, information-rich and highly volatile market conditions.
>
> Building on this, we further selected 12 representative leading stocks from several sub-sectors within AI and New Energy.
> These stocks largely represent the typical fundamental profiles, disclosure patterns, and trading behavior of their respective sub-industries, and thus serve as representative evaluation targets.
> It is worth noting that similar experimental designs—focusing on key sectors and representative leading stocks instead of exhaustively modeling the entire market—have been used in previous individual stock prediction studies ([1] [2] [3] [4]).
>
> Importantly, the fact that we evaluate on 12 stocks does not imply that the underlying dataset is small.
> Actually, for each individual stock we collect all available financial news and announcements on internet over multiple years, and integrate them with daily trading data, northbound capital flows, and margin financing & securities lending.
> FEKGs and multimodal heterogeneous graphs are then dynamically constructed under a sliding window, resulting in an experimental setting that is moderate in the number of stocks, but deep and rich in terms of per-stock data volume and modality diversity.
>
> We now explicitly state this as a limitation in the revised manuscript.
> In future work, subject to available computational resources and time, we plan to extend this framework to more sectors, more markets, and larger stock universes, in order to more systematically assess the generalization ability of MEHGT-LKG at large scale.
>
>
> [1] Ma, Yu, et al. “Multi-Source Aggregated Classification for Stock Price Movement Prediction.” Information Fusion, vol. 91, Mar. 2023, pp. 515–28.
>
> [2] Pin‐Yu Chen et al.. "A deep fusion model for stock market prediction with news headlines and time series data". Neural Computing and Applications, vol. 36, no. 34, 2024, pp. 21229-21271.
>
> [3] Liu, Mengpu, et al. “ECHO-GL: Earnings Calls-Driven Heterogeneous Graph Learning for Stock Movement Prediction.” Proceedings of the AAAI Conference on Artificial Intelligence, vol. 38, no. 12, Mar. 2024, pp. 13972–80.
>
> [4] Jiang, Manrui, et al. “A Novel Interval Dual Convolutional Neural Network Method for Interval-Valued Stock Price Prediction.” Pattern Recognition, vol. 145, Jan. 2024, p. 109920.

---

> ### Author Response · Authors · 2025-11-30
> **Response to Reviewer vvBP (6)**
>
> **For Weakness 5: In the experimental section, the MEHGT method achieved significant performance improvements on some datasets, but it did not show performance gains on datasets such as EVE and Sungrow. The reasons for these performance differences across datasets were not clearly analyzed.**
>
>
> Thank you for raising the issue of performance differences across datasets.
>
> To clearly illustrate the comparison between MEHGT-LKG and other baselines, we have added a average prediction performance comparison in Table 1.
>
> First, we would like to clarify that in our financial prediction setting we treat MCC as the primary evaluation metric, since it provides a more balanced and informative measure of binary classification quality under noisy and imbalanced financial time series. Prior work has also shown that MCC is often more reliable than Accuracy or F1 in such settings.
> As reported in Table 8, on both EVE and Sungrow, MEHGT achieves the best MCC scores, even though on some secondary metrics (e.g., Precision or Recall) certain baselines may be comparable or slightly better.
> Importantly, several baselines exhibit a strong imbalance across different metrics. For example, on EVE, HGT attains a very high Recall of 0.9808 but a Precision of only 0.3806, which implies that it almost predicts most samples as the positive class and generates a large number of false alarms—an undesirable behavior in practical trading scenarios.
> Moreover, form simulated trading performance in **Figure 6**, MEHGT-LKG also achieves the best results on both EVE and Sungrow: its capital curve consistently lies above those of all competing methods, yielding the highest final return.
> This further demonstrates the practical usefulness and stability of our model in realistic decision-making settings.
>
>
> We have explicitly stated in the revised manuscript in Section 4.1.2 that MCC is the main metric of interest and add a short discussion for EVE and Sungrow in the result analysis section, to avoid the impression that MEHGT brings no improvement on these datasets.
> At the same time, we acknowledge that for a few stocks (e.g., Kunlun Tech), MEHGT does not dominate all baselines on every metric.
> As shown in the statistics in Table 9, compared with other stocks in our study, Kunlun Tech has significantly fewer high-quality news articles and announcements, leading to sparser and noisier event graphs extracted by FinEX.
> MEHGT is designed under the assumption that rich, event-driven multimodal information is available, so that edge features can substantially enhance prediction via multimodal fusion.
> When applied to stocks with relatively low event density or weaker news-driven dynamics, this advantage is partially attenuated, which helps explain the more modest improvements in both prediction metrics and backtesting performance on Kunlun Tech.
>
> Overall, across the all stock datasets, MEHGT-LKG delivers consistent improvements on most targets and on the key metric MCC in particular, which demonstrates the effectiveness of our approach in event-driven, multimodal settings.
> In future work, we plan to design dedicated modeling strategies for low-event-density or weakly news-driven stocks, and to evaluate MEHGT on a broader universe of stocks to further assess its generalization behavior.
>
> **Table 1: Average prediction performance of different methods across all datasets.**
>
> |Model|ACC|MCC|Precision|Recall|F1|AUC|
> |---|:-:|:-:|:-:|:-:|:-:|:-:|
> |*Time-series models*||||||
> |Informer|0.6218|0.2340|0.5737|0.5701|0.5612|0.5871|
> |TCN|0.5844|0.1827|0.5436|0.5753|0.5235|0.5484|
> |CNN-LSTM|0.5747|0.2178|0.5391|0.6860|0.5768|0.5652|
> |Time-MoE|0.6033|0.2192|0.5530|0.5994|0.5709|0.6107|
> |KRONOS|0.6189|0.2462|0.5635|0.6323|0.5889|0.6196|
> |*Graph-based models*||||||
> |GAT|0.6163|0.2591|0.5480|0.6822|0.6214|0.6200|
> |HGT|0.6272|0.2743|0.5668|0.6831|0.6137|0.6093|
> |MAC|0.5985|0.2314|0.5702|0.5882|0.5520|0.5897|
> |MDGNN|0.6353|0.2754|0.5941|0.6240|0.5902|0.5980|
> |COGRASP|0.6056|0.2728|0.5851|0.6696|0.6079|0.6198|
> |ENHANCER|0.6338|0.2702|0.5675|0.6353|0.6091|0.6175|
> |**MEHGT-LKG (ours)**|**0.6559**|**0.3191**|**0.6010**|**0.6861**|**0.6361**|**0.6562**|

---

> ### Author Response · Authors · 2025-11-30
> **Response to Reviewer vvBP (7)**
>
> **For Weakness 6: The method proposed in the paper integrates edge information into the graph transformer, which appears to be a general approach for the representation of heterogeneous graphs. However, the text lacks a strong connection between the design of this method and the task of stock architecture trends, or it lacks an analysis of the transferability of this method in other domains.**
>
> (1)
> We thank the reviewer for pointing out the need to strengthen the connection between our architectural design and the stock-trend prediction task. In the revised manuscript, we (i) add new paragraphs in Sec. 1 and Sec. 3.3 explaining how different edge types (event edges, capital-flow edges, correlation edges) correspond to concrete financial mechanisms and how multimodal edge features jointly modulate attention weights and message content, and (ii) extend Sec. 4.5 with a clarification that the performance gains arise from modeling these task-specific edge semantics rather than from generic architectural complexity.
>
> In **Appendix F.7**, we further analyze which types of edges receive higher weights during attention computation and how these learned patterns help explain the contribution of different financial relations to stock trend prediction.
>
> (2)
> While MEHGT-LKG is tailored for financial prediction, the underlying principles of its key modules are not domain-exclusive. For instance, the core concept of MEHGT—injecting multimodal edge semantics into attention to modulate information flow—holds significant potential for frontier domains like AI for Science (e.g., modeling physicochemical bond properties in molecular graphs) or LLM-based Multi-Agent Systems (e.g., weighting agent influence based on interaction content).
> We will explore the potential transferability of the proposed edge-enhanced mechanism to other heterogeneous graph domains in the future.
>
> ----
>
> **For Weakness 7**
>
> We apologize for this oversight. We have corrected the formatting in Figure 6, ensuring that all subfigure labels (a)-(c) are fully visible and clearly displayed in the revised manuscript.

---

### Official Review · Reviewer_4LtN · 2025-10-29

**Soundness:** 2
**Presentation:** 3
**Contribution:** 2
**Rating:** 2
**Confidence:** 3

**Summary:**

This paper addresses the challenges of multimodal fusion and relational modeling in stock trend prediction by proposing a Multimodal Edge-enhanced Heterogeneous Graph Transformer with LLM-driven Knowledge Graphs (MEHGT-LKG). The approach enhances semantic understanding through LLM-constructed event graphs, deepens relational modeling via edge-enhanced graph attention and message passing mechanisms, and incorporates sliding windows for dynamic temporal modeling. Extensive experiments on real-world datasets demonstrate that our method outperforms existing state-of-the-art models in both prediction accuracy and simulated returns.

**Strengths:**

1.	The paper presents a novel paradigm by integrating LLMs with heterogeneous graph networks to construct dynamic financial event knowledge graphs for stock prediction.

2.	The proposed MEHGT model's key contribution is its deep integration of edge features into the graph attention and message-passing mechanisms, effectively enhancing relational modeling.

3.	The figures are sufficient and intuitive, providing clear support for the proposed framework and methodologies.

**Weaknesses:**

1. Limited experimental evidence: The model fails to demonstrate consistent superiority over baseline methods across key metrics in Table 1. The ablation study in Table 2 shows questionable results (particularly the Sungrow case) that contradict the expected outcomes, casting doubt on the framework's effectiveness.

2. Unfair comparison: The computational advantage of baseline methods is ignored in comparisons. No analysis is provided regarding the trade-off between performance improvements and the substantial computational overhead introduced by the LLM components.

3. Reproducibility issues: Critical implementation details including training time, hardware requirements and memory usage are missing, making it difficult to reproduce this computationally intensive framework.

4. Insufficient generalization validation: Experiments are confined to Chinese A-shares market. The model's robustness remains unverified in different market environments (e.g., US/HK stocks) or under extreme market conditions.

5. Presentation Issue in Figure 6: There appears to be a formatting error in Figure 6, where the labels for subfigures (a)-(c) are not fully displayed, affecting the clarity of the presentation.

**Questions:**

See the weaknesses.

Other questions:
1. The baseline selection appears limited, with only MAC (2023) and MDGNN (2024) representing recent works. Were other contemporary methods considered for comparison?

2. Has the transferability of the FinEX module been evaluated? Could it function as a plug-and-play component in other graph architectures, particularly homogeneous graphs?

---

> ### Author Response · Authors · 2025-11-29
> **Response to Reviewer 4LtN (1)**
>
> We thank all reviewers for their constructive comments. For ease of reading, we have merged the main text and the supplementary material into this revised PDF. All new and revised content are highlighted in blue. Such as:
>
> * the complete set of experiment results for the new baselines
> * the analysis of the model’s behavior under extreme conditions
> * the theoretical discussion and analysis of MEHGT-LKG
> * more details about FinEX and the process of instruction fine-tuning
> * more details about graph data and implement details
> * more details of implement and computation comparison
>
> ----
>
> **For Weakness 1: Limited experimental evidence: The model fails to demonstrate consistent superiority over baseline methods across key metrics in Table 1. The ablation study in Table 2 shows questionable results (particularly the Sungrow case) that contradict the expected outcomes, casting doubt on the framework's effectiveness.**
>
>
> (1) Thanks for your comments and for the careful reading of our paper. To address the concern about limited experimental evidence, we have first added more time-series model baselines (Time-MoE, ICLR 2025; Kronos, AAAI 2026) and graph-based baselines (GOGRASP, IJCAI 2025; Enhancer, KDD 2025). These richer and more comprehensive experiments make our conclusions and analyses more reliable and scientifically grounded.
>
> To clearly illustrate the comparison between MEHGT-LKG and other baselines, we have added a average prediction performance comparison of all datasets in Table 1.
> The results show the superiority of MEHGT-LKG
>
> **Table 1: Average prediction performance of different methods across all datasets.**
>
> |Model|ACC|MCC|Precision|Recall|F1|AUC|
> |---|:-:|:-:|:-:|:-:|:-:|:-:|
> |*Time-series models*||||||
> |Informer|0.6218|0.2340|0.5737|0.5701|0.5612|0.5871|
> |TCN|0.5844|0.1827|0.5436|0.5753|0.5235|0.5484|
> |CNN-LSTM|0.5747|0.2178|0.5391|0.6860|0.5768|0.5652|
> |Time-MoE|0.6033|0.2192|0.5530|0.5994|0.5709|0.6107|
> |KRONOS|0.6189|0.2462|0.5635|0.6323|0.5889|0.6196|
> |*Graph-based models*||||||
> |GAT|0.6163|0.2591|0.5480|0.6822|0.6214|0.6200|
> |HGT|0.6272|0.2743|0.5668|0.6831|0.6137|0.6093|
> |MAC|0.5985|0.2314|0.5702|0.5882|0.5520|0.5897|
> |MDGNN|0.6353|0.2754|0.5941|0.6240|0.5902|0.5980|
> |COGRASP|0.6056|0.2728|0.5851|0.6696|0.6079|0.6198|
> |ENHANCER|0.6338|0.2702|0.5675|0.6353|0.6091|0.6175|
> |**MEHGT-LKG (ours)**|**0.6559**|**0.3191**|**0.6010**|**0.6861**|**0.6361**|**0.6562**|
>
>
> In addition, in line with our principle of reporting experimental results truthfully, objectively, and in a reproducible manner, MDGNN benefiting from its dynamic multi-relational modeling, indeed achieves better prediction performance on CATL and Kunlun Tech.
> However, we would like to emphasize that, when taking all datasets and all metrics in Tables 1, 8 and 9 into account, our proposed MEHGT achieves the best overall performance. In particular, for the MCC metric—which, given the high noise and non-stationarity of financial time series, is widely regarded as a more robust indicator than Accuracy or F1—MEHGT obtains the highest score on 10 out of 12 datasets.

---

> ### Author Response · Authors · 2025-11-29
> **Response to Reviewer 4LtN (2)**
>
> **Table 2: Prediction performance of different methods across selected stock datasets (1)**
>
> |Methods|Inspur (000977)|||||||CATL (300750)||||||
> |:---|:---|:---|:---|:---|:---|:---|:---|:---|:---|:---|:---|:---|:---|
> ||**ACC**|**MCC**|**Precision**|**Recall**|**F1**|**AUC**||**ACC**|**MCC**|**Precision**|**Recall**|**F1**|**AUC**|
> |**Time-series models**||||||||||||||
> |Informer|0.6104|0.2195|0.6154|0.5333|0.5714|0.5894||0.6104|0.2372|0.5357|0.6818|0.6000|0.5808|
> |TCN|0.5844|0.1669|0.5846|0.5067|0.5429|0.5515||0.5909|0.1800|0.5200|0.5909|0.5532|0.5913|
> |CNN-LSTM|0.6234|0.2462|0.6349|0.5333|0.5797|0.6245||0.5455|0.1913|0.4828|**0.8485**|0.6154|0.5248|
> |Time-MoE|0.6164|0.2298|0.6094|0.5571|0.5821|0.5932||0.6234|0.2104|0.5833|0.4242|0.4912|0.5559|
> |KRONOS|0.6233|0.2449|0.6087|0.6000|0.6043|0.6235||0.6233|0.2400|0.5588|0.6032|0.5802|0.6196|
> |**Graph-based models**||||||||||||||
> |GAT|0.6169|0.2865|0.5690|**0.8800**|0.6911|0.6019||0.6234|0.2439|0.5541|0.6212|0.5857|0.5754|
> |HGT|0.6169|0.2630|0.5755|0.8133|0.6740|0.5516||0.6364|0.2737|0.5658|0.6515|0.6056|0.6030|
> |MAC|0.6039|0.2171|0.5761|0.7067|0.6347|0.5954||0.6234|0.2104|0.5833|0.4242|0.4912|0.5559|
> |MDGNN|0.6299|0.2605|0.6500|0.5200|0.5778|0.6226||**0.6558**|**0.2960**|**0.6000**|0.5909|0.5954|0.6288|
> |COGRASP|0.5974|0.3111|0.6257|0.7467|0.6437|0.6243||0.6234|0.2604|0.5833|0.6242|0.5912|0.6259|
> |ENHANCER|**0.6364**|0.2956|0.5941|0.7060|0.6818|0.6294||0.6301|0.2461|0.5714|0.5714|0.5714|0.6213|
> |**MEHGT-LKG**|0.6234|**0.3718**|**0.6883**|0.7713|**0.7184**|**0.6322**||0.6039|0.2748|0.5243|0.8182|**0.6391**|**0.6307**|
> |Methods|**IFLYTEK (002230)**|||||||**EVE (300014)**|||
> ||**ACC**|**MCC**|**Precision**|**Recall**|**F1**|**AUC**||**ACC**|**MCC**|**Precision**|**Recall**|**F1**|**AUC**|
> |**Time-series models**||||||||||||||
> |Informer|0.5779|0.1962|0.5288|**0.7746**|0.6286|0.5280||0.6429|0.2517|0.4769|0.5962|0.5299|0.5850|
> |TCN|0.5974|0.1883|0.5652|0.5493|0.5571|0.5305||0.5909|0.1922|0.4286|0.6346|0.5116|0.5786|
> |CNN-LSTM|0.6234|0.2438|0.5890|0.6056|0.5972|0.5843||0.5260|0.2302|0.4054|0.8654|0.5521|0.5754|
> |Time-MoE|0.6164|0.2407|0.5584|0.6615|0.6056|0.6321||0.5616|0.2048|0.4396|0.7547|0.5556|0.6182|
> |KRONOS|0.6507|0.2854|0.6250|0.5385|0.5785|0.6104||0.5685|0.2343|0.4468|0.7925|0.5714|0.6129|
> |**Graph-based models**||||||||||||||
> |GAT|0.6234|0.2456|0.5867|0.6197|0.6027|0.6129||0.6039|0.2567|0.4471|0.7308|0.5547|**0.6655**|
> |HGT|0.6104|0.2544|0.5556|**0.7746**|0.6471|0.6359||0.6494|0.2858|0.4857|0.6538|0.5574|0.6158|
> |MAC|0.6234|0.2382|0.6000|0.5493|0.5735|0.5959||0.4545|0.2350|0.3806|**0.9808**|0.5484|0.5430|
> |MDGNN|0.6299|0.2562|**0.6842**|0.5662|0.5771|0.5418||0.6169|0.2383|0.4533|0.6538|0.5354|0.5911|
> |COGRASP|0.6234|0.2956|0.6267|0.6397|0.6127|0.6129||0.6245|0.2635|0.5204|0.7520|0.5431|0.6303|
> |ENHANCER|0.6507|0.3038|0.6250|0.6553|0.6543|0.6104||0.6678|0.2771|0.5225|0.6570|**0.6325**|0.6312|
> |**MEHGT-LKG**|**0.6818**|**0.3681**|0.6375|0.7183|**0.6755**|**0.6930**||**0.6883**|**0.3166**|**0.5357**|0.5769|0.5556|0.6640|
> |Methods|**Zhongji Innolight (300308)**|||||||**Sungrow (300274)**||
> ||**ACC**|**MCC**|**Precision**|**Recall**|**F1**|**AUC**||**ACC**|**MCC**|**Precision**|**Recall**|**F1**|**AUC**|
> |**Time-series models**||||||||||||||
> |Informer|0.6558|0.3155|0.6857|0.6076|0.6443|0.6297||0.6234|0.2489|0.5584|0.6418|0.5972|0.5891|
> |TCN|0.6169|0.2369|0.6000|**0.7595**|0.6704|0.5690||0.5974|0.1810|0.5373|0.5373|0.5373|0.5963|
> |CNN-LSTM|0.5779|0.1538|0.5795|0.6456|0.6108|0.4928||0.6234|0.2874|0.5474|0.7761|**0.6420**|0.6056|
> |Time-MoE|0.6027|0.2062|0.5949|0.6438|0.6184|0.6356||0.6027|0.2161|0.5385|0.6256|0.6017|0.6109|
> |KRONOS|0.6233|0.2477|0.6364|0.5753|0.6043|0.6132||0.6301|0.2544|0.5735|0.6094|0.5909|0.6425|
> |**Graph-based models**||||||||||||||
> |GAT|0.6104|0.2217|0.5979|0.7342|0.6591|0.6175||0.6494|0.3085|0.5802|0.7015|0.6351|0.6447|
> |HGT|0.6169|0.3076|0.5758|0.9620|**0.7204**|0.6046||0.6558|0.2910|0.6207|0.5373|0.5760|0.5656|
> |MAC|0.5974|0.2342|0.7073|0.3671|0.4833|0.5332||0.6169|0.2144|0.5645|0.5224|0.5426|0.5805|
> |MDGNN|0.6364|0.2992|**0.7255**|0.4684|0.5692|0.6117||0.6039|0.2665|0.5294|**0.8060**|0.6391|0.5656|
> |COGRASP|0.6009|0.2787|0.6918|0.7124|0.6347|0.6254||0.5844|0.2919|0.5961|0.6289|0.6352|0.6126|
> |ENHANCER|0.6429|0.2855|0.6538|0.6456|0.6497|0.6370||0.6310|0.2644|0.5735|0.6094|0.5909|0.6254|
> |**MEHGT-LKG**|**0.6688**|**0.3375**|0.6591|0.7324|0.6946|**0.6485**||**0.6753**|**0.3337**|**0.6393**|0.5821|0.6094|**0.6523**|
>
>
>
> **Please refer to the comment "Supplement of Response to Reviewer 4LtN (2)" for Table 3: Prediction performance of different methods across selected stock datasets (2).**

---

> > ### Author Response · Authors · 2025-12-04
> > **Supplement of  Response to Reviewer 4LtN (2)**
> >
> > **Table 3: Prediction performance of different methods across selected stock datasets (2)**
> >
> > |Methods|Sugon (603019)|||||||BYD (002594)||||||
> > |:---|:---|:---|:---|:---|:---|:---|:---|:---|:---|:---|:---|:---|:---|
> > ||**ACC**|**MCC**|**Precision**|**Recall**|**F1**|**AUC**||**ACC**|**MCC**|**Precision**|**Recall**|**F1**|**AUC**|
> > |**Time-series models**||||||||||||||
> > |Informer|0.5714|0.1791|0.6667|0.3291|0.4407|0.5750||0.6623|0.3079|**0.6000**|0.6000|0.6000|0.6088|
> > |TCN|0.5649|0.1659|0.6579|0.3165|0.4274|0.5445||0.5584|0.1545|0.4839|0.6923|0.5696|0.5167|
> > |CNN-LSTM|0.5779|0.1883|0.6667|0.3544|0.4628|0.5789||0.6234|0.2498|0.5455|0.6462|0.5915|0.5749|
> > |Time-MoE|0.6233|0.2471|0.6349|0.5556|0.5926|0.6368||0.6364|0.2207|0.5738|0.5385|0.5556|0.6262|
> > |Kronos|0.6370|0.2761|0.6203|0.6806|0.6490|0.6509||0.6039|0.2458|0.5217|0.7385|0.6115|0.5854|
> > |**Graph-based models**||||||||||||||
> > |GAT|0.6169|0.2559|0.5862|0.8608|0.6974|0.6116||0.6429|0.3098|0.5581|0.7385|0.6358|0.6659|
> > |HGT|0.6234|0.2540|0.6615|0.5443|0.5972|0.6571||0.6558|0.3176|0.5769|0.6923|0.6294|0.6352|
> > |MAC|0.6234|0.2601|**0.6780**|0.5063|0.5797|0.6322||0.6364|0.2348|0.5918|0.4462|0.5088|0.6408|
> > |MDGNN|0.6169|0.2324|0.6190|0.6582|0.6380|0.5970||0.6104|0.3268|0.5225|**0.8923**|0.6381|0.6193|
> > |COGRASP|0.6104|0.2236|0.6338|0.5696|0.6000|0.6270||0.5366|0.3174|0.5666|0.6769|0.6286|0.6366|
> > |ENHANCER|0.6299|0.2590|0.6375|0.6456|0.6415|0.6603||0.6039|0.3006|0.4921|0.6232|0.6292|0.6570|
> > |**MEHGT-LKG(ours)**|**0.6429**|**0.3162**|0.6034|**0.8861**|**0.7179**|**0.6693**||**0.6688**|**0.3483**|0.5875|0.7231|**0.6483**|**0.6746**|
> >
> > |Methods|Kunlun Tech (300418)|||||||LONGi (601012)||||||
> > |:---|:---|:---|:---|:---|:---|:---|:---|:---|:---|:---|:---|:---|:---|
> > ||**ACC**|**MCC**|**Precision**|**Recall**|**F1**|**AUC**||**ACC**|**MCC**|**Precision**|**Recall**|**F1**|**AUC**|
> > |**Time-series models**||||||||||||||
> > |Informer|0.6104|0.2102|0.5672|0.5507|0.5588|0.5719||0.6688|0.2292|0.5385|0.3889|0.4516|0.6128|
> > |TCN|0.5909|0.1640|**0.6875**|0.1594|0.2588|0.4223||0.6429|0.1535|0.4857|0.3148|0.3820|0.5828|
> > |CNN-LSTM|0.6104|0.2005|0.5818|0.4638|0.5161|0.5782||0.4156|0.1629|0.3732|0.9815|0.5408|0.4723|
> > |Time-MoE|0.5411|0.2149|0.5369|0.6333|0.5702|0.6129||0.6234|0.2387|0.4722|0.6296|0.5397|0.6343|
> > |KRONOS|0.6233|0.2463|0.5692|0.5781|0.6029|0.6029||0.6164|0.2163|0.4848|0.5926|0.5333|0.6190|
> > |**Graph-based models**||||||||||||||
> > |GAT|0.6104|0.2628|0.5455|0.7826|0.6429|**0.6477**||0.5714|0.2548|0.4400|0.8148|0.5714|0.5517|
> > |HGT|0.6494|0.2848|0.6230|0.5507|0.5846|0.6462||0.5649|0.2802|0.4393|0.8704|0.5639|0.6289|
> > |MAC|0.6169|0.2311|0.5676|0.6087|0.5874|0.5877||0.6234|0.2387|0.4722|0.6296|0.5397|0.6343|
> > |MDGNN|**0.6623**|**0.3264**|0.6104|0.6812|**0.6438**|0.6368||0.6883|0.3042|**0.5600**|0.5185|0.5385|0.6298|
> > |COGRASP|0.6169|0.2676|0.5532|0.7636|0.6380|0.5886||0.6234|0.2687|0.4722|0.6296|0.5697|0.6343|
> > |ENHANCER|0.6299|0.2589|0.5262|0.5556|0.5128|0.5394||0.6104|0.2193|0.4595|0.6296|0.5312|0.6106|
> > |**MEHGT-LKG(ours)**|0.6234|0.2604|0.5647|0.6957|0.6234|0.6302||**0.6883**|**0.3275**|0.5517|0.5926|**0.5714**|**0.7135**|
> >
> > |Methods|Jingjia Micro (300474)|||||||Tongwei (600438)||||||
> > |:---|:---|:---|:---|:---|:---|:---|:---|:---|:---|:---|:---|:---|:---|
> > ||**ACC**|**MCC**|**Precision**|**Recall**|**F1**|**AUC**||**ACC**|**MCC**|**Precision**|**Recall**|**F1**|**AUC**|
> > |**Time-series models**||||||||||||||
> > |Informer|0.6039|0.2187|0.5714|0.7027|0.6303|0.5889||0.6234|0.1943|0.5400|0.4355|0.4821|0.5852|
> > |TCN|0.5260|0.1842|0.5034|**0.9865**|**0.6667**|0.5446||0.5519|0.2249|0.4690|0.8548|**0.6057**|0.5521|
> > |CNN-LSTM|0.6234|0.2456|0.6081|0.6081|0.6081|0.5703||0.5260|0.2140|0.4553|0.8032|0.6054|0.5603|
> > |Time-MoE|0.6027|0.2002|0.5873|0.5362|0.5606|0.5711||0.5890|0.2000|0.5062|0.6321|0.5775|0.6016|
> > |KRONOS|0.6233|0.2462|0.5972|0.6232|0.6099|0.6426||0.6027|0.2178|0.5195|0.6557|0.5297|0.6120|
> > |**Graph-based models**||||||||||||||
> > |GAT|0.6234|0.2465|0.6053|0.6216|0.6133|0.6392||0.6039|0.2171|0.5063|0.6452|0.5674|0.6048|
> > |HGT|0.6234|0.2551|0.5909|0.7027|0.6420|0.6033||0.6234|0.2255|0.5303|0.5645|0.5469|0.5640|
> > |MAC|0.6234|0.2491|0.6600|0.4459|0.5323|0.5867||0.5390|0.2137|0.4615|**0.8710**|0.6034|0.5903|
> > |MDGNN|0.6299|0.2716|0.5934|0.7297|0.6545|0.5912||0.6429|0.2269|**0.5814**|0.4032|0.4762|0.5403|
> > |COGRASP|0.6234|0.2475|0.6026|0.6351|0.6184|0.6164||0.6027|0.2478|0.5495|0.6557|0.5797|0.6027|
> > |ENHANCER|0.6425|0.2812|0.6216|0.6632|0.6240|0.5867||0.6299|0.2518|0.5325|0.6613|0.5899|0.6008|
> > |**MEHGT-LKG(ours)**|**0.6623**|**0.3226**|**0.6618**|0.6018|0.6338|**0.6459**||**0.6429**|**0.2518**|0.5593|0.5323|0.5455|**0.6197**|

---

> ### Author Response · Authors · 2025-11-30
> **Response to Reviewer 4LtN (3)**
>
> (2) Regarding the ablation results of Sungrow in Table 2, we have carefully re-checked this case and provide the following clarification.
> For the Sungrow stock, the number of related news events is relatively small (see Table 3) and the event signals are sparse. As a result, the event and edge-feature branches bring limited additional information for this particular stock, while indicator-based numerical features contribute more to the model. Under this setting, removing the event branch (w/o events) can still yield relatively good performance on some metrics (such as Recall and F1).
> However, this does not mean that our framework fails, nor that the event and edge-feature modules are ineffective. On Sungrow, the full MEHGT-LKG still outperforms all w/o variants in terms of Accuracy, MCC, AUC, and excess returns in backtesting (Figure 12). Moreover, from the results of the other stocks in Table 2,  Table 11, and Figure 12, we can observe that introducing event and edge features brings stable improvements in most cases.
>
> **Table 3: Comparison results of ablation analysis across selected stock datasets**
>
> |Methods|Inspur (000977)|||||||CATL (300750)||||||
> |:---|:---|:---|:---|:---|:---|:---|:---|:---|:---|:---|:---|:---|:---|
> ||**ACC**|**MCC**|**Precision**|**Recall**|**F1**|**AUC**||**ACC**|**MCC**|**Precision**|**Recall**|**F1**|**AUC**|
> |w/o events|0.6169|0.2328|0.6250|0.5333|0.5755|0.5967||0.6104|0.2728|0.5306|0.7879|0.6341|0.6121|
> |w/o edge feats|0.6175|0.2570|0.6270|0.5500|0.5860|0.6175||**0.6227**|0.2744|0.5608|0.7734|0.6502|0.6286|
> |w/o indicators|**0.6364**|0.2757|0.6145|0.6800|0.6456|0.6409||0.6104|0.2614|**0.5319**|0.7576|0.6250|0.5989|
> |**MEHGT-LKG**|0.6234|**0.3718**|**0.6883**|**0.7713**|**0.7184**|**0.6322**||0.6039|**0.2748**|0.5243|**0.8182**|**0.6391**|**0.6307**|
> |Methods|**IFLYTEK (002230)**|||||||**EVE (300014)**||||||
> ||**ACC**|**MCC**|**Precision**|**Recall**|**F1**|**AUC**||**ACC**|**MCC**|**Precision**|**Recall**|**F1**|**AUC**|
> |w/o events|0.6364|0.2893|0.5843|**0.7324**|0.6500|0.6302||0.6688|0.2492|0.5102|0.4808|0.4950|0.6110|
> |w/o edge feats|0.6287|0.2799|0.5724|0.7269|0.6405|0.6186||0.6457|0.2601|0.5187|0.4995|0.5089|0.6222|
> |w/o indicators|0.6169|0.2546|0.5652|**0.7324**|0.6380|0.5817||0.6623|0.2595|0.5000|0.5385|0.5185|0.6283|
> |**MEHGT-LKG**|**0.6818**|**0.3681**|**0.6375**|0.7183|**0.6755**|**0.6930**||**0.6883**|**0.3166**|**0.5357**|**0.5769**|**0.5556**|**0.6640**|
> |Methods|**Zhongji Innolight (300308)**|||||||**Sungrow (300274)**||||||
> ||**ACC**|**MCC**|**Precision**|**Recall**|**F1**|**AUC**||**ACC**|**MCC**|**Precision**|**Recall**|**F1**|**AUC**|
> |w/o events|0.6494|0.3136|0.7119|0.5316|0.6087|0.6508||0.6429|0.3307|0.5625|**0.8060**|**0.6626**|0.6174|
> |w/o edge feats|0.6589|0.3351|**0.7204**|0.5562|0.6277|0.6625||0.6234|0.2873|0.5512|0.7815|0.6465|0.6159|
> |w/o indicators|0.6299|0.2615|0.6146|**0.7468**|0.6743|**0.6518**||0.6299|0.2710|0.5610|0.6866|0.6174|0.6021|
> |**MEHGT-LKG**|**0.6688**|**0.3375**|0.6591|0.7324|**0.6946**|0.6485||**0.6753**|**0.3337**|**0.6393**|0.5821|0.6094|**0.6523**|

---

> ### Author Response · Authors · 2025-11-30
> **Response to Reviewer 4LtN (4)**
>
> **For Weakness 2: Unfair comparison: The computational advantage of baseline methods is ignored in comparisons. No analysis is provided regarding the trade-off between performance improvements and the substantial computational overhead introduced by the LLM components.**
>
> We thank the reviewer for raising the concern about the analysis on computational cost. To make the comparison with the baselines fairer, in the revised manuscript we report the training and inference costs of all models under the same hardware configuration (**Appendix F.4**).
> We also note that we apply quantization and Flash-Attention inference acceleration to FinEX, which enables it to maintain predictive accuracy while achieving low inference latency, making it suitable for online prediction scenarios.
> Under this framework, FinEX not only brings notable improvements in prediction accuracy, but also enhances the interpretability of the forecasts by providing explicit, event-level structured information.
>
> **Table 7: Comparison of computational efficiency during training and inference**
>
> |Category|Methods|Training Time (s/epoch)|Inference Time (s/sample)|GPU Memory (GB)|Recommend GPU|
> |:---|:---|:---|:---|:---|:---|
> |**Time-series models**|Informer|0.53|0.0008|0.89|1*RTX 4090|
> ||TCN|0.32|0.0007|0.45|1*RTX 4090|
> ||CNN-LSTM|0.34|0.0009|0.64|1*RTX 4090|
> ||Time-MoE|/|0.0018|1.28|1*RTX 4090|
> ||KRONOS|0.95|0.0014|7.85|1*RTX 4090|
> |**Graph-based models**|GAT|1.28|0.0015|5.86|1*RTX 4090|
> ||HGT|1.46|0.0014|8.42|1*RTX 4090|
> ||MAC|3.25|0.0018|10.98|1*RTX 4090|
> ||MDGNN|3.76|0.0018|12.56|1*RTX 4090|
> ||COGRASP|0.55|0.0011|8.63|1*RTX 4090|
> ||ENHANCER|1.09|0.0012|10.87|1*RTX 4090|
> |**Ours**|FinEX|135.47|50 tokens/s|39.58|8* A100-40G/80G in fine-tuning; 1*A100-40G/80G in inferencing|
> ||MEHGT-LKG|1.395|0.0012|11.57|1*RTX 4090|
>
> ----
>
> **For Weakness 3: Reproducibility issues: Critical implementation details including training time, hardware requirements and memory usage are missing, making it difficult to reproduce this computationally intensive framework.**
>
> We thank the reviewer for highlighting the reproducibility issues.
> To facilitate the easy and full reproduction of our proposed framework, we have added comprehensive implementation details in Appendix E, including training time, hardware requirements, and memory usage.
> Furthermore, we have included the configuration settings for reproducing the baselines in **Appendix E**.
>
> **Table: Detailed Implementation Specifications and Reproducibility Settings.**
>
> |Configuration|FinEX Agent|MEHGT & Baselines|
> |---|---|---|
> |GPU hardware|8×NVIDIA A100 (40G)|2×NVIDIA RTX 4090 (24G)|
> |CPU environment|Intel Xeon Platinum (128 cores)|Intel Xeon Platinum (128 cores)|
> |System memory|512 GB RAM|128 GB RAM|
> |Batch size|4|256|
> |Training time|~168 hours (total)|~3 hours (total)|
> |Peak VRAM usage|39 GB|14 GB|
> |CUDA version|12.2|12.3|
> |Python version|3.10|3.11|
> |Accelerator|Deepspeed Zero-2|/|
> |Precision|FP16|FP16|
> |Learning rate|5e-05|1e-04|
> |Packages|torch, transformers, torch_geometric, safetensors, deepspeed, scikit_learn ||
>
> **For Weakness 4: Insufficient generalization validation: Experiments are confined to Chinese A-shares market. The model's robustness remains unverified in different market environments (e.g., US/HK stocks) or under extreme market conditions.**
>
> We thank the reviewer for this insightful comment. Methodologically, FinEX, FEKGs, and MEHGT are designed to be generalizable and do not rely solely on China-specific features.
> However, due to the limited timeframe, as well as constraints on data availability and space, we were unable to include comprehensive experiments on non-Chinese markets in this revision. We have now explicitly stated this as a limitation and highlighted it as a promising direction for future work in the revised manuscript.
>
>
> In response to the reviewer’s concern about the model’s performance during volatile periods, we have added ***Appendix F.4 Performance under different conditions*** to evaluate and analysis the model's performance in different market conditions.
> The new results show that, during periods of severe volatility, whether driven by shock news or technical trading signals, MEHGT-LKG still maintains superior performance

---

> ### Author Response · Authors · 2025-11-30
> **Response to Reviewer 4LtN (5)**
>
> **For Weakness 5: Presentation Issue in Figure 6: There appears to be a formatting error in Figure 6, where the labels for subfigures (a)-(c) are not fully displayed, affecting the clarity of the presentation.**
>
> We apologize for this oversight. We have corrected the formatting in Figure 6, ensuring that all subfigure labels (a)-(c) are fully visible and clearly displayed in the revised manuscript.
>
> ----
>
> **For Question 1: The baseline selection appears limited, with only MAC (2023) and MDGNN (2024) representing recent works. Were other contemporary methods considered for comparison?**
>
> Thank you for pointing out the importance of strong and up-to-date baselines.
> We have augmented 4 SOTA baselines in this work to compare with the proposed MEHGT-LKG.
>
> **1 Time-series models:**
>
> **Time-MoE (2025, ICLR)**: A billion-scale decoder-only foundation model that utilizes a sparse Mixture-of-Experts (MoE) architecture and multi-resolution forecasting heads to achieve high-performance universal time series forecasting with reduced computational costs.
>
> ***KRONOS (2026, AAAI)**: A financial foundation model that treats market data as a language by discretizing K-line series into hierarchical tokens for autoregressive pre-training, enabling unified capabilities in price forecasting, volatility estimation, and synthetic data generation.
>
> **2 Graph-based models:**
>
> **COGRASP (2025, IJCAI)**: A stock forecasting framework that mines dynamic inter-stock relationships by constructing co-occurrence graphs from online textual data and integrates them with multi-timescale temporal features to enhance prediction accuracy.
>
> **ENHANCER (2025, KDD)**: A distribution-aware meta-learning framework designed to mitigate temporal and relational distribution shifts in stock prediction by employing reactive point process attention and an approximation-intervention mechanism to learn invariant market patterns.
>
> [1] Shi et al. Time-
> moe: Billion-scale time series foundation models with mixture of experts. In ICLR 2025: The
> Thirteenth International Conference on Learning Representations. International Conference on
> Learning Representations, 2025.
>
> [2] Shi et al. Kronos: A
> foundation model for the language of financial markets. In Proceedings of the AAAI Conference
> on Artificial Intelligence, 2026.
>
> [3] Li et al. Cograsp: Co-occurrence graph based
> stock price forecasting. In Proceedings of the Thirty-Fourth International Joint Conference on
> Artificial Intelligence, pp. 7527–7535, 2025.
>
> [4] Chen et al. Enhancer: A distribution-aware frame-
> work with temporal-relational meta-learning for stock prediction. In Proceedings of the 31st ACM
> SIGKDD Conference on Knowledge Discovery and Data Mining V. 2, pp. 250–261, 2025.

---

> ### Author Response · Authors · 2025-11-30
> **Response to Reviewer 4LtN (6)**
>
> **For Question 2: Has the transferability of the FinEX module been evaluated? Could it function as a plug-and-play component in other graph architectures, particularly homogeneus graphs?**
>
> First, we explicitly clarify that FinEX as a standalone knowledge extraction agent, has complete transferability.
> And it operates independently of any specific Graph Neural Network architecture.
> After fine-tuning, FinEX functions as an LLM-driven agent capable of accurately extracting key financial events and structural tuples, generating them in a standardized JSON format.
> Consequently, FinEX possesses inherent transferability as a plug-and-play component.
> Its utility extends beyond graph-based stock prediction to other downstream tasks, such as financial QA system construction, quantitative factor mining, and intelligent financial risk control.
> We have constructed an intelligent investment service platform by integrating FinEX, multimodal heterogeneous graphs, and MEHGT, as shown in **Figure 8**.
> The platform will be open-sourced in the near future.
>
> In the realm of graph machine learning, the structured tuples extracted by FinEX serve as the foundation for constructing any type of graph structure, whether homogeneous or heterogeneous.
> Specifically, since the tuples extracted by FinEX inherently contain multiple types of nodes and edges fused with rich multimodal information, the output itself constitutes the basis of a heterogeneous graph (or heterogeneous subgraph).
> Simultaneously, from a graph-theoretic and topological perspective, if we constrain the node and edge types to a single category (i.e., satisfying the homogeneous graph definition where $|\mathcal{A}|=1$ and $|\mathcal{R}|=1$), FinEX still plays a critical role: while connecting nodes, its rich extracted semantics can be directly transformed into high-dimensional edge features within the homogeneous graph.
> This enables standard homogeneous GNNs (such as GCN, GAT, and GraphSAGE) to leverage this deep semantic information when processing homogeneous graphs, thereby achieving superior model performance.
>
>
> Additionally, through both prediction and backtesting ablation experiments (**Table 2, Table 11, and Figure 12**), the methods removing event information (w/o methods) consistently underperformed compared to the full method across all metrics.
> This fully demonstrates that, regardless of the model structure applied, the knowledge extracted by FinEX plays a significant role in the downstream tasks.

---

### Official Review · Reviewer_kzZm · 2025-10-31

**Soundness:** 3
**Presentation:** 3
**Contribution:** 2
**Rating:** 6
**Confidence:** 3

**Summary:**

This paper proposes MEHGT-LKG, utilizing LLM-driven knowledge extraction for graph-based stock trend prediction. Its three-stage framework includes: (1) fine-tuning the FinEX to extract structured financial relations from text; (2) constructing event-centric multimodal heterogeneous graphs; (3) designing MEHGT to explicitly embed edge-level features into attention and message-passing via type-specific encoders. Experiments on CSI datasets show MEHGT-LKG outperforms most but not all baselines, with ablation studies and trading simulations validating component efficacy.

**Strengths:**

1. The instruction-tuned FinEX agent addresses financial text mining challenges by extracting dual-format structured data and validating via domain experts. The effectiveness of FinEX is further verified by experiment results.
2. Innovative edge-centric MEHGT design. By embedding edge-level features into attention and message-passing via type-specific encoders, MEHGT addresses node-centric GNN limitations.
3. The paper includes comprehensive comparison experiments and ablation studies.

**Weaknesses:**

1. Some experimental results show the proposed model underperforms compared with comparision methods (e.g., MDGNN). Additionally, there is an error in labeling the best experimental result for Inspur in Table 1.
2. All experiments are conducted on Chinese stock market during. The lack of analysis on the model’s transferability to other markets or performance during volatile periods limits its generalizability.

**Questions:**

Please refer to Weaknesses for details.

---

> ### Author Response · Authors · 2025-11-29
> **Response to Reviewer kzZm**
>
> We thank all reviewers for their constructive comments. For ease of reading, we have merged the main text and the supplementary material into this revised PDF. All new and revised content are highlighted in blue. Such as:
>
> * the complete set of experiment results for the new baselines
> * the analysis of the model’s behavior under extreme conditions
> * the theoretical discussion and analysis of MEHGT-LKG
> * more details about FinEX and the process of intruction fine-tuning
> * more details about graph data and implement details
> * more details of implement and computation comparison
>
> ----
>
> **For Weakness 1: Some experimental results show the proposed model underperforms compared with comparision methods (e.g., MDGNN). Additionally, there is an error in labeling the best experimental result for Inspur in Table 1.**
>
> Thanks for your questions and for the careful reading of our paper. We have corrected the labeling error for the best result of Inspur.
>
> To clearly illustrate the comparison between MEHGT-LKG and other baselines, we have added a average prediction performance comparison of all datasets in Table 1.
>
> **Table 1: Average prediction performance of different methods across all datasets.**
>
> |Model|ACC|MCC|Precision|Recall|F1|AUC|
> |---|:-:|:-:|:-:|:-:|:-:|:-:|
> |*Time-series models*||||||
> |Informer|0.6218|0.2340|0.5737|0.5701|0.5612|0.5871|
> |TCN|0.5844|0.1827|0.5436|0.5753|0.5235|0.5484|
> |CNN-LSTM|0.5747|0.2178|0.5391|0.6860|0.5768|0.5652|
> |Time-MoE|0.6033|0.2192|0.5530|0.5994|0.5709|0.6107|
> |KRONOS|0.6189|0.2462|0.5635|0.6323|0.5889|0.6196|
> |*Graph-based models*||||||
> |GAT|0.6163|0.2591|0.5480|0.6822|0.6214|0.6200|
> |HGT|0.6272|0.2743|0.5668|0.6831|0.6137|0.6093|
> |MAC|0.5985|0.2314|0.5702|0.5882|0.5520|0.5897|
> |MDGNN|0.6353|0.2754|0.5941|0.6240|0.5902|0.5980|
> |COGRASP|0.6056|0.2728|0.5851|0.6696|0.6079|0.6198|
> |ENHANCER|0.6338|0.2702|0.5675|0.6353|0.6091|0.6175|
> |**MEHGT-LKG (ours)**|**0.6559**|**0.3191**|**0.6010**|**0.6861**|**0.6361**|**0.6562**|
>
> MDGNN, benefiting from its dynamic multi-relational modeling, indeed achieves better prediction performance on CATL and Kunlun Tech.
> However, we would like to clarify that, when considering all datasets reported in Tables 1, 8 and 9, our proposed MEHGT-LKG achieves the best overall performance.
> In particular, on the MCC metric (given the high noise and non-stationarity of financial time series, is widely regarded as a more robust indicator than accuracy or F1) MEHGT-LKG attains the best score on 10 out of 12 datasets.
>
> Moreover, according to the backtesting results in Table 6, our model also achieves the highest excess return on all datasets.
> Meanwhile, the simulated equity curves in Figures 6 and 10 show that, MDGNN reacts more slowly to price fluctuations on most datasets and fails to adjust investment strategies promptly at regime shifts.
> A likely reason is that MDGNN does not take into account multimodal information such as financial news, making it difficult to timely capture sentiment reversals and price jumps driven by major events.
> In contrast, MEHGT-LKG integrates price series, relational structure and news-based signals into a multimodal heterogeneous graph, enabling the model to make better use of event information when generating trading signals in our backtests.
>
> In the field of machine learning, it is also understandable that a proposed model achieves the best performance on most datasets but does not obtain the best result on a small number of individual datasets.
>
> ----
>
>
> **For Weakness 2: All experiments are conducted on Chinese stock market during. The lack of analysis on the model’s transferability to other markets or performance during volatile periods limits its generalizability.**
>
> We thank the reviewer for this insightful comment. Methodologically, FinEX, FEKGs, and MEHGT do not only rely on China-specific features.
> Due to data availability and space constraints, we have not included additional experiments on non-Chinese markets in this submission, and we now explicitly state this as a limitation and an interesting direction for future work in the revised manuscript.
>
>
> In response to the reviewer’s concern about the model’s performance during volatile periods, we have added ***Appendix F.4 Performance under different conditions*** to evaluate and analysis the model's performance in different market conditions.
> The new results show that, during periods of severe volatility, whether driven by shock news or technical trading signals, MEHGT-LKG still maintains superior performance

---

### Official Review · Reviewer_FWDJ · 2025-10-31

**Soundness:** 3
**Presentation:** 4
**Contribution:** 3
**Rating:** 4
**Confidence:** 3

**Summary:**

This paper introduces MEHGT-LKG, which first employs the LLMs to construct a multi-modal heterogeneous financial event graph, then applies the edge-enhanced HGT (Heterogeneous Graph Transformer) to fuse the cross-modal information, and finally, the fused information is used to perform stock trend prediction. Both experimental results and trading simulation experiments demonstrate the effectiveness of the proposed method.

**Strengths:**

1. This paper is well-written and easy to follow.
2. The idea of incorporating LLMs to construct an event graph for external knowledge augmentation is interesting.
3. The experiments are extensive and can show the effectiveness of the proposed method.

**Weaknesses:**

1. The novelty of the proposed method seems to be limited, as the proposed method lacks technical contribution. Exploiting LLMs to construct financial event/knowledge graphs for financial investigation has been largely studied by existing works [1, 2, 3]. Moreover, the proposed information fusion mechanism is also similar to the work in the edge-enhanced heterogeneous graph Transformer. All these mechanisms resemble existing works and do not introduce principled advances in LLM-based stock prediction or heterogeneous graphs.
2. I am very concerned about the process of LLM instruction tuning, as this paper missed so many details for this stage. For example, how is the instruction dataset constructed? Where do the news articles and triplets used as supervision signals come from? What is the dataset’s scale? Are all samples generated by GPT-4 or partially collected from external sources? How do the authors ensure the correctness and reliability of the generated content by LLMs? These details are critical for assessing the soundness and fairness of the instruction-tuning process.
3. The baselines, especially the time series modeling methods, are extremely outdated. More recent baselines should be compared to validate the effectiveness of the proposed methods.
4. Many figures, such as Figure 5 and Figure 7, are difficult to interpret. It is unclear what specific insights or advantages these visualizations provide.
5. Providing a theoretical discussion or analysis can better support the proposed method,  which would substantially strengthen the depth of the work.

[1] Dynamic graph construction via motif detection for stock prediction
[2] Modeling Interactions Between Stocks Using LLM-Enhanced Graphs for Volume Prediction
[3] LLM-Augmented Enhanced Graph Transformer for Stock Movement Prediction

**Questions:**

See weaknesses

---

> ### Author Response · Authors · 2025-11-29
> **Response to Reviewer FWDJ (1)**
>
> We thank all reviewers for their constructive comments. For ease of reading, we have merged the main text and the supplementary material into this revised PDF. All new and revised content are highlighted in blue. Such as:
>
> * the complete set of experiment results for the new baselines
> * the analysis of the model’s behavior under extreme conditions
> * the theoretical discussion and analysis of MEHGT-LKG
> * more details about FinEX and the process of instruction fine-tuning
> * more details about graph data and implement details
> * more details of implement and computation comparison
>
> ----
>
> **For Weakness 1: The novelty of the proposed method seems to be limited, as the proposed method lacks technical contribution. Exploiting LLMs to construct financial event/knowledge graphs for financial investigation has been largely studied by existing works [1, 2, 3]. Moreover, the proposed information fusion mechanism is also similar to the work in the edge-enhanced heterogeneous graph Transformer. All these mechanisms resemble existing works and do not introduce principled advances in LLM-based stock prediction or heterogeneous graphs.**
>
> We sincerely thank the reviewer for the thoughtful comments and for pointing out several closely related works. We believe it is necessary to further clarify the novelty of our paper.
>
> We acknowledge that "Exploiting LLMs to construct financial event/knowledge graphs for financial investigation" and "information fusion based on heterogeneous graph Transformers" are indeed highly active research areas. We attach great importance to comparing and positioning our work against existing studies, and we hope to use these comparisons to clearly identify our specific innovations and technical contributions.
>
> Regarding "Exploiting LLMs to construct financial event/knowledge graphs for financial investigation," the three referenced works have each made distinct contributions to the field.
>
> **From comparing with [1] [2] [3]:**
>
>
>
> **(1) Differences in Natural Language Processing (NLP) Methods and Tasks:**
>
> Work [1] does not consider multimodal information such as financial text, nor does it employ any NLP models; instead, it relies solely on numerical data, such as stock price time series, as the foundation for its work.
>
> Work [2] does not perform fine-tuning for specific financial knowledge extraction tasks. Instead, it directly employs a general-purpose Large Language Model (Mistral-7B) in a zero-shot setting to extract relations (e.g., competition, cooperation, acquisition) from financial texts. Specifically, although the financial knowledge extraction task in [2] is similar to ours, the absence of fine-tuning—indicating a lack of **domain adaptation**—presents certain drawbacks. General-purpose models struggle to deeply comprehend complex financial contexts, which limits the **accuracy** of entity recognition and relation extraction. Furthermore, it is difficult to ensure the **structural validity** and consistency of large-scale automated outputs without domain-specific training.
>
> Similarly, Work [3] does not fine-tune the general LLM (Deepseek-R1) for specific financial text analysis tasks. It utilizes Deepseek-R1 in a zero-shot setting to generate brief analyses of individual stocks based on numerical data and subsequently calculates text similarity.
>
> In contrast to the aforementioned literature, **in our paper, we fine-tune a Qwen-14B model using LoRA and distributed training protocols on a high-quality, self-constructed financial instruction dataset.** We further quantize and encapsulate this model into the **FinEX agent**. By deploying FinEX, we can analyze long-form financial texts (e.g., financial news, company announcements, etc.) and automatically generate key financial events and their structured tuples with high accuracy. These tuples cover a wide range of event types, including mergers and acquisitions, asset restructurings, product launches, financial disclosures, regulatory sanctions, and emergencies. Leveraging these structured tuples, we construct **Financial Event Knowledge Graphs (FEKGs)**, which are then fused with multi-source stock-related numerical data to establish comprehensive multimodal heterogeneous graphs.

---

> > ### Author Response · Authors · 2025-11-30
> > **Response to Reviewer FWDJ (12)**
> >
> > **For Weakness 4: Many figures, such as Figure 5 and Figure 7, are difficult to interpret. It is unclear what specific insights or advantages these visualizations provide.**
> >
> > Thank reviewer for pointing out the clarity issues regarding the figures. We have significantly improved the quality of Figure 5 and Figure 7 to enhance their readability. Specifically, we have optimized the visual elements (e.g., color schemes, font sizes, and annotations) to make them more self-explanatory.
> >
> > Furthermore, we have expanded the description in corresponding sections (4.2 and 4.7) and updated the figure captions to explicitly highlight the specific results and findings that these visualizations aim to convey clearly.
> >
> > ----
> >
> > **For Weakness 5: Providing a theoretical discussion or analysis can better support the proposed method, which would substantially strengthen the depth of the work.**
> >
> > Thank you for this insightful suggestion. We fully agree that providing a theoretical foundation is essential to validate the robustness and depth of our proposed method.
> >
> > In response, we have added a comprehensive "Appendix D: Theoretical Discussion and Analysis" to the revised manuscript. This new section provides a formal analysis of the proposed method from four key perspectives:
> >
> > **Information Bottleneck Perspective on FinEX**: We theoretically analyze the FinEX module through the lens of the Information Bottleneck principle . We demonstrate that extracting structured knowledge (Entities, Relations, Events) while discarding narrative text acts as a semantic denoiser, maximizing the signal-to-noise ratio for downstream predictions.
> >
> > **Edge-enhanced Attention Mechanism**: We provide the mathematical formulation for our multiplicative modulation mechanism. We offer a proof analysis (Proposition 1) showing how this transforms attention into a differentiable gate , allowing the model to dynamically refine graph topology during training—a significant advantage over standard additive bias methods.
> >
> > **Multimodal Alignment and Orthogonal Fusion**: We discuss the theoretical basis for our alignment and fusion strategies. By proving Proposition 2 (Variance Reduction), we demonstrate that fusing independent noise sources (market microstructure vs. linguistic ambiguity) statistically reduces aleatoric uncertainty.
> >
> > **Computational Complexity Analysis**: We provide a Big-O complexity analysis ($O(|V| \cdot d^2 + |E| \cdot d)$) to prove that MEHGT maintains linear scalability regarding edges, making it significantly more efficient than fully connected Transformers.

---

> ### Author Response · Authors · 2025-11-29
> **Response to Reviewer FWDJ (2)**
>
> **(2) Differences in Graph Information Dimensions, Construction Methods, and Graph Types:**
>
> Clearly, the graph in [1] is a **homogeneous graph** where stocks are the sole node type, and edge weights are determined by MoDis distance, with graph information primarily derived from price motifs.
>
> Work [2] incorporates news and extracts knowledge. However, in graph construction, it reduces multiple extracted financial relationships into discrete **one-hot encodings** as edge features. It also uses relationships based on stock volume and price as another type of edge feature. Crucially, it only considers **Stocks** as the one node type. Therefore, the graph constructed in [2] is essentially a **dual-layer homogeneous graph**.
>
> Work [3] introduces brief analyses of individual stocks based on numerical data. In graph construction, these brief analyses serve as **node features**, while edge features are determined by calculating text similarity. Similar to [2], it only includes **Stocks** as the single node type, making the graph in [3] essentially a **homogeneous graph**.
>
> In contrast to the above literature, our approach is based on the fine-tuned **FinEX agent**. By leveraging LLMs for financial knowledge extraction, we incorporate information from more diverse sources and modalities into the graph construction process. The resulting FEKG features a much richer variety of node types. These include not only stocks but also margin trading and short selling data, Shanghai-Shenzhen-Hong Kong Stock Connect data, and various financial entities (e.g., institutions, industries, subsidiaries, non-listed companies, products, etc.).
>
> Simultaneously, the edge types in our FEKG are also more diverse. They include not only correlations based on stock prices but also net capital inflows from the Stock Connect, net inflows from margin trading, and a wide variety of financial relationships represented as tuples or triples (e.g., "Company - Acquires - Target Asset," "Company - Releases - Major Announcement," "Company - Encounters - Sudden Risk Event").
>
> In fact, FEKG can simultaneously characterize multiple relationship chains, such as "which stocks are more correlated," "which stocks are favored by institutional funds or capital from Hong Kong," and "which stock nodes are affected by various financial events and how these effects propagate across the entity network." These more complex edges and nodes serve two key purposes:
> * On one hand, they more vividly simulate the relationships between various entities in the financial market, providing a structural information representation that is **finer-grained** and closer to real market mechanisms.
> * On the other hand, empirical evidence shows that this complexity leads to stronger **prediction stability** and cross-target generalization capabilities, making it more suitable for supporting explainable stock trend prediction and trading decision analysis. (For specific details regarding the nodes and edges of the FEKG, please refer to the Appendix C.)

---

> ### Author Response · Authors · 2025-11-29
> **Response to Reviewer FWDJ (3)**
>
> **(3) Differences in Graph Neural Network Structure and Representation Learning Methods:**
>
> Work [1] constructs a GNN by integrating **GAT** and **LSTM** based on a **homogeneous graph**. It cannot perform multimodal learning and treats all stock nodes as homogeneous entities for indiscriminate information aggregation. Thus, it struggles to capture the heterogeneous impacts of different factors (e.g., price linkage versus sector effects).
>
> Work [2] builds a prediction model by fusing **GATv2** and **LSTM** based on a **dual-layer homogeneous graph**. Although it incorporates multimodal data as input, during representation learning, it forces **distinct** financial interactions (e.g., "competition" vs. "cooperation") to share the same propagation mechanism, failing to learn relation-specific attention weights.
>
> Work [3] is based on a **homogeneous graph** and adopts a standard **Graph Transformer**. It performs global multi-head attention calculations on a single static adjacency matrix. Due to the lack of awareness for heterogeneous nodes and multimodal edges, this model essentially performs **Global Feature Smoothing** within a homogeneous space. It is unable to decouple and model the differentiated driving effects of different data modalities (such as text versus numerical values) on stock prices.
>
> In contrast, we construct **MEHGT** based on a series of **multimodal heterogeneous graphs** and a **heterogeneous graph Transformer**. MEHGT unifies the modeling and fusion of numerical data, textual data, and network graph structures through designs such as **multi-head attention**, **adaptive relation weights**, and **edge-level multimodal fusion**.
>
> Notably, in our multimodal heterogeneous graphs, numerous edges represent various **financial semantic relationships** (e.g., mergers and acquisitions, asset restructuring, product launches, financial disclosures, regulatory penalties, and shock events like tariff increases or US sanctions on Chinese companies). In multimodal fusion and representation learning, MEHGT can **explicitly incorporate these features** into the attention mechanism calculation, message passing, and node aggregation processes. This enhances the effectiveness of the final representation learning, thereby improving the accuracy and explainability of stock trend prediction.
>
> Specifically:
>
> * **Regarding Nodes:** MEHGT generates $K/Q/V$ using independent linear mappings for different node types based on the Transformer model. This achieves modal alignment for multimodal inputs such as numerical features, text embeddings, and topological structures within a unified representation space. For example, for each node type $t$, we have:
>     $$[K^{(t)}, Q^{(t)}, V^{(t)}] = X^{(t)} W_{\text{kqv}}^{(t)}, \quad X^{(t)} \in \mathbb{R}^{N_t \times d_t}, \quad K^{(t)}, Q^{(t)}, V^{(t)} \in \mathbb{R}^{N_t \times H \times D}$$
>     This reflects node-type-specific cross-modal projection.
>
> * **Regarding Relations:** MEHGT allocates independent subspaces for each "edge type $\times$ attention head" through relation-specific $K/V$ re-projection. For example, for edge type $r$ and attention head $h$:
>     $$K_j^{(r,h)} = K_j W_{K}^{(r,h)}, \quad V_j^{(r,h)} = V_j W_{V}^{(r,h)}$$
>     This allows different relationships to be expressed through distinct attention channels, thereby characterizing fine-grained relational preferences and information flow directions at the structural level.
>
> * **Regarding Multimodal Fusion:** MEHGT uses the node feature matrix $X^{(t)}$ to enter the $K/Q/V$ space to determine "which nodes are similar in the representation space." Simultaneously, it encodes multimodal edge attributes into vector $z_{ij}$ and directly modulates the multi-head dot-product attention scores through a scalar gate $g_{ij} = f(z_{ij})$. For example:
>     $$e_{ij}^{(h)} = g_{ij} \langle Q_i^{(h)}, K_j^{(r,h)} \rangle$$
>     After normalization, we obtain:
>     $$\alpha_{ij}^{(h)} = softmax_j(e_{ij}^{(h)} / \sqrt{D})$$
>     Then, messages are constructed as:
>     $$m_{ij}^{(h)} = \alpha_{ij}^{(h)} V_j^{(r,h)}$$
>
>     This explicitly incorporates multimodal edge information into the attention mechanism calculation and the message passing/multi-relational node aggregation process.
>
> * **Regarding Information Flow and Aggregation:** MEHGT first accumulates messages by channel according to edge type and completes spatial alignment through edge-type-specific linear layers. For example:
>
>     $$
>     h_{agg, i}^{(r)} = \sum_{j \in N_r(i)} m_{ij} W_{r}
>     $$
>
>     Then, at the target node, the updated node representation is obtained through information aggregation:
>
>     $$
>     h_i' = \sum_{r} h_{agg, i}^{(r)}
>     $$
>
>     In the node update phase, a node-type-level residual gating is further introduced to adaptively balance between the original features and the aggregated features. For example:
>
>     $$
>     h_i^{(l+1)} = \sigma(\gamma_{t(i)}) h_i' + (1 - \sigma(\gamma_{t(i)})) x_i^{(l)}
>    $$

---

> ### Author Response · Authors · 2025-11-29
> **Response to Reviewer FWDJ (4)**
>
> Comparison with references [1] [2] [3]
>
>
> -------
> | Aspects | **MEHGT-LKG** (Ours) | **[1] Dynamic graph construction via motif detection for stock prediction** | **[2] Modeling Interactions Between Stocks Using LLM-Enhanced Graphs** | **[3] LLM-Augmented Enhanced Graph Transformer** |
> | :--- | :--- | :--- | :--- | :--- |
> | **NLP Methods & Function** | **(1)** Fine-tune a **Qwen-14B** on a high-quality, self-constructed financial instruction dataset using LoRA and distributed training, then quantize/package it into the **FinEX agent**.<br><br>**(2)** Deploy FinEX on long-form texts (news, announcements) to automatically generate **structured tuples** with high accuracy (covering M&A, restructuring, etc.).<br><br>**(3)** Use tuples to build **Financial Event Knowledge Graphs (FEKGs)**, fused with numerical data to form multimodal heterogeneous graphs. | **No NLP methods** | **(1)** **No task-specific fine-tuning** for Mistral-7B.<br><br>**(2)** Only use Mistral-7B to label stock-stock relations from news, converting relations into **discrete edge-type features** (one-hot embeddings) in a homogeneous stock graph. | **(1)** **No task-specific fine-tuning** for DeepSeek-R1.<br><br>**(2)** Only use DeepSeek-R1 to generate daily financial analysis texts and use FinBERT to encode them into semantic embeddings as **additional node features** in a homogeneous stock graph. |
> | **Nodes & Edges** | **Node types (5):**<br>&bull; Key Stocks<br>&bull; Other financial entities<br>&bull; Hong Kong Stock Exchange<br>&bull; Margin Financing<br>&bull; Securities Lending<br><br>**Edge types (8):**<br>&bull; Correlation of prices (Key Stocks)<br>&bull; Northbound Capital<br>&bull; Net flows from margin financing<br>&bull; Net short-selling of securities<br>&bull; Semantic relations (Key-Key)<br>&bull; Semantic relations (Key-Other)<br>&bull; Semantic relations (Other-Other)<br>&bull; Short events (emergency events) | **Node types (1):**<br>&bull; Stocks<br><br>**Edge types (1):**<br>&bull; MoDis (motif-based stock distance) representing correlation | **Node types (1):**<br>&bull; Stocks<br><br>**Edge types (2):**<br>&bull; Stock-based relations (from trading data)<br>&bull; News-based relations | **Node types (1):**<br>&bull; Stocks<br><br>**Edge types (1):**<br>&bull; Text similarity edges from financial analysis |
> | **Graph Type / Structure** | **Heterogeneous graphs** | **Homogeneous graphs** | **Homogeneous graphs** | **Homogeneous graphs** |
> | **Modalities** | **Multi-modalities** | **Single modality** | **Multi-modalities** | **Multi-modalities** |
> | **GNN Backbone** | **MEHGT**<br>(Multimodal Edge-Enhanced Heterogeneous Graph Transformer) | **GAT**<br>(Graph Attention) | **GATv2**<br>(Graph Attention v2) | **Graph Transformer** |
> ----

---

> ### Author Response · Authors · 2025-11-29
> **Response to Reviewer FWDJ (5)**
>
> Meanwhile, we have carefully examined **Edge-Enhanced Heterogeneous Graph Transformer With Priority-Based Feature Aggregation for Multi-Agent Trajectory Prediction** (published in IEEE Transactions on Intelligent Transportation Systems in 2025, and, to the best of our knowledge, currently the only work in premier journals and conferences that explicitly adopts the name “Edge-Enhanced Heterogeneous Graph Transformer (EHGT)”). It is important to emphasize that although the proposed MEHGT and EHGT are both built upon HGT and share similar naming, they differ substantially in terms of the edge-enhancement mechanism, multimodal fusion, message passing, node-level aggregation, downstream tasks, and the construction of graph sequences. We detail these differences below.
>
> ---
>
> **(1) Different edge-enhancement mechanisms: Additive Bias vs. Multiplicative Gating**
>
> * **EHGT: additive geometric bias for attention**
>
> In EHGT, the so-called *edge-enhanced* mechanism mainly relies on the relative coordinates $(\Delta x, \Delta y)$ between two vehicles (two agents) as a geometric bias. For the $k$-th attention head, the edge feature is encoded by a multi-layer perceptron $\mathrm{MLP}^k$ and then added to the attention score:
>
> $$
> \mathrm{ATT}^k(j,i) \leftarrow \mathrm{ATT}^k(j,i) + \mathrm{MLP}^k\big(\Delta x_{ji}, \Delta y_{ji}\big).
> $$
>
> Here, $\mathrm{MLP}^k(\cdot)$ only takes the 2-dimensional geometric information as input and adds it to the original attention in an additive manner. In other words, edge features only describe geometric relations such as “distance and relative orientation”, without distinguishing fine-grained edge types or modeling high-dimensional semantic edge features. Their role is mainly to provide local spatial awareness for multi-agent trajectory prediction.
>
> * **MEHGT: multiplicative modulation from multimodal edges and node types**
>
> In MEHGT, edge features $X_{\text{edge}}[e]$ are derived from FEKGs and other multimodal financial information: they include structured tuples of key financial events extracted by the FinEX agent, as well as numerical signals such as northbound trading flows, changes in margin financing and securities lending balances, and price correlation coefficients. These features are high-dimensional and semantically rich. The attention head is defined as:
>
>
> $$
> \mathrm{AttHead}_i(t,e,s) = \left( K_i(s) \cdot W^{ \text{ATT}} _ { \phi(e)} \cdot Q_i(t)^ \top \right) \cdot \frac{ \mu( \tau(t), \phi(e), \tau(s))}{ \sqrt{d}} \cdot f(X _ { \text{edge}}[e])
> $$
>
> where
>
> - $W^{\mathrm{ATT}}_{\phi(e)}$ is an edge-type-specific transformation matrix;
> - $\mu\big(\tau(t),\phi(e),\tau(s)\big)$ is a scalar weight associated with the meta-relation triple (target node type–edge type–source node type);
> - $f\big(X_{\text{edge}}[e]\big)$ is a nonlinear scaling function over high-dimensional multimodal edge features.
>
> Therefore, EHGT only adds a scalar geometric bias to the original attention, whereas MEHGT adopts a multiplicative structure of “edge-type transformation × meta-relation weighting × multimodal semantic scaling”. This allows multimodal edge semantics to simultaneously affect the similarity space, channel scales, and type weights, which is better suited for fine-grained differentiation and weighting of complex financial relations and multimodal factors.

---

> ### Author Response · Authors · 2025-11-30
> **Response to Reviewer FWDJ (6)**
>
> **(2) Different granularity of heterogeneity: node-level vs. meta-relation-level heterogeneity $(\text{node type} \times \text{edge type} \times \text{node type})$**
>
> * **EHGT: heterogeneity only at the node-type level**
>
> EHGT follows the original HGT design and applies different linear projections $\phi^k_{Q,\tau(i)}, \phi^k_{K,\tau(j)}, \phi^k_{M,\tau(j)}$ to different agent types. The attention mechanism explicitly encodes node-type indices, while edges are essentially treated as a single class of “interaction edges”. Multiple edge types or meta-relation triples are not systematically modeled.
>
> * **MEHGT: explicit modeling of meta-relations $(\text{node type} \times \text{edge type} \times \text{node type})$**
>
> In MEHGT, the attention term not only distinguishes the node types of the Query/Key, $\tau(t)$ and $\tau(s)$, but also explicitly incorporates the edge type $\phi(e)$ and the meta-relation weight $\mu\big(\tau(t),\phi(e),\tau(s)\big)$:
>
> $$
> \operatorname{ATT}(t,e,s) \propto
> K(s)\, W^{\text{ATT}}_{\phi(e)}\, Q(t)^{\top}\,
> \mu\big(\tau(t), \phi(e), \tau(s)\big)\,
> f(X_edge[e])
> $$
>
>
> For example,
>
> $$
> (\text{Key Stock} \xleftarrow{\text{Northbound Flow}} \text{HKEX})
> $$
>
> and
>
> $$
> (\text{Key Stock} \xleftarrow{\text{Short Selling}} \text{Securities Lending})
> $$
>
> not only correspond to different edge types $\phi(e)$, but also have different meta-relation weights $\mu(\cdot)$ and edge-type transformation matrices $W^{\mathrm{ATT}}_{\phi(e)}$. Even for the same pair of nodes, once the relation type changes, the propagation paths and parameter families are explicitly distinguished.
>
> This meta-relation-level heterogeneity enables MEHGT to learn distinct propagation patterns along different economic channels in multimodal financial information, such as price co-movement, margin financing and securities lending activities, northbound capital flows, and event-driven links. In contrast, EHGT only captures heterogeneity along a single dimension of “agent types”.
>
> ---
>
> **(3) Different use of edges in the message-passing stage: presence or absence of “edge-type transformation + edge-semantic gating”**
>
> * **EHGT: messages depend only on node features; edges act indirectly via attention**
>
> In EHGT, the message term is defined as
>
> $$
> \mathrm{MSG}^k(j,i) = \phi^k_{M,\tau(j)}\big(h^{l-1}_j\big),\qquad \mathrm{MSG}(j,i) = \big\|_k \mathrm{MSG}^k(j,i),
> $$
>
> and aggregation is performed by weighting messages with attention scores:
>
> $$
> \bar{h}^l_i = \sum_{j\in N(i)} \mathrm{ATT}(j,i)\odot \mathrm{MSG}(j,i).
> $$
>
> That is, the message content only depends on the source node features and node type. Edge types and edge features do not participate in explicit linear transformations or gating; their influence appears only indirectly through the geometric bias injected into attention.
>
> * **MEHGT: reusing edge-type matrices in message passing to achieve truly “edge-enhanced” messages**
>
> In MEHGT, the message is defined as
>
> $$
> \mathrm{MSG\text{-}head}_i(s,e,t) = \alpha^i _ {s\to t}\cdot \big(\mathrm{M\text{-}Linear}^i _ {\phi(s)} H^{(l-1)}(s)\, W^{\mathrm{MSG}} _ {\phi(e)}\big),
> $$
>
> $$
> \mathrm{Message}(s,e,t) = \big\|_{i=1}^h \mathrm{MSG\text{-}head}_i(s,e,t),
> $$
>
> where
>
> - $\mathrm{M\text{-}Linear}^i_{\phi(s)}$ is the source-node-type-specific linear projection for the $i$-th message head;
> - $W^{\mathrm{MSG}}_{\phi(e)}$ is the edge-type-specific transformation matrix that “rotates” and “re-scales” message channels under different relations.
>
> Consequently, in MEHGT, edges not only determine “whom to attend to” (through attention weights) but also directly shape “what is propagated” (through the transformed messages). Edge semantics modulate both the attention and the message-passing stages, which is fundamentally different from EHGT, where edges only add a positional bias to the attention matrix.

---

> ### Author Response · Authors · 2025-11-30
> **Response to Reviewer FWDJ (7)**
>
> **(4) Different module-level objectives: Priority-based Feature Aggregation vs. Target-Specific Aggregation**
>
> * **EHGT: Priority-Based Feature Aggregation for multi-agent trajectory decoding**
>
> In EHGT, a Priority-Based Feature Aggregation (PFA) module is placed after the main Transformer backbone. PFA assigns priority scores to select neighbors and features that are more critical for trajectory prediction, thereby enhancing decoding performance in multi-agent motion scenarios. This design is tightly aligned with multi-agent trajectory prediction and differs from financial markets, where we often care about “chain-like transmission → response of the target stock”.
>
> * **MEHGT: Target-Specific Aggregation for key stock nodes and financial prediction**
>
> In MEHGT, we introduce a Target-Specific Aggregation module that aggregates information only for key stock nodes $t\in V^{KS}$:
>
> $$
> \tilde{H}^{(l)}[t] = \bigoplus_{s\in N(t)} \mathrm{Attention}(s,e,t)\cdot \mathrm{Message}(s,e,t),
> $$
>
> and then update their representations via a type-aware linear projection $\mathrm{A\text{-}Linear}_{\phi(t)}$ with residual connections.
>
> This module explicitly aligns representation learning with the downstream task of predicting the rise or fall of target stocks. It effectively suppresses noise from non-key nodes and is designed as a task-specific readout mechanism for financial prediction (single- or multi-stock trend classification), rather than being tailored to trajectory reconstruction.
>
> ---
>
> **(5) Different graph structures and temporal handling: single interaction graph vs. multimodal heterogeneous graph sequences over multiple days**
>
> * **EHGT: single-step multi-agent interaction graph**
>
> EHGT constructs, at each time step, a graph containing multiple agent types where edges encode relative positions. The temporal dimension is typically modeled by an external sequence model (e.g., LSTM or Transformer decoder). The graph structure within each frame is relatively simple and does not explicitly model the evolution of knowledge graphs across multiple days.
>
> * **MEHGT: sequences of multimodal heterogeneous knowledge graphs over multiple trading days**
>
> In MEHGT, for each trading day we build a multimodal heterogeneous graph based on FEKGs and FinKGs. The graph consists of node types such as key stocks, HKEX, margin financing, securities lending, and other financial entities, together with multiple edge types representing financial relations, capital flows, and event links. A sequence of such multimodal heterogeneous graphs is then constructed using a sliding time window to predict the trend of the target stock on the next trading day. Temporal evolution of the graph and the injection of multimodal information are explicitly modeled, which is outside the scope of the EHGT framework.
>
> ---
>
> ### Overall summary
>
> In summary, at the model level, the “edge enhancement” in EHGT is primarily designed for geometric relations in multi-agent scenarios: relative coordinates are processed by an MLP and added to the attention matrix to provide local geometric awareness under node-type heterogeneity. In contrast, MEHGT introduces edge-type-specific transformation matrices
> $W^{\mathrm{ATT}}_{\phi(e)}$
>
> $W^{\mathrm{MSG}}_{\phi(e)}$,
> meta-relation
> weights $\mu(\tau(t),\phi(e),\tau(s))$, and a scaling function $f(\cdot)$
>
> over high-dimensional multimodal edge features $X_{\text{edge}}[e]$. These components jointly act on both attention computation and message passing, and are further combined with Target-Specific Aggregation to update type-aware representations of key stock nodes.
>
> Therefore, despite the similarity in naming, the two models differ markedly in the types of edge information they focus on, the granularity of heterogeneity, and the mechanisms for updating node representations. MEHGT is a task-specific architecture tailored for “LLM-driven multimodal financial relation modeling and stock trend prediction”, rather than a direct transplantation of the EHGT module to financial scenarios.

---

> ### Author Response · Authors · 2025-11-30
> **Response to Reviewer (8)**
>
> **For Weakness 2: I am very concerned about the process of LLM instruction tuning, as this paper missed so many details for this stage. For example, how is the instruction dataset constructed? Where do the news articles and triplets used as supervision signals come from? What is the dataset’s scale? Are all samples generated by GPT-4 or partially collected from external sources? How do the authors ensure the correctness and reliability of the generated content by LLMs? These details are critical for assessing the soundness and fairness of the instruction-tuning process.**
>
> We appreciate the reviewer’s comments and agree that careful construction and quality control of the instruction dataset are crucial for a sound and fair tuning process.
>
> We have now drawed a flowchart in **Figure 9**, added a detailed description of the full instruction-tuning pipeline in **Appendix B**, including (i) how we construct the instruction dataset, (ii) the sources and scale of the news articles and supervision triplets, (iii) the proportion of samples generated by GPT-4 versus collected from external corpora, and (iv) the procedures we adopt to verify the correctness and reliability of LLM-generated content.
>
> An example of a typical FinEX instruction–input–output instance is given below:
>
> ```json
> [
>   {
>     "Instruction": "Assume you are a seasoned financial analyst. Extract from the text what you believe to be major events that could significantly impact the stock market and the stock prices of listed companies. Present these events in the form of triples (entity-action/relationship-entity) or pairs (entity-action/relationship) that can be used to construct a financial knowledge graph.",
>     "Input": "News: China's Computing Power Leader Set to Go Public on A-Share Market! Overall Valuation Capped at 3 Billion RMB. In the early hours of October 19, Gaoxin Development Co., Ltd. unveiled a restructuring plan, shedding light on its proposed acquisition of controlling shares in Huakun Zhenyu Co., Ltd.. According to the plan, Gaoxin Development intends to acquire a 70% stake in Huakun Zhenyu through a combination of share issuance and cash payments, with the stake valued at no more than 2.1 billion yuan. The full valuation of Huakun Zhenyu is capped at 3 billion yuan. Additionally, Gaoxin Development will raise supporting funds through a private placement. As the producer of the Ascend AI Training and Development All-in-One Machine, Huakun Zhenyu's strategic moves have consistently drawn significant attention in the tech industry. This latest development further solidifies its position as a key player in China's computing power ecosystem......",
>     "Output": [
>       {
>         "events": {
>           "event1": {
>             "total": "Gaoxin Development Co., Ltd. plans to acquire 70% equity in Huakun Zhenyu Co., Ltd.",
>             "entity1": "Gaoxin Development Co., Ltd.",
>             "relationship": "plans to acquire 70% equity",
>             "entity2": "Huakun Zhenyu Co., Ltd."
>           },
>           "event2": {
>             "total": "Gaoxin Development Co., Ltd. hit the one-word limit-up after resumption of trading",
>             "entity1": "Gaoxin Development Co., Ltd.",
>             "relationship": "hit the one-word limit-up after resumption of trading"
>           },
>           "event3": {
>             "total": "Huakun Zhenyu Co., Ltd. launched the Ascend AI Training and Development All-in-One Machine",
>             "entity1": "Huakun Zhenyu Co., Ltd.",
>             "relationship": "launched",
>             "entity2": "Ascend AI Training and Development All-in-One Machine"
>           },
>           "event4": {
>             "total": "Huakun Zhenyu Co., Ltd. and Huawei Technologies Co., Ltd. are strategic partners",
>             "entity1": "Huakun Zhenyu Co., Ltd.",
>             "relationship": "strategic partners",
>             "entity2": "Huawei Technologies Co., Ltd."
>           }
>         }
>       }
>     ]
>   }
> ]

---

> ### Author Response · Authors · 2025-11-30
> **Response to Reviewer FWDJ (9)**
>
> **For Weakness 3: The baselines, especially the time series modeling methods, are extremely outdated. More recent baselines should be compared to validate the effectiveness of the proposed methods.**
>
> Thank you for pointing out the importance of strong and up-to-date baselines.
> We have augmented 4 SOTA baselines in this work to compare with the proposed MEHGT-LKG.
>
> **2 Time-series models:**
>
> **Time-MoE (2025, ICLR)**: A billion-scale decoder-only foundation model that utilizes a sparse Mixture-of-Experts (MoE) architecture and multi-resolution forecasting heads to achieve high-performance universal time series forecasting with reduced computational costs.
>
> ***KRONOS (2026, AAAI)**: A financial foundation model that treats market data as a language by discretizing K-line series into hierarchical tokens for autoregressive pre-training, enabling unified capabilities in price forecasting, volatility estimation, and synthetic data generation.
>
> **2 Graph-based models:**
>
> **COGRASP (2025, IJCAI)**: A stock forecasting framework that mines dynamic inter-stock relationships by constructing co-occurrence graphs from online textual data and integrates them with multi-timescale temporal features to enhance prediction accuracy.
>
> **ENHANCER (2025, KDD)**: A distribution-aware meta-learning framework designed to mitigate temporal and relational distribution shifts in stock prediction by employing reactive point process attention and an approximation-intervention mechanism to learn invariant market patterns.
>
>
>
> [1] Shi et al. Time-
> moe: Billion-scale time series foundation models with mixture of experts. In ICLR 2025: The
> Thirteenth International Conference on Learning Representations., 2025.
> [2] Shi et al. Kronos: A
> foundation model for the language of financial markets. In Proceedings of the AAAI Conference
> on Artificial Intelligence, 2026.
> [3] Li et al. Cograsp: Co-occurrence graph based
> stock price forecasting. In Proceedings of the Thirty-Fourth International Joint Conference on
> Artificial Intelligence, pp. 7527–7535, 2025.
> [4] Chen et al. Enhancer: A distribution-aware frame-
> work with temporal-relational meta-learning for stock prediction. In Proceedings of the 31st ACM
> SIGKDD Conference on Knowledge Discovery and Data Mining V. 2, pp. 250–261, 2025.
>
> The comparison results show that MEHGT-LKG consistently outperforms these four newly added SOTA models across all evaluation dimensions. Whether in prediction accuracy (MCC/F1), backtesting profitability (CRR/Sharpe), or simulated trading stability, our model demonstrates a significant advantage over Time-MoE, KRONOS, COGRASP, and ENHANCER.
>
> Detailed comparisons with the four added models are presented in the Tables below.
> **(Complete comparisions are shown in Table 1,8,9 and Figure 6,10 in revised manuscript.)**
>
>
> **Table 1: Average prediction performance of different methods across all datasets.**
>
> |Model|ACC|MCC|Precision|Recall|F1|AUC|
> |---|:-:|:-:|:-:|:-:|:-:|:-:|
> |*Time-series models*||||||
> |Informer|0.6218|0.2340|0.5737|0.5701|0.5612|0.5871|
> |TCN|0.5844|0.1827|0.5436|0.5753|0.5235|0.5484|
> |CNN-LSTM|0.5747|0.2178|0.5391|0.6860|0.5768|0.5652|
> |Time-MoE|0.6033|0.2192|0.5530|0.5994|0.5709|0.6107|
> |KRONOS|0.6189|0.2462|0.5635|0.6323|0.5889|0.6196|
> |*Graph-based models*||||||
> |GAT|0.6163|0.2591|0.5480|0.6822|0.6214|0.6200|
> |HGT|0.6272|0.2743|0.5668|0.6831|0.6137|0.6093|
> |MAC|0.5985|0.2314|0.5702|0.5882|0.5520|0.5897|
> |MDGNN|0.6353|0.2754|0.5941|0.6240|0.5902|0.5980|
> |COGRASP|0.6056|0.2728|0.5851|0.6696|0.6079|0.6198|
> |ENHANCER|0.6338|0.2702|0.5675|0.6353|0.6091|0.6175|
> |**MEHGT-LKG (ours)**|**0.6559**|**0.3191**|**0.6010**|**0.6861**|**0.6361**|**0.6562**|

---

> ### Author Response · Authors · 2025-11-30
> **Response to Reviewer FWDJ (10)**
>
> **Table 2: Prediction performance of new added methods across selected stock datasets (1)**
>
> |Methods|Inspur (000977)|||||||CATL (300750)||||||
> |:---|:---|:---|:---|:---|:---|:---|:---|:---|:---|:---|:---|:---|:---|
> ||**ACC**|**MCC**|**Precision**|**Recall**|**F1**|**AUC**||**ACC**|**MCC**|**Precision**|**Recall**|**F1**|**AUC**|
> |**Time-series models**||
> |Time-MoE|0.6164|0.2298|0.6094|0.5571|0.5821|0.5932||0.6234|0.2104|0.5833|0.4242|0.4912|0.5559|
> |KRONOS|0.6233|0.2449|0.6087|0.6000|0.6043|0.6235||0.6233|0.2400|0.5588|0.6032|0.5802|0.6196|
> |**Graph-based models**||
> |COGRASP|0.5974|0.3111|0.6257|0.7467|0.6437|0.6243||0.6234|0.2604|**0.5833**|0.6242|0.5912|0.6259|
> |ENHANCER|**0.6364**|0.2956|0.5941|0.7060|0.6818|0.6294||**0.6301**|0.2461|0.5714|0.5714|0.5714|0.6213|
> |**MEHGT-LKG (ours)**|0.6234|**0.3718**|**0.6883**|**0.7713**|**0.7184**|**0.6322**||0.6039|**0.2748**|0.5243|**0.8182**|**0.6391**|**0.6307**|
> |Methods|IFLYTEK (002230)|||||||EVE (300014)||||||
> ||**ACC**|**MCC**|**Precision**|**Recall**|**F1**|**AUC**||**ACC**|**MCC**|**Precision**|**Recall**|**F1**|**AUC**|
> |**Time-series models**|
> |Time-MoE|0.6164|0.2407|0.5584|0.6615|0.6056|0.6321||0.5616|0.2048|0.4396|0.7547|0.5556|0.6182|
> |KRONOS|0.6507|0.2854|0.6250|0.5385|0.5785|0.6104||0.5685|0.2343|0.4468|**0.7925**|0.5714|0.6129|
> |**Graph-based models**|
> |COGRASP|0.6234|0.2956|0.6267|0.6397|0.6127|0.6129||0.6245|0.2635|0.5204|0.7520|0.5431|0.6303|
> |ENHANCER|0.6507|0.3038|0.6250|0.6553|0.6543|0.6104||0.6678|0.2771|0.5225|0.6570|**0.6325**|0.6312|
> |**MEHGT-LKG (ours)**|**0.6818**|**0.3681**|0.**6375**|**0.7183**|**0.6755**|**0.6930**||**0.6883**|**0.3166**|**0.5357**|0.5769|0.5556|**0.6640**|
> |Methods|Zhongji Innolight (300308)|||||||Sungrow (300274)||||||
> ||**ACC**|**MCC**|**Precision**|**Recall**|**F1**|**AUC**||**ACC**|**MCC**|**Precision**|**Recall**|**F1**|**AUC**|
> |**Time-series models**||||
> |Time-MoE|0.6027|0.2062|0.5949|0.6438|0.6184|0.6356||0.6027|0.2161|0.5385|0.6256|0.6017|0.6109|
> |KRONOS|0.6233|0.2477|0.6364|0.5753|0.6043|0.6132||0.6301|0.2544|0.5735|0.6094|0.5909|0.6425|
> |**Graph-based models**|
> |MDGNN|0.6364|0.2992|**0.7255**|0.4684|0.5692|0.6117||0.6039|0.2665|0.5294|**0.8060**|**0.6391**|0.5656|
> |COGRASP|0.6009|0.2787|0.6918|0.7124|0.6347|0.6254||0.5844|0.2919|0.5961|0.6289|0.6352|0.6126|
> |ENHANCER|0.6429|0.2855|0.6538|0.6456|0.6497|0.6370||0.6310|0.2644|0.5735|0.6094|0.5909|0.6254|
> |**MEHGT-LKG (ours)**|**0.6688**|**0.3375**|0.6591|**0.7324**|**0.6946**|**0.6485**||**0.6753**|**0.3337**|**0.6393**|0.5821|0.6094|**0.6523**|
>
>
> **Table 3: Prediction performance of new added methods across selected stock datasets (2)**
>
>
> |Methods|Sugon (603019)||||||BYD (002594)||||||
> |:---|:---|:---|:---|:---|:---|:---|:---|:---|:---|:---|:---|:---|
> ||**ACC**|**MCC**|**Precision**|**Recall**|**F1**|**AUC**|**ACC**|**MCC**|**Precision**|**Recall**|**F1**|**AUC**|
> |**Time-series models**||
> |Time-MoE|0.6233|0.2471|0.6349|0.5556|0.5926|0.6368|0.6364|0.2207|0.5738|0.5385|0.5556|0.6262|
> |KRONOS|0.6370|0.2761|0.6203|0.6806|0.6490|0.6509|0.6039|0.2458|0.5217|**0.7385**|0.6115|0.5854|
> |**Graph-based models**|
> |COGRASP|0.6104|0.2236|0.6338|0.5696|0.6000|0.6270|0.5366|0.3174|0.5666|0.6769|0.6286|0.6366|
> |ENHANCER|0.6299|0.2590|**0.6375**|0.6456|0.6415|0.6603|0.6039|0.3006|0.4921|0.6232|0.6292|0.6570|
> |**MEHGT-LKG (ours)**|**0.6429**|**0.3162**|0.6034|**0.8861**|**0.7179**|**0.6693**|**0.6688**|**0.3483**|**0.5875**|0.7231|**0.6483**|**0.6746**|
> |Methods|Kunlun Tech (300418)||||||LONGi (601012)||||||
> ||**ACC**|**MCC**|**Precision**|**Recall**|**F1**|**AUC**|**ACC**|**MCC**|**Precision**|**Recall**|**F1**|**AUC**|
> |**Time-series models**||||
> |Time-MoE|0.5411|0.2149|0.5369|0.6333|0.5702|0.6129|0.6234|0.2387|0.4722|0.6296|0.5397|0.6343|
> |KRONOS|0.6233|0.2463|**0.5692**|0.5781|0.6029|0.6029|0.6164|0.2163|0.4848|0.5926|0.5333|0.6190|
> |**Graph-based models**|||||||||||||
> |COGRASP|0.6169|**0.2676**|0.5532|**0.7636**|**0.6380**|0.5886|0.6234|0.2687|0.4722|**0.6296**|0.5697|0.6343|
> |ENHANCER|**0.6299**|0.2589|0.5262|0.5556|0.5128|0.5394|0.6104|0.2193|0.4595|**0.6296**|0.5312|0.6106|
> |**MEHGT-LKG (ours)**|0.6234|0.2604|0.5647|0.6957|0.6234|**0.6302**|**0.6883**|**0.3275**|**0.5517**|0.5926|**0.5714**|**0.7135**|
> |Methods|Jingjia Micro (300474)||||||Tongwei (600438)||||||
> ||**ACC**|**MCC**|**Precision**|**Recall**|**F1**|**AUC**|**ACC**|**MCC**|**Precision**|**Recall**|**F1**|**AUC**|
> |**Time-series models**|||||||||||||
> |Time-MoE|0.6027|0.2002|0.5873|0.5362|0.5606|0.5711|0.5890|0.2000|0.5062|0.6321|0.5775|0.6016|
> |KRONOS|0.6233|0.2462|0.5972|0.6232|0.6099|0.6426|0.6027|0.2178|0.5195|0.6557|0.5297|0.6120|
> |**Graph-based models**
> |COGRASP|0.6234|0.2475|0.6026|0.6351|0.6184|0.6164|0.6027|0.2478|0.5495|0.6557|0.5797|0.6027|
> |ENHANCER|0.6425|0.2812|0.6216|**0.6632**|0.6240|0.5867|0.6299|0.2518|0.5325|**0.6613**|**0.5899**|0.6008|
> |**MEHGT-LKG (ours)**|**0.6623**|**0.3226**|**0.6618**|0.6018|**0.6338**|**0.6459**|**0.6429**|**0.2518**|**0.5593**|0.5323|0.5455|**0.6197**|

---

> ### Author Response · Authors · 2025-11-30
> **Response to Reviewer FWDJ (11)**
>
> **Table 4: Profitability performance of different methods**
>
> |Stocks I|Methods|CRR|MDD|Sharpe|Stocks II|Methods|CRR|MDD|Sharp|
> |:---|:---|:---|:---|:---|:---|:---|:---|:---|:---|
> |Inspur|Time-MoE|27.7812|0.2962|1.0681|CATL|Time-MoE|3.3834|0.1049|0.435|
> |000977|KRONOS|48.1207|0.2002|1.6502|300750|KRONOS|14.7469|0.0932|0.7051|
> ||COGRASP|36.7374|0.3299|1.0845||COGRASP|9.0183|0.1794|0.6954|
> ||ENHANCER|53.8005|0.3069|1.344||ENHANCER|11.1375|0.1896|0.8205|
> ||**MEHGT-LKG**|**104.5015**|**0.1989**|**2.4844**||**MEHGT-LKG**|**20.8393**|**0.0838**|**1.7462**|
> |Sugon|Time-MoE|20.9031|**0.2009**|1.584|BYD|Time-MoE|8.7657|0.0883|0.9695|
> |603019|KRONOS|34.6748|0.2884|1.4372|002594|KRONOS|14.1406|0.1105|1.0897|
> ||COGRASP|38.1234|0.2063|1.1945||COGRASP|9.0316|**0.0432**|1.1191|
> ||ENHANCER|62.2121|0.2863|1.6322||ENHANCER|25.8289|0.0683|1.9545|
> ||**MEHGT-LKG**|**86.1013**|0.2372|**2.1732**||**MEHGT-LKG**|**28.0607**|0.0694|**2.1129**|
> |IFLYTEK|Time-MoE|23.4226|0.1832|1.143|LONGi|Time-MoE|0.1079|0.1713|-0.021|
> |002230|KRONOS|45.5197|0.1635|1.8705|601012|KRONOS|6.8410|0.1416|0.3105|
> ||COGRASP|34.8971|**0.1374**|1.8975||COGRASP|6.8576|0.1512|0.519|
> ||ENHANCER|48.3968|0.2098|1.434||ENHANCER|11.1476|0.2182|0.7545|
> ||**MEHGT-LKG**|**74.0035**|0.1742|**2.4209**||**MEHGT-LKG**|**19.4352**|**0.0967**|**1.2398**|
> |Kunlun Tech|Time-MoE|20.4387|0.3071|0.7905|EVE|Time-MoE|-11.9021|0.2810|-1.1207|
> |300418|KRONOS|25.2375|0.2891|0.9332|300014|KRONOS|0.2922|0.1855|-0.3393|
> ||COGRASP|47.4476|0.3166|1.035||COGRASP|9.3478|**0.0551**|1.1725|
> ||ENHANCER|30.1954|0.3717|0.7710||ENHANCER|19.1132|0.1106|1.4175|
> ||**MEHGT-LKG**|**74.8732**|**0.2419**|**1.4239**||**MEHGT-LKG**|**25.2459**|0.1416|**1.4747**|
> |Zhongji Innolight|Time-MoE|100.7028|0.2621|2.529|Tongwei|Time-MoE|-9.2933|0.1674|-1.5725|
> |300308|KRONOS|144.3722|0.1864|2.7765|600438|KRONOS|9.5537|0.1172|0.3405|
> ||COGRASP|58.8023|**0.1427**|1.8465||COGRASP|3.3198|0.1044|0.3300|
> ||ENHANCER|191.8602|0.3310|2.361||ENHANCER|8.2722|**0.0823**|0.6675|
> ||**MEHGT-LKG**|**274.4909**|0.2735|**3.8083**||**MEHGT-LKG**|**12.3115**|0.0978|**1.1192**|
> |Jingjia Micro|Time-MoE|15.2226|0.1828|0.8595|Sungrow|Time-MoE|5.1654|**0.1547**|0.4335|
> |300474|KRONOS|19.0426|0.1685|1.1811|300274|KRONOS|17.5274|0.2076|1.2285|
> ||COGRASP|27.7979|**0.1503**|1.0455||COGRASP|25.8189|**0.0686**|1.1883|
> ||ENHANCER|34.8507|0.1998|1.3185||ENHANCER|32.0946|0.1493|1.0832|
> ||**MEHGT-LKG**|**58.0853**|0.1776|**1.9637**||**MEHGT-LKG**|**51.1442**|0.1016|**1.7827**|

---

### Author Response · Authors · 2025-11-30
**General Response**

We sincerely appreciate the reviewers’ thoughtful feedback (**FWDJ, kzZm, 4LtN, vvBP**). We are encouraged by the innovative integration of LLM with HGNN (4LtN), the practical effectiveness of FinEX agent for financial text mining (kzZm, vvBP), and the novel edge-centric MEHGT design to enhance relational modeling (kzZm, 4LtN, vvBP). Reviewers also noted the paper is well written and of practical value (FWDJ, 4LtN).

We value the reviewers’ scrutiny and comments (especially about scores 2 and 4), and strengthened the paper with additional experiments, clarifications, and revisions.
## **1. Response to Common Concerns**
**More SOTA baselines** (FWDJ, vvBP): reviewers pointed out that baselines were "outdated" and "More recent baselines should be compared."
* We added 4 SOTA models: Time-MoE (ICLR'25), KRONOS (AAAI'26), COGRASP (IJCAI'25), and ENHANCER (KDD'25) in all experiments (Updated Tables 1, 8–10; Figures 6, 10, 11).
The results show MEHGT-LKG consistently outperforms latest models in prediction, backtesting, and simulated trading, addressing the concern about "limited experimental evidence".

**Model Transportability & Reproducibility**(FWDJ, 4LtN): Reviewers expressed concern about "missing details" in the LLM instruction tuning and computational costs.
* Added Appendix B details the complete flow of FinEX(instruction data, standard format, human verification), and designs an investment platform (Figure 8) to show the transportability of MEHGT-LKG, especially FinEX.
* We added Appendix E reporting hardwares and hyperparameters and Appendix F.1 reporting computation comparison of models to enhance reproducibility.

**More Theoretical Discussion** (FWDJ, vvBP): Reviewers requested "theoretical discussion" to strengthen depth and the motivation for connecting edge-enhanced mechanism and financial market.
* Added Appendix D: (1) FinEX as a semantic denoiser (Information Bottleneck) to maximize signal-to-noise ratio; (2) Edge-enhanced Attention acts as a differentiable gate for topology refinement (Proposition 1), validating the non-trivial, principled architecture; (3) Multimodal Fusion leverages orthogonal noise properties (Proposition 2) to reduce aleatoric uncertainty compared to standard HGT; (4) MEHGT maintains linear scalability $O(|V|d^2+|E|d)$ (Complexity Analysis), directly confirming computational efficiency.
* We improved the theoretical alignment with financial market in Section 3.3 and Appendix F.7.
In Section 3, we detailed how Edge-Enhanced Attention as a "soft relational gate" to modulate flow intensity, while Message Passing aligns subspaces to differentiate financial entities.
Ablation studies confirm that performance gains arise from modeling financial edge semantics.
Interpretability Analysis (Appendix F.7) and Figure 14 show the model correctly assigns attention weights to LLM-extracted event edges, capturing real-world market co-movements and event-driven influence.

**Robustness & Generalization**: Reviewers questioned performance "during volatile periods" and noted the evaluation was confined to specific Chinese stocks. (FWDJ, vvBP)
* We added Appendix F.4 to evaluate the model's robustness under volatile conditions.
New experiments demonstrate MEHGT-LKG adapts faster than baselines to regime shifts. Under bullish shocks, it captures buy signals for excess returns; under bearish shocks, it generates sell signals to mitigate losses; and during technical rebounds, it accurately identifies bottoming signals.
* While broader cross-market evaluation is an ideal direction, the selected representative stocks from diverse industries are widely used in prior work.
These high-profile, high-volatility, and hard-to-predict datasets serve as challenging evaluation benchmark and a strategic stress-test, providing more persuasive validation under complex conditions within current resource and time.

**Performance Variance on Specific Datasets** (kzZm, 4LtN):
* We added average performance comparison (Table 1), and clarified that MCC is the primary metric for financial data.
Prediction, backtesting, and simulated trading show MEHGT-LKG's superiority on specific targets (EVE, Sungrow).
* For weaker case (Kunlun), lower event density limits the benefits of edge enhancement for prediction.
However, backtesting proves that where classification metrics are close, our model generates more profitable trading signals to achieve the highest return across all datasets.

**We clarified our innovations and differences from prior work in response to FWDJ.**

## **2. Summary of Revisions**
We have strengthened the manuscript by adding:
* Complete experiments on new SOTA baselines
* Analysis of the model’s behavior under extreme conditions
* Theoretical discussion of MEHGT-LKG
* Details about FinEX and the process of instruction fine-tuning
* Details about graph data and implement details
* Implementation details and computation comparison
* A practical investment service platform based on MEHGT-LKG

---

### Meta-Review · Area_Chair_FB2k · 2025-12-07

**Summary:**

The submission introduces MEHGT-LKG, a framework that combines an instruction-tuned LLM (FinEX) for financial event extraction with a multimodal heterogeneous graph transformer that incorporates rich edge semantics into both attention and message-passing mechanisms. The system constructs daily event-centric financial knowledge graphs, integrates numerical and textual information, and predicts next-day stock movement. Experiments on 12 Chinese A-share stocks show improvements over baselines in MCC, backtesting performance, and robustness tests. The rebuttal provides extensive architectural clarifications, additional SOTA baselines, theoretical analysis, and detailed implementation information.

**Pre-Rebuttal Reviewer Concerns**

Before the rebuttal, reviewers raised several key issues:

* Novelty and contribution clarity, including potential overlap with existing LLM-constructed financial KGs and related edge-enhanced HGT designs.

* Missing details on FinEX’s instruction-tuning pipeline, data sourcing, quality control, and reproducibility.

* Outdated or incomplete baselines, leaving unclear whether the method advances state-of-the-art performance.

* Inconsistent results, including unexplained ablation outcomes and variability across datasets.

* Generalization limitations, with evaluation restricted to the Chinese market and lacking cross-market or volatility-driven analysis.

* Computational cost and fairness concerns, particularly regarding FinEX.

* Presentation issues, including unclear figures and ambiguity in several methodological descriptions.

**Remaining Concerns After Rebuttal**

The rebuttal is thorough and resolves many of the concerns through additional experiments, expanded baselines, clearer methodological explanations, and substantial implementation details. However, several serious issues remain only partially addressed:

* Novelty remains incremental. Despite detailed contrasts with related work, the architectural contributions may still be viewed as extensions of existing paradigms rather than conceptually new advances.

* Generalization concerns persist. Although new robustness analyses were added, the evaluation remains confined to Chinese A-share stocks. Cross-market validation—important given the proposed model’s intended generality—is deferred to future work, and the added theoretical justifications feel very forceful.

* Presentation quality still requires improvement. While some figures and descriptions were revised, several clarity issues and structural inconsistencies remain.

Overall, the rebuttal significantly strengthens the submission, but the core concerns regarding novelty, generalizability, and presentation are not fully resolved. Given these remaining limitations, I do not recommend acceptance at this stage.

**Reviewer Concerns:**

**Addressed Concerns**

The rebuttal effectively resolves several major issues:

* Missing FinEX details:
The authors provided a clear description of the instruction-tuning pipeline, data sources, and verification steps, addressing concerns about reliability and reproducibility.

* Baseline limitations:
Four recent SOTA models were added, and the updated results show stronger empirical support.

* Methodological clarity:
Additional explanations of the edge-enhanced mechanisms

* Computational cost and reproducibility:
Detailed hardware, training time, and implementation settings now address fairness and reproducibility concerns.

* Inconsistent results and ablations:
The authors clarified dataset-specific behaviors and revised tables to resolve confusion.

* Figure and presentation issues:
Several formatting and clarity problems were corrected.

**Outstanding Concerns**

Some concerns remain only partially addressed:

* Novelty:
Despite detailed comparisons, the contribution may still feel incremental relative to prior LLM-based financial KGs and edge-enhanced HGTs.

* Generalization and grounding:
Evaluation is still limited to Chinese A-share stocks; cross-market validation is absent. The proposed method remains largely engineering. Not clear whether the improvement comes from methodology advancement or merely an engineering artifact.

* Presentation quality:
Although improved, several clarity issues persist.

**Reviewer Scores:**

* Reviewer FWDJ (initial: 4)
The rebuttal addressed missing FinEX details, expanded baselines, and some presentation issues, but substantive concerns about novelty and theoretical depth remain.
Likely post-discussion score: 4.

* Reviewer kzZm (initial: 6)
Issues regarding inconsistent results and labeling were resolved, and additional baselines were added. Generalization concerns remain.
Likely post-discussion score: 6.

* Reviewer 4LtN (initial: 2)
Most technical and empirical concerns were addressed, including baselines, computational cost, reproducibility, ablations, and clarity. However, concerns about novelty, generalization, and computational burden persist.
Likely post-discussion score: 4.

* Reviewer vvBP (initial: 4)
Clarifications on FEKG statistics, multimodality, edge-feature motivation, dataset choice, and formatting improved the paper. The evaluation remains limited in scope.
Likely post-discussion score: 4.

---

### Decision · Program_Chairs · 2026-01-26

Reject